# Mechanics of *Drosophila* wing deployment

Simon Hadjaje[1], Ignacio Andrade-Silva [1,2], Marie-Julie Dalbe[3],
Raphaël Clément [4] ✉ & Joel Marthelot [1] ✉

During their final transformation, insects emerge from the pupal case and deploy their wings within minutes. The wings deploy from a compact origami structure, to form a planar and rigid blade that allows the insect to fly. Deployment is powered by a rapid increase in internal pressure, and by the subsequent flow of hemolymph into the deployable wing structure. Using a combination of imaging techniques, we characterize the internal and external structure of the wing in *Drosophila melanogaster*, the unfolding kinematics at the organ scale, and the hemolymph flow during deployment. We find that, beyond the mere unfolding of the macroscopic folds, wing deployment also involves wing expansion, with the stretching of epithelial cells and the unwrinkling of the cuticle enveloping the wing. A quantitative computational model, incorporating mechanical measurements of the viscoelastic properties and microstructure of the wing, predicts the existence of an operating point for deployment and captures the dynamics of the process. This model shows that insects exploit material and geometric nonlinearities to achieve rapid and efficient reconfiguration of soft deployable structures.

During development, tissues and organs undergo dramatic morphological transformations[1–5]. The biological and physical mechanisms that coordinate these transformations in space and time are a central question in developmental biology. The shaping of the fruit fly wing *Drosophila melanogaster* is a paradigmatic example of such continuous organ-scale transformation, that combines growth and buckling of the wing epithelium to shape the larval and pupal wings into a highly folded origami structure[6–9]. However, how the origami eventually deploys into a functional wing minutes after pupal emergence remains largely unknown. Early works have suggested a mechanical basis for wing deployment in insects, associated with an increase in hemolymph pressure in the open circulatory system[10–13] under hormonal control[14,15], but an integrative physical model of this rapid transformation is lacking. Here, we describe wing actuation as a fluid-structure interaction problem based on experimental characterization of mechanical properties and unfolding kinematics at the organ and tissue scales. We show that wing pressurization, combined with constitutive properties and tissue geometry, results in an operating point that the insect uses for robust deployment. This work sheds light on a neglected but fascinating morphogenetic process at the organ scale. It will also inform engineering strategies for deployable structures whose shape evolves from compact and folded to extended and operational, with applications ranging from biomedical to aerospace[16], and for morphing artificial matter that continuously changes shapes[17], with applications ranging from programmable mechanical metamaterials[18–22] to soft robotics[23–26].

## Results

### Hemolymph injection throughout the wing causes deployment
We first design a protocol to image wing deployment macroscopically. Upon eclosion, the fly is glued to a thin fiber mounted on a micromanipulator, and deployment is recorded using bright-field microscopy from dorsal and lateral views. Figure 1a shows snapshots of the dorsal view of the wing deployment (see Supplementary Movie 1). The wings start out highly folded, with macroscopic folds along the longitudinal veins. The two wings deploy simultaneously, dramatically changing shape from a compact three-dimensional structure to a wing with a transient downward curvature during deployment to a fully deployed flat wing (Fig. 1a and Supplementary Note 2). Additional characteristic features of deployment are revealed by lateral

[1]Aix-Marseille University, CNRS, IUSTI & Turing Centre for Living Systems (CENTURI), Marseille, France. [2]Departamento de Física, Facultad de Ciencias Físicas y Matemáticas, Universidad de Chile, Santiago, Chile. [3]Aix-Marseille University, CNRS, Centrale Mediterranée, IRPHE, Marseille, France. [4]Aix-Marseille University, CNRS, IBDM & Turing Centre for Living Systems (CENTURI), Marseille, France. ✉e-mail: raphael.clement@univ-amu.fr; joel.marthelot@univ-amu.fr

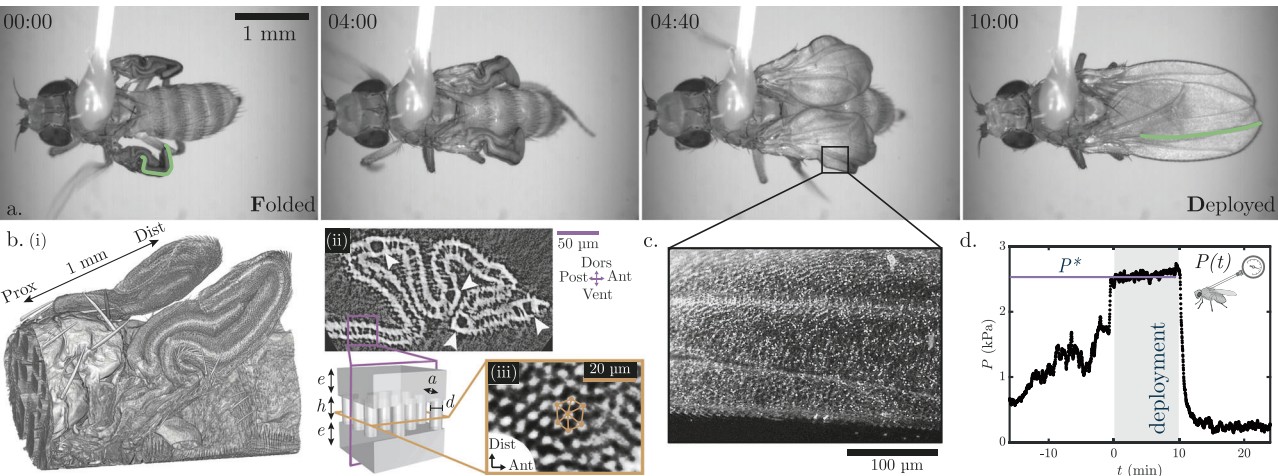

**Fig. 1 | Geometry and pressure in the deploying wing. a** Snapshots of the wings deployment with one of the longitudinal veins highlighted in green. **b** (i) Micro-tomography of the folded wings. (ii) Micro-CT scan cross-section showing macroscopic folds, vein structure (white arrows), and internal pillars. (iii) Perpendicular section revealing the hexagonal pillars organization. The sketch summarizes the wing structure: two plates (thickness $e$) connected with pillars (height $h$, diameter $d$) organized in a hexagonal lattice (interpillar distance $a$). **c** Fluorescent beads (white dots, hemolymph markers) flowing in a deploying wing. **d** In vivo recording of the internal pressure $P(t)$ of a newly emerged insect. Wings deployment takes place throughout the grayed-out segment at a constant pressure $P^*$.

observation. The strong activity of the pharyngeal pump (see Supplementary Movie 2) indicates that the fly swallows air prior to deployment[27], resulting in inflation of the ptilinum, a pouch located on the fly's head. The abdominal contractions that occur simultaneously contribute to an increase in internal pressure, which results in the injection of hemolymph into the wing, causing it to deploy[10]. Note that the air swallowed by the insect remains confined to the digestive tract and that there is no air-liquid interface in the circulation of hemolymph or in the wing during deployment. At the end of the deployment, the ptilinum deflates, and the abdomen relaxes, indicating a decrease in internal pressure. After deployment, the final shape of the wing is stabilized by cuticle tanning and sclerotization[28].

To gain insight into the microscopic structure of the wing, we used high-resolution X-ray micro-CT just before deployment (Fig. 1b(i) and Supplementary Movie 3). A cross-section normal to the proximo-distal axis (Fig. 1b(ii)) reveals that the wing is composed of two continuous, folded plates of thickness $e = 6.5\,\mu m$, connected by regularly spaced pillars of height $h = 7.5\,\mu m$. Veins, hollow tubular structures indicated by white arrows in Fig. 1b(ii), are already visible in the folded wings. The cross-section normal to the dorso-ventral plane reveals that the pillars are arranged in a regular hexagonal lattice of period $a = 6.7\,\mu m$ and diameter $d = 3.3\,\mu m$ (Fig. 1b(iii)). The inside of the wing thus forms a two-dimensional porous network, as sketched in Fig. 1b.

To understand how the rise in pressure guides the structure deployment, we first examine the hemolymph distribution in the wing. Prior to deployment, fluorescent beads are injected into the abdomen of the fly. Figure 1c shows that the beads permeate the entire wing surface during deployment, indicating that hemolymph is injected throughout the entire structure in the two-dimensional porous network, rather than just confined to the veins as observed in adults (see ref. 29 and Supplementary Movie 4), and commonly assumed during deployment[7,30].

The internal pressure during deployment is measured by stinging the thoracic region between the two wings (the scutellum) of newly emerged flies, with a capillary connected to a pressure transducer (see Supplementary Movie 5). The pressure increases over approximately 15 min before reaching a plateau of $P^* \approx 2.5$ kPa, as shown in Fig. 1d. The wings deploy at this constant pressure. Following deployment, the pressure suddenly drops to a lower value of a few hundred pascals. This coincides with the rapid deflation of the ptilinum, which is highly inflated during deployment, and the relaxation of muscle contractions.

The abdomen goes from an elongated and stretched state to a shorter and relaxed state. The abdominal and thoracic muscles involved in the increase in pressure degenerate shortly after deployment[31]. The fly thus transiently generates a few kilopascals to deploy its wings within a few minutes. Remarkably, this transformation bears similarity to that of artificial inflatable structures, such as air mattresses or paddle boards, in which pressure guides in-plane expansion of two plates connected by pillars[32].

## Wings not only unfold but also expand

To quantify deployment, we use the natural landmarks provided by the wing veins. The three-dimensional position of the vein network is reconstructed using micro-CT images in the folded state (Fig. 1b(ii)). This network is preserved in the deployed wing, allowing for a direct comparison between the folded and deployed states (Fig. 2a and Supplementary Note 1). To characterize the kinematics, we first quantify the deployment $\Lambda = L/L_F$, where $L$ is the length of the segment joining the extremities of a longitudinal vein, and $L_F$ is the corresponding initial folded length, as shown in the inset in Fig. 2b. $\Lambda$ is thus dominated by the kinematic opening of macroscopic folds. We find that the deployment dynamics are highly reproducible between individuals (see Supplementary Movie 6), with wing overall length increasing by a factor of 3 within 2 min (Fig. 2b).

Next, we investigate whether the deployment of the folded wing is solely due to the kinematic opening of macroscopic folds, similar to the unfolding of inextensible origami paper[18–20] or the wing of adult Coccinellidae[33], or whether it involves a combination of structural unfolding and tissue expansion. To measure changes in tissue metrics, we compare the arclength of veins in the folded and deployed states. In Fig. 2c, we plot the arclength of the deployed vein $l_D$ against the initial folded arclength $l_F$. To complement these measurements, we track the deformation of three individual veins (marked in blue, yellow, and green in Fig. 2a) using bright-field microscopy observations of dorsal and lateral views (circular markers in Fig. 2c). Notably, we find that the wings not only unfold during deployment but also expand by approximately 1.6 times their original length. Veins oriented in different directions present the same stretch, indicating that expansion is isotropic in the plane. Therefore, the rapid and dramatic deployment of the wings appears to be a dual morphing response that involves unfolding macroscopic folds, and isotropic tissue expansion. Next, we address how this tissue expansion is manifested at the cellular level.

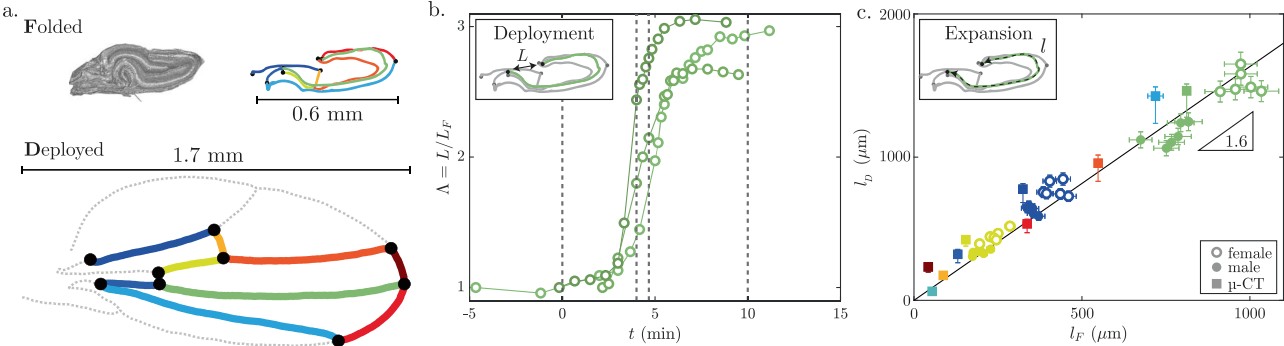

**Fig. 2 | Unfolding and expansion. a** Veins network in the folded state (from micro-CT scans) and in the deployed state (from bright-field microscopy images). Each color represents a specific vein. **b** Macroscopic wing deployment $\Lambda = L/L_F$ as a function of time for 3 different flies. $L$ is the length between the two ends of the longitudinal vein highlighted in green in (**a**) and Fig. 1a, and $L_F$ is the initial folded length. Dotted vertical lines correspond to the snapshots in Fig. 1a. **c** Arclength of deployed veins $l_D$ versus folded veins $l_F$. Each color represents a vein shown in (**a**). Length measurements from micro-CT are represented by square markers, while veins tracked on videos are represented by circular markers (empty for females, full for males). The solid line is a linear fit of all experiments. Folded veins from micro-CT ($l_F$, ■) are coming from measurement on 2 male wings, and the corresponding deployed veins ($l_D$, ■) are obtained from bright-field images of 10 male wings (5 different flies), the data are presented as mean values ± max/min measurements. Veins are tracked on 6 flies deploying their wings, 3 females ○, and 3 males ●; the data are presented as mean values ±5% relative error due to video tracking.

## Epithelial cells stretch while the cuticle only unwrinkles

In order to gain a deeper understanding of the kinematics of expansion at the cellular level, we conduct observations of cross-sections of wings normal to the proximo-distal axis (dotted plane in the sketch in Fig. 3a) using transmission electron microscopy (TEM) in both the folded and deployed states. Micrographs (Fig. 3b) show that the folded wing is composed of two layers of epithelial cells (one ventral, the other dorsal, shown in green) separated by a lumen, into which hemolymph flows. The epithelial cell layer is covered by a thin, wrinkled layer of cuticle (shown in red). As the wings unfold and expand, the cuticle unwrinkles while maintaining a constant thickness of 200 nm in the adult wing. This suggests that the cuticle flattens without undergoing any significant stretching. The corresponding micrograph (Fig. 3b, bottom) is obtained a few hours after deployment, at which point the epithelial cells have already undergone apoptosis, detached, and been washed out of the wing by the action of the wing hearts[34,35], leaving only the cells surrounding the veins. The variation in wing thickness under pressure is measured using two-photon microscopy. We observe that wing thickness is approximatively doubled, increasing the space between the two cell layers during deployment (see wing thickness in Fig. 3e and Supplementary Movie 7).

Using optical profilometry, we measure the topography of the apical surface of the folded and deployed wing (solid line plane in the sketch in Fig. 3a). The wrinkles are organized in a hexagonal pattern, corresponding to the cellular tiling (see Fig. 3c). From cell centroids, we generate the Voronoi tessellation in order to compute the 2D surface area and maximum height of the wrinkles. Figure 3e shows that the height of the wrinkles decreases while the 2D surface area increases during deployment, indicating a general flattening of the wing surface. This increase in cell surface area is confirmed by direct measurement in the folded state and during deployment using spinning disk microscopy (Fig. 3d). Note that the cell shape after deployment shows no significant shape anisotropy, again indicating isotropic tissue expansion. We label the apical surface of the cells, i.e., the surface where the cuticle adheres, to segment individual cells and reconstruct their shape in 3D. In folded wings, the apical surface of the cells is shaped like a volcano, as shown in the inset of Fig. 3c (see also Supplementary Movie 8 and Supplementary Note 4). The integration of the 3D outer surface (see boxplot 3D area in Fig. 3e) confirms that the cuticle has sufficient surface area to unwrinkle without stretching. Moreover, the volcano-shaped wrinkle is isometric to a plane and allows the cuticle to unwrinkle without stretching during wing expansion. In summary, tissue expansion that occurs during wing deployment causes stretching and flattening of epithelial cells, and unwrinkling of the cuticle covering cells' apical surface.

## Coupling between constitutive properties and geometry reveals an operating point for deployment

The kinematic observation of the wing microstructure is summarized in Fig. 4a. The ventral and dorsal plates are each composed of a bilayer of epithelial cells covered by a thin, rigid, and wrinkled cuticle. During deployment and subsequent expansion, the epithelial cells flatten, and their apparent apical surface area increases while the rigid cuticle unwrinkles without stretching. The two plates are connected by regularly spaced pillars organized in a hexagonal lattice. Pillars containing microtubule bundles and microfilaments, observed in the early pupa[36,37] and just before eclosion[38,39], extend from the apical junctions to the basal junctions of the cells, thereby limiting the separation of the two plates under pressure.

To gain insight into the mechanics of the deployment process, we start out by conducting tensile tests on dissected folded wings. One wing extremity is fixed to a motorized linear stage, and a constant velocity of 10 μm/s is imposed while the other wing extremity is attached to a rigid fiber connected to a load sensor (see Supplementary Movie 9). The force $F$ is normalized by the wing's initial cross-sectional surface area $S_0$, which was obtained through micro-CT scanning. In Fig. 4b, we plot $F/S_0$ as a function of $\Lambda = L/L_F$. Two distinct regimes are observed: (i) for small values of $\Lambda \leq 1.25$, the macroscopic folds open, and the material undergoes negligible stretching. The stress remains close to zero, showing that almost no force is required to unfold the origami-like macroscopic folds. (ii) Once the main folds open, the wing also stretches, and stress increases. To disentangle the effects of macroscopic unfolding and tissue stretching, and determine the actual stiffness of the material, we follow the arclength $l$ of a longitudinal vein. In Fig. 4c, we plot the true stress $\sigma = F\lambda/S_0$ under the assumption of incompressibility as a function of the true strain $\ln(\lambda) = \ln(l/l_F)$ to account for large deformations. We extract Young's modulus of the material $E \approx 113$ kPa, which captures the linear relationship observed at low strain (mean value from $n = 6$ experiments). Stiffening is observed at larger strains, which we interpret as the effect of unwrinkling of the rigid cuticle layer as the tissue expands (see Supplementary Note 6 in which this stiffening is replicated in FEM). This phenomenology bears similarities to the stiffening observed in artificial systems, where length variations are buffered by buckling[40,41]. To make sense of these observations, we construct a mechanical model of expansion that couples structural deformation to the

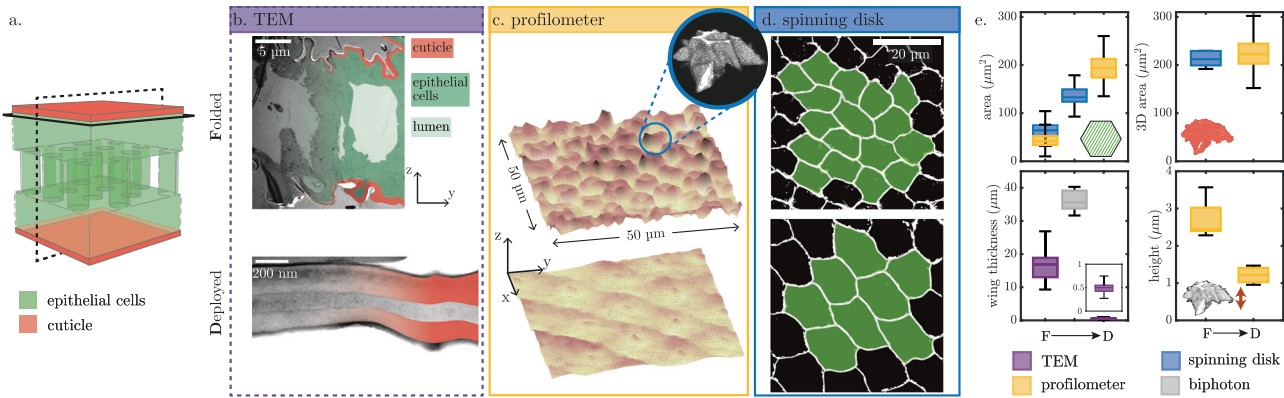

**Fig. 3 | Cells stretching and cuticle unwrinkling. a** Schematic view of a wing microscopic structure: 2 monolayers of epithelial cells (green) covered by a thin rigid cuticle (red) and connected by pillars. **b** Cross-section following the dotted frame in (**a**), transmitted electron microscopy (TEM). **c** Wing surface scans (optical profilometer, 50 × 50 μm²). Top right inset: 3D reconstruction of the volcano-shaped wrinkle on the apical surface of an epithelial cell (spining disk confocal microscopy, Utrophin:GFP). **d** Epithelial cells contour (spining disk confocal microscopy, Ecad:GFP). For (b-d) top images correspond to folded wings, while bottom images correspond to deployed wings. **e** Evolution of the main geometric characteristics from folded (F) to deployed (D) stages (from top left to bottom right diagram): 2D epithelial cell surface area; 3D surface area integration of the cuticle;

total wing thickness; and wrinkles height. Data are obtained by TEM (purple boxes, see (**b**)), profilometer scans (yellow, (**c**)), spinning disk (blue, (**d**)), and two-photon images (gray, see Supplementary Fig. 5d). Each boxplot displays the median, the lower and upper quartiles, and the minimum and maximum values, and is obtained with the following sample sizes. Area: spinning disk (folded wing: 16 z-stacks, deploying wing: 7 z-stacks), profilometer (folded wing: 3 scans, deployed wing: 3 scans). 3D area: spinning disk (folded wing: 5 z-stacks), profilometer (deployed wing: 3 scans). Wing thickness: TEM (folded wing: 2 scans, deployed wing: 3 scans), two-photon (deploying wing: 2 scans). Height: profilometer (folded wings: 8 scans, deployed wings: 3 scans).

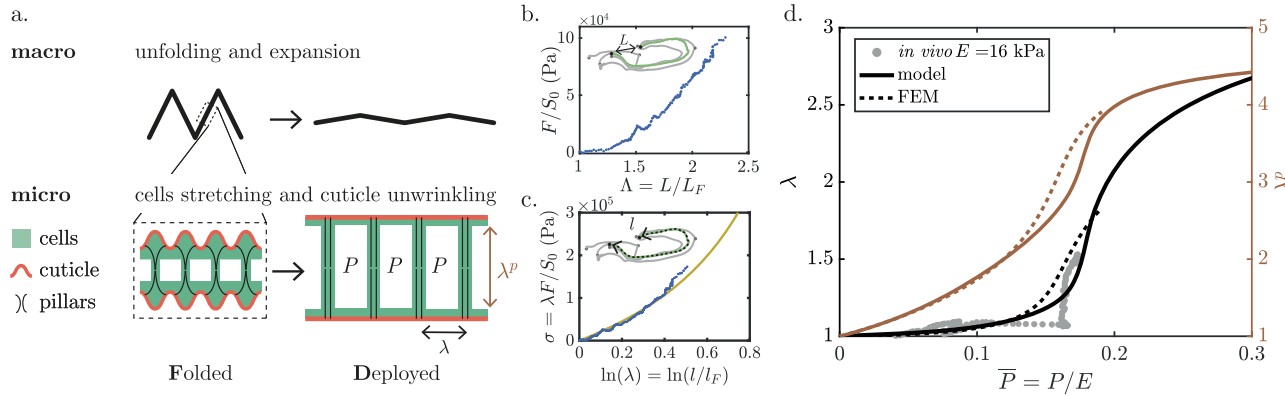

**Fig. 4 | Mechanics of wing deployment. a** Schematic of the two-scale deployment mechanism. Macroscopic scale (top): wing unfolding and expansion. Microscopic scale (bottom): cellular stretching and subsequent cuticle unwrinkling and pillar straightening. The sketch shows a cross-section of the folded (left) and deployed (right) wing structure with wrinkles and pillars under pressure. **b** $F/S_0$ versus deployment $\Lambda = L/L_F$ obtained from the tensile test of the wing. **c** True stress

$\sigma = \lambda F/S_0$ versus true strain $\ln(\lambda) = \ln(l/l_F)$. Yellow line: hyperelastic Gent model ($E = 200$ kPa, $J_m = 20$). **d** Prediction of in-plane stretch $\lambda$ (black, left axis) and pillars vertical stretch $\lambda^p$ (orange, right axis) as a function of the normalized applied pressure $\overline{P} = P/E$ for a hyperelastic Gent material (model: solid lines; FEM: dashed lines). Pressure measurements normalized by a Young's modulus of $E = 16$ kPa are shown as gray markers.

hemolymph pressure increase. Deformation involves in-plane stretch of the plates $\lambda$ and vertical stretch of the pillars $\lambda^p$, as well as thinning of the plates and pillars due to volume conservation (see sketch in Fig. 4a). Stresses in the pillars and plates compensate for the increase in internal pressure $P$. To capture the large-strain stress stiffening due to the in-plane unwrinkling process and the straightening of the microtubule bundles and microfilaments along the pillars, we consider a phenomenological hyperelastic Gent model (see yellow line in Fig. 4c) with two parameters: the Young's modulus $E$ and the stress stiffening captured by the limiting value $J_m$. We find that $\lambda$ and $\lambda^p$ are solutions of a differential non-linear system of equations[42], which we solve numerically (see Methods for computation details). Fig. 4d presents the theoretical predictions for $\lambda$ and $\lambda^p$ as a function of the applied non-dimensional pressure $\overline{P} = P/E$ (black and orange line, respectively). The model is supported by FEM numerical simulations and predicts an operating point $\overline{P} \approx 0.18$, at which the wing undergoes

considerable deformation while maintaining a low operating pressure. At this operating pressure, the predicted vertical stretch of the pillars $\lambda^p \sim 4$ is in good agreement with the measurement of the total wing thickness obtained by two-photon microscopy (Fig. 3e). By fitting our experimental measurements on the $\lambda - \overline{P}$ diagram, we find Young's modulus of 16 kPa (gray markers in Fig. 4d), which is slightly smaller than the value obtained with the tensile test.

To further evaluate the fluid-structure model, we artificially inflate flies sacrificed just after they emerge from their pupal cases (see Supplementary Note 10 and Supplementary Movie 10). We observe a very non-linear behavior of in-plane deformation and pressure, consistent with our theoretical model. When the pressure is maintained at a plateau below 10 kPa, the wings do not deploy. Deployment occurs at operating pressures between 10 and 16 kPa. At pressures above 17 kPa, the pillars can no longer support the stress and eventually rupture, and we observe the formation of blisters and balloon-like wings. With

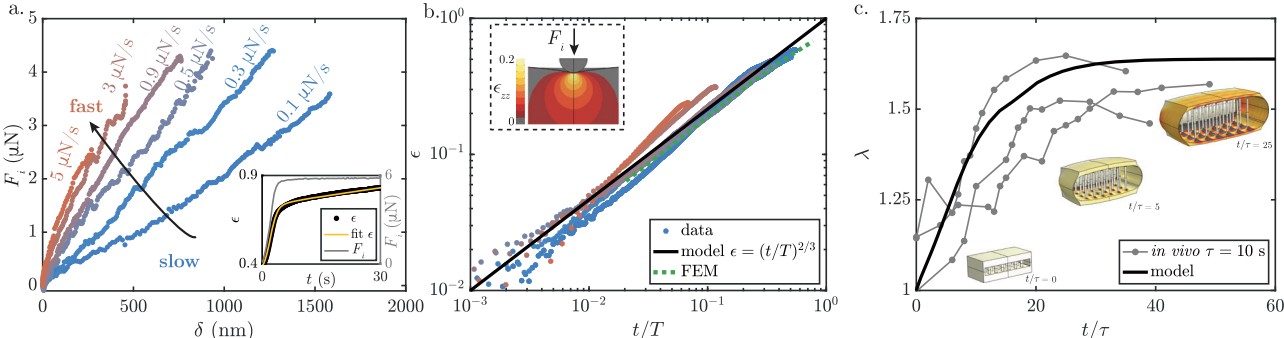

**Fig. 5 | Dynamics of wing deployment. a** Force-displacement curves obtained from nano-indentation experiments on folded wings, from slow indentation ($\dot{F}_i = 10^{-1}\,\mu$N/s, blue) to faster ($\dot{F}_i = 10^{1}\,\mu$N/s, red). Inset: Deformation $\epsilon = (\delta/R)^{1/2}$ measured over time for a creep experiment during which a constant-force $F_i$ is applied (gray line). Experimental data (in black) are fitted with the model (yellow line). **b** $\epsilon$ versus normalized time $t/T$: both experimental data (from **a**, colored markers) and FEM numerical simulations of viscoelastic bilayer indentation (green dashed line) collapse on the model $\epsilon = (t/T)^{2/3}$ (black line). Inset: local strain in the indentation direction $\epsilon_{zz}$ from FEM simulations. **c** In-plane stretch $\lambda$ as a function of $t/\tau$. Experimental measurements (normalized with $\tau = 10$ s): gray markers; model: solid line. Snapshots of FEM simulations at $t/\tau = 0$, 5, and 25 are shown (color bar: strain in one of the in-plane directions).

Young's modulus measured in the tensile test, we obtain an operating point of $P/E \sim 0.15$, which is consistent with the one predicted by the model (see Fig. 4d).

## Deployment dynamics is governed by the viscoelastic properties of the tissue

We now seek to gain insight into the dynamics of deployment. A first hypothesis would be that the observed timescale results from the fluid-structure interaction between the flow of viscous hemolymph and the deformation of the flexible porous network. For a typical hemolymph viscosity $\eta_f \sim 1$ mPa.s[43,44], the characteristic timescale $\eta_f L^2/(Ehe)$ based on the coupling between wing elasticity and fluid viscosity[45] is on the order of milliseconds. This is several orders of magnitude shorter than the observed deployment timescale, indicating the presence of an additional dissipation mechanism.

In order to ascertain the viscoelastic properties of the wing, we perform nano-indentation measurements on the folded wing. A force $F_i$ is applied at different loading rates while the indentation depth $\delta$ is measured (Fig. 5a). The effective stiffness appears to increase with the loading rate, a feature typical of viscoelastic materials. This indicates the existence of another source of dissipation in the material, with a characteristic time $\tau = \eta/E_i$ where $\eta$ is the viscosity of the material and $E_i$ the effective elastic response in indentation, with contributions from the epithelial cells as well as the cuticle.

Creep experiments (inset of Fig. 5a), which consist of a loading phase at a constant rate followed by a constant-force relaxation phase, are accurately represented by a three elements Maxwell-Jeffrey model comprising a dashpot in series with a dashpot in parallel with an elastic spring of stiffness $E_i$. The system exhibits two distinct timescales: a short timescale $\tau_1 = 1.9 \pm 0.3$ s and a long relaxation time, $\tau_2 = 37 \pm 12$ s. The fit provides an estimate for the effective elastic response of the bilayer in indentation $E_i \sim 1.6$ MPa. This effective stiffness value is in good agreement with finite elements method (FEM) simulations (inset of Fig. 5b), obtained by indenting a bilayer composed of a thin elastic film (cuticle with Young's modulus $E_f = 100$ MPa and thickness 200 nm) deposited on an elastic substrate (epithelial cells with Young's modulus $E = 100$ kPa).

In the nano-indentation experiment, the long relaxation time can be neglected at the scale of the experiment (see Methods), and the model is reduced to a simple Kelvin-Voigt model consisting of a dashpot in parallel to an elastic spring with a single characteristic time. Assuming Hertz contact, the strain of a viscoelastic Kelvin-Voigt material under indentation can be described by the equation $\tau\dot{\epsilon} + \epsilon = F_i/(E_i R^2 \epsilon^2)$, where $\epsilon = (\delta/R)^{1/2}$ and $R = 4.7\,\mu$m the radius of the indenter. For a constant loading rate $\dot{F}_i$, and in the limit of $t \ll \tau$, the

strain can be approximated by $\epsilon \sim (t/T)^{2/3}$, where $T = (2R^2 E_i \tau/(3\dot{F}_i))^{1/2}$. The experimental data collapse on this master curve (Fig. 5b), with a single value, $E_i\tau$, being fitted to the curves. A similar procedure is applied to six samples, each subjected to a loading pattern of 21 loading and unloading cycles with increasing loading rates. This yields $E_i\tau \approx 10{-}47$ MPa.s and $\tau \approx 6{-}29$ s, in agreement with previous measurements of viscous time in epithelial cells[46]. FEM indentation of the bilayer, which accounts for the viscoelastic properties of the substrate with a viscous timescale $\tau = 10$ s, as depicted by the dashed green curve in Fig. 5b, aligns with the experimental curves (see also Supplementary Note 9).

We extend our quasistatic model of wing deployment to the dynamic case by taking into account the viscoelasticity of the material. In this scenario, the hyperelastic material has an internal timescale $\tau$, and is subjected to a pressure step $\overline{P}^*$ at $t = 0$. The model for $\lambda$ as a function of $t/\tau$ (solid line in Fig. 5c) is supported by FEM simulations (inset Fig. 5c and Supplementary Movie 11). Snapshots of the FEM are shown at $t/\tau = 0$, 5, and 25, illustrating how the system deforms both in-plane and vertically. The system ultimately reaches its equilibrium state within a characteristic time of $\sim 10\tau$. This prediction is compared with experimental measurements of the time evolution of $\lambda$ obtained by bright-field microscopy. We plot $\lambda$ as a function of time normalized by the measured internal timescale of the material obtained with the indentation experiment: $\tau = 10$ s. The experimental data align with the theoretical prediction and the FEM without additional fitting parameters. The equilibrium state of the deployment and its dynamics are well captured by our model, in which a visco-hyperelastic material is subjected to an increase in pressure. This strongly suggests that deployment is solely due to an increase in pressure and that its dynamics are regulated by the internal timescale of the wing material itself.

## Discussion

Wing deployment in insects results from the coupling between material properties, geometry, and fluid loading. Using the fruit fly *Drosophila melanogaster* as a model system, we first describe the complex structure of the folded wing, which consists of two sheets of epithelial cells separated by a lumen and covered by a wrinkled rigid cuticle. The sheets are held together by a hexagonal lattice of pillars. Deployment occurs a few minutes after emergence, as hemolymph flows in the lumen under a pressure increase of a few kPa. During deployment, we observe the macroscopic unfolding of the wing folds, along with tissue expansion. This expansion is due to the flattening and stretching of epithelial cells and to the unwrinkling of the cuticle that covers the cells' volcano-shaped apical surface. We then characterize the

mechanical properties of the wings. Taking into account the geometry and strain-stiffening hyperelasticity due to cuticle unwrinkling and pillar stretching, we develop an effective model that analytically predicts in-plane expansion around an operating value of $P/E$. Finally, we extend our analysis to the dynamics of deployment. We model the viscoelastic behavior of epithelial cells using nano-indentation experiments. The implementation of dissipation in the effective model captures the dynamics and is in agreement with experimental observations and FEM calculations.

Our model is based on an effective description of the complexity of the biological response and depends only on a finite set of parameters measured independently in the experiments: an elastic Young's modulus $E$, a strain stiffness parameter $J_m$, and a viscous time $\tau$ of the material. Despite its simplicity, the model gives a remarkable account of the kinematics and dynamics observed in vivo. In particular, it predicts the existence of an operating pressure $P/E = 0.18$ at which most of the deployment occurs. The pressure measured in vivo is slightly lower than the pressure predicted by the model or observed when we artificially inflate sacrificed flies after their removal from their pupal case. This discrepancy may be due to the fact that, for the sake of simplicity, we model the horizontal plates and vertical pillars with the same effective constitutive law. Another potential source of discrepancy may be that tensile and indentation tests are carried out on dissected folded wings. Evaporation or cuticle hardening could artificially stiffen the moduli compared to in vivo conditions.

We observe that the wings tend to curl up when deployed under artificial conditions. This probably indicates differential variations in the elastic properties of the ventral and dorsal plates. It may also be indicative of an active regulating mechanism that ensures in-plane deployment of the wings when the fly is alive. Although this curling is reminiscent of the phenotype observed in *Curly* mutant flies, we show that the two phenomena are quantitatively different by comparing the curvature during deployment (see Supplementary Note 2 and Supplementary Movie 12).

We proposed an unprecedented morphometric and mechanical characterization of insect wing deployment. A physical model summarizes our results and shows that insects exploit the coupling between geometric and material nonlinearities to achieve efficient deployment around an operating point. This physical actuation is biologically triggered after eclosion by a hormone-controlled behavior: the repeated intake of air and abdominal muscle contractions leading to internal pressure increase. This provides an interesting example of post-developmental morphogenesis: the groundwork is laid during larval and pupal development when the wings form, grow, undergo eversion, and finally fold in the pupal case. However, upon eclosion the wings are still folded, thus not yet functional. The massive and rapid shape change that occurs as a result of air ingestion and abdominal contractions is mandatory to make them operational, and can be delayed by the insect if conditions are unfavorable[27]. Wing deployment is, therefore, a peculiar morphogenetic process: on the one hand, it takes place after development and does not require cellular growth or mechanical activity, and on the other, it requires active behavior of the animal to occur.

Our work also paves the way for future studies aimed at understanding the pressure-driven expansion of deployable biological structures and addressing the wide diversity of wing sizes and shapes in insects[47,48]. In addition to its interest in fundamental biomechanics, improving our understanding of the mechanics of such deployable structures will find applications in the fields of material morphing physics and the engineering of flexible structures.

## Methods
### Fly imaging
**Strains and stock.** Wild-type Oregon-R flies are used throughout this study, with the exception of spinning disk fluorescent images, for which Ecad:GFP and Utrophin:GFP flies are used. Flies are maintained on standard fly food at 18°C in a temperature-controlled chamber with 12h light-dark cycles.

**Bright-field microscopy.** To record the deployment of the wings, a fly is collected immediately after eclosion from its pupal case and placed on a $CO_2$ pad to anesthetize and manipulate it. A micromanipulator is used to approach and glue a thin fiber (120 μm diameter) to the dorsal part of the thorax of the fly, ensuring that the wings are free to move. The fly is then raised above the pad, and the deployment of the wings is recorded from both a top view using a binocular microscope (Leica MZ16) and a lateral view using a Leica Z16 APO lens connected to two cameras (Imaging Source 37DFKBUX264).

**Hemolymph flow visualization.** To visualize hemolymph flow through the wings during deployment, fluorescent beads are injected into the fly's abdomen (500 nm pink beads, Drummond nanoject II). The fly is then attached to a thin stick and placed under a fluorescent binocular microscope (Nikon SMZ1000).

**Profilometer characterization of the wrinkles.** Wings are dissected and molded on uncured elastomer (Kerr Polyvinylsiloxane Impression Material type 3). The use of a mold provides a rigid sample after polymerization eliminating hairs that could mask surface features. Subsequently, the molds are scanned using a profilometer with an optical pen (micromeasure 2, optical pen CL1-MG140, 1.3 μm lateral resolution, 48 nm axial resolution). The surface 3D coordinates are analyzed using ImageJ and Matlab. Any overall surface curvature is corrected, the Voronoi tessellation of the wrinkles is generated, and the hexagonal organization, typical height, distance between wrinkles, and surface area of individual cells in both folded and deployed stages are extracted (see Fig. 3c, yellow boxplot Fig. 3e for quantification and Supplementary Note 5).

**Micro-CT.** Micro-CT is performed at CEREGE (UM34, MATRIX platform). Flies ($N = 2$) are chemically fixed (glutaraldehyde + phosphate-buffered saline + paraformaldehyde), dehydrated with ethanol and dried with supercritical $CO_2$ (critical point bypass), and placed in a Kapton tube for imaging. A male is scanned over a 12-h period, with the scan centered on the wings with an x20 objective (resolution of 0.8 μm/voxel). A female is scanned over a 27-h period, with the scan centered on a specific region of interest on the right wing with an x40 objective (0.32 μm/voxel). ImageJ (version 2.14.0/1.54f) is employed for post-treatment, 3D image visualization (plugins 3D viewer and volume viewer), veins tracking, and geometrical characterization of the wings. The geometrical parameters (plates thickness $e$, pillars diameter $d$, pillars height $h$, and interpillar distance $a$) are measured by threshold, ultimate points, and Voronoi tesselation on high-resolution scans (see Fig. 1b(ii-iii) and Supplementary Note 3).

**Transmitted electron microscope.** Transmitted electron microscopy (TEM, FEI Tecnai g2 200 kv) is conducted at the IBDM electronic microscopy platform. Wings are dissected from newly emerged flies (i), and adult flies 4h after wings deployment (ii). Wings are then fixed using cryofixation and cryosubstitution. The total wing thickness is quantified before and after deployment using ImageJ (see boxplot wing thickness in Fig. 3e for quantification).

**Spinning disk confocal microscope.** A spinning disk confocal microscope is used to obtain the epithelial cells contour and the 3D apical cell surface from folded and deployed wings. The microscope is equipped with CSU-X1 spinning disk unit (Yokogawa) mounted on a Nikon Ti eclipse stand, 100x/1.49NA378 Nikon objective, EMCCD Andor iXon3 DU897, and MicroManager software. Folded wings are dissected from newly emerged flies, placed on a microscope slide, and

covered with a drop of oil to avoid desiccation. A similar protocol is employed for deployed wings, which must be dissected shortly after deployment. Indeed, epithelial cells undergo apoptosis and are washed out from the wing after deployment, such that mature wings no longer exhibit fluorescence. The cell contours of Ecad:GFP flies are obtained with the Tissue Analyzer plugin in ImageJ, and quantification (cell surface area, cell-to-cell distance) is conducted in Matlab (Math-Works R2023b). The apical cell surface is imaged by performing a high-resolution $z$-stack on Utrophin:GFP wings. Utrophin, an Actin-binding protein, tags the apical cell cortex. The scans are visualized and thresholded in ImageJ, and the 3D surface areas of individual cells are integrated into Matlab (see Supplementary Movie 8 and inset in Fig. 3c).

**Two-photon microscopy.** A two-photon microscope, equipped with a Zeiss 510 NLO (Inverse-LSM), a femtosecond laser (Laser Mai Tai DeepSee HP), and a 40x/1.2 C Apochromat objective, is used to measure the total wing thickness in vivo. The fly thorax is attached to a stick, and the insect is placed such that the wing plane is aligned with the optical path of the microscope. Wing deployment is concomitant with body movement on the order of the size of the insect (~1 mm), which makes the measurement of the wing thickness (~10 μm) during deployment in a live fly somewhat challenging. Nevertheless, we could capture fast $z$-stacks of the wing section starting from the wing distal tip. The analysis is conducted using ImageJ (see Supplementary Movie 7 and Fig. 3e for quantification). We measure a total wing thickness (composed of the height of the pillars $h$ and the thickness of the plates $e$) of ≈36 μm during deployment, i.e., a 2-fold increase compared to folded wings cross-section. This yields a stretch of the pillars in the vertical direction of $\lambda^p \approx 4.8$.

## Mechanical measurements

**Pressure.** The internal pressure of the insect is quantified as the wings deploy using an in-house experimental setup, mounted on a vibration-isolated optical table. The pressure probe is composed of a pressure sensor (Honeywell 24PCBFA6D) connected to a rigid microfluidic channel. A glass capillary (Clark capillary glass GC100-15, 1 mm outside diameter, 0.58 mm internal diameter, tip size 10–50 μm) previously pulled (Sutter Instrument P97-4832) is filled with low viscosity silicon (AS 4 Wacker-Chemie, 0.004 mPa.s) and secured to one end of the channel, while a motorized syringe at the other end allows for volume control of the system. The tip of the syringe is filled with deionized water to create an oil/water meniscus in the large section of the capillary. This prevents an oil/hemolymph interface at the capillary tip, which could result in an additional Laplace pressure. The position and the shape of the oil/water meniscus are tracked to correct for any additional pressure due to a possible flux (hydraulic resistance) or non-flat meniscus (Laplace pressure). Flies with folded wings are collected immediately following their emergence from the pupal case and placed on the $CO_2$ pad. A micromanipulator is utilized to puncture the insect scutellum with the capillary mounted on the pressure probe. The $CO_2$ pad is then removed, and the wing deployment is observed from both top and side views. A LabVIEW interface (NI LabVIEW 2017) is employed to synchronize the pressure measurements, volume control of the syringe, and the images captured by the two cameras. Reproducibility was tested for $N = 4$ different flies (see Supplementary Note 10).

**Tensile tests.** Tensile tests are conducted on folded fly wings to characterize their mechanical properties. Wings are dissected from newly emerged flies and glued at their proximal extremity to a motorized linear stage (glue Loctite AA 352, linear stage PI VT-80). The distal extremity of the wing is glued to a load sensor (Magtrol MBB-02-0.05) previously calibrated with known loads. A water tank is placed in close proximity to the wing to prevent desiccation. Following the

application of the glue, a period of several minutes is allowed for it to harden before the linear stage is moved (see Supplementary Movie 9). The entire setup is controlled via a custom Labview interface, which includes monitoring the experiments with two cameras (providing top and side views), controlling the linear stage, and recording the force from the load sensor. The setup is placed on a vibration-isolated optical table. The raw measured force, $F$, is normalized by the cross-sectional surface area $S_0$ of the folded wing obtained via micro-CT scan to extract the engineering stress $F/S_0$ (see Supplementary Note 3 for details on obtaining $S_0$). Upon stretching, we assume incompressibility and the cross-sectional surface area of the wing to decrease as $S = S_0/\lambda$. Consequently, the true stress $\sigma$, can be expressed as $\sigma = F/S = \lambda F/S_0$. We use ImageJ to track one of the longitudinal veins (underlined in green Fig. 4b and Supplementary Fig. 8b) during stretching and compute the tissue stretch $\lambda = l/l_F$ where $l$ is the vein arclength, and $l_F$ is the corresponding initial folded length. The macroscopic deployment, $\Lambda$, is defined as the ratio between the end-to-end distance of the vein, $L$, and its corresponding initial value, $L_F$ (see Fig. 4b for a plot of $F/S_0$ versus $\Lambda$). Figure 4c depicts the true stress, $\sigma$, as a function of the true strain $\ln(\lambda)$. A linear fit at small stretch yields the Young's modulus, $E$.

**Nano-indentation.** Nano-indentation experiments are conducted on folded and adult wings (Supplementary Notes 7, 9) using a nano-indenter (Hysitron TI Premier). Folded wings are dissected from flies just after emergence from the pupal case. The wing is then carefully placed on a thin layer of viscous epoxy glue coated on a microscope slide. This procedure allows the wing to slightly embed in the epoxy while leaving the wing's top surface in the open air. Once the epoxy has cured, providing a rigid substrate for the wing, the microscope slide is secured in the nano-indenter with magnets. Two types of tests are conducted: (i) loading cycles with increasing loading rates (maximum force 5 μN, rates $\dot{F}_i = 10^{-1} - 10^1$ μN/s) and (ii) creep tests with constant-force (3 experiments at $F_i = 3.2$, 6, and 8.1 μN). All experiments are performed within a maximum of 30 min after wing dissection. During this period, no time-dependent effects on the wing's mechanical properties are observed. The raw force, $F_i$, and indentation depth, $\delta$, are analyzed using Matlab.

An indentation model of a viscoelastic material is constructed in order to extract the effective elasticity in indentation, $E_i$, and the viscosity, $\eta$, of the wing. The loading cycle tests are well captured by a simple Kelvin-Voigt material undergoing Hertz-like indentation. The total stress of a Kelvin-Voigt material is the combination of a linear elastic response and a viscous dissipation, such that $\sigma_i = E_i \epsilon + \eta \dot{\epsilon}$. This model is then coupled with Hertz's theory of contact (see Supplementary Note 7 for prediction of elastic Hertz contact), $\sigma_i = F_i/(R\delta)$ and $\epsilon = (\delta/R)^{1/2}$. The strain of a viscoelastic Kelvin-Voigt material under indentation is thus the solution of the equation:

$$\tau \dot{\epsilon} + \epsilon = \frac{F_i}{E_i R^2 \epsilon^2}, \tag{1}$$

with $\tau = \eta/E_i$. Indentation cycles are performed at constant loading rate, so that $F_i = \dot{F}_i t$, with $\dot{F}_i$ increasing with each cycle. In this case, strain can be solved from Eq. (1):

$$\epsilon = \left[ \frac{\dot{F}_i \tau}{3R^2 E_i} \left( 3\frac{t}{\tau} - 1 + e^{\frac{-3t}{\tau}} \right) \right]^{1/3} \underset{t \ll \tau}{\sim} \left( \frac{3\dot{F}_i}{2R^2 E_i \tau} \right)^{1/3} t^{2/3} = \left( \frac{t}{T} \right)^{2/3} \tag{2}$$

where $T = (2R^2 E_i \tau/(3\dot{F}_i))^{1/2}$. The experimental data are observed to collapse to a single line with a slope of 2/3 in a log–log scale in Fig. 5b. The data are fitted to the model in this limit to find $\eta = E_i \tau \approx 10 - 47$ MPa.s. This range corresponds to the dispersion obtained by applying this procedure on six different samples, each

subjected to a loading pattern of 21 loading and unloading cycles with increasing loading rates.

The experimental creep test comprises two distinct phases: a loading phase during which the force is increased at a constant rate, and a subsequent phase in which the force remains constant (see the force curve in gray in the Inset of Fig. 5a). The resulting strain $\epsilon$ relaxed slowly over time, a behavior which is well captured by a three elements Maxwell-Jeffrey model comprising a dashpot $\eta_2$ in series to a Kelvin-Voigt element (a dashpot $\eta_1$ in parallel with a spring $E_i$). When coupled with Hertz contact, we obtain:

$$\tau_1\tau_2\ddot{\epsilon} + \tau_2\dot{\epsilon} = \frac{F_i}{E_iR^2}\frac{1}{\epsilon^2} + \frac{\tau_1+\tau_2}{E_iR^2}\left(\frac{\dot{F}_i}{\epsilon^2} - \frac{2F_i\dot{\epsilon}}{\epsilon^3}\right) \quad (3)$$

where $\tau_1 = \eta_1/E_i$ and $\tau_2 = \eta_2/E_i$. Eq. (3) is solved numerically, and the creep experimental data are fitted with $E_i$, $\tau_1$, and $\tau_2$ as fitting parameters (see Inset Fig. 5a, yellow line, for the fit).

Note that in the nano-indentation experiments, the solutions of the three-element Maxwell-Jeffrey model reduce to the simpler Kelvin-Voigt model described above. A loading of the form $F_i = \dot{F}_it$ is imposed and we look for solutions of the form $\epsilon \approx (t/T)^\alpha$. For $\alpha = 2/3$, there are two groups of terms that evolve as $-t^{-1/3}$ and $-t^{-4/3}$. For short times, the dynamics are governed by terms in the power of $-4/3$. Balancing these terms, we find the expression $\epsilon(t) \approx (t/T_0)^{2/3}$, with

$$T_0 = \left(\frac{2R^2E_i\tau_{eff}}{3\dot{F}_i}\right)^{1/2}, \quad (4)$$

where $\tau_{eff} = \tau_1\tau_2/(\tau_1 + \tau_2)$ which tends toward $\tau_1$ for $\tau_1 \ll \tau_2$. Numerical simulations of Eq. (3) for different values of $\dot{F}_i$ confirm the above scaling.

## Model

**Wing geometry and force balance.** We consider two infinite plates of thickness $e$, connected with pillars of diameter $d$, height $h$, organized in a hexagonal lattice of pitch $a$. We assume that both plates undergo spatially homogeneous equibiaxial extension in $\mathbf{e}_x$ and $\mathbf{e}_y$, so that the principal stretches in the plates read $\lambda_x = \lambda_y = \lambda$. We assume that the material is incompressible, which implies that $\lambda_x\lambda_y\lambda_z = 1$ and $\lambda_z = 1/\lambda^2$. For the pillars, we consider a uniaxial extension in the $\mathbf{e}_z$ direction. Incompressibility and isotropy lead to the following equation $\lambda_x^p\lambda_y^p\lambda^p = (\lambda_x^p)^2\lambda^p = 1$, superscript $p$ refers to the pillars, and we denote $\lambda_z^p = \lambda^p$, so $(\lambda_x^p)^2 = 1/\lambda^p$.

A force balance in the plates relates the Cauchy stress in the principal directions $\sigma_{xx} = \sigma_{yy} = \sigma$ to the internal pressure $P$[42]. As the plates become thinner by a factor of $1/\lambda^2$ and the pillar height increases by $\lambda^p$, we obtain the equation $2\sigma e/\lambda^2 = Ph\lambda^p$ and:

$$\sigma = P\frac{h}{2e}\lambda^p\lambda^2 = P\frac{\Psi}{1-\Psi}\lambda^p\lambda^2 \quad (5)$$

with $\Psi = h/(h+2e)$ the relative height of the pillars in the non-deformed geometry.

Similarly, for the pillars, the longitudinal Cauchy stress $\sigma_z^p = \sigma^p$ compensates for the increase in internal pressure, so that $\sigma^pA^p/\lambda^p = P(A\lambda^2 - A^p/\lambda^p)$ where $A^p$ is the area of the pillars in the plane and $A$ is the total area of the plates in the plane. We introduce the density of the pillars in the plane $\Phi = A^p/A = \pi/(2\sqrt{3})(d/a)^2$, such that:

$$\sigma^p = P\left(\frac{2\sqrt{3}}{\pi}\left(\frac{a}{d}\right)^2\lambda^p\lambda^2 - 1\right) = P\left(\frac{\lambda^p\lambda^2}{\Phi} - 1\right) \quad (6)$$

In order to solve these equations and obtain the values of $(\lambda, \lambda^p)$ as a function of $P$, it is necessary to define the material's constitutive law.

**Linear elastic model.** In this section, we assume that the structure is composed of an isotropic homogeneous linear elastic material with Young's modulus $E$ and Poisson ratio $\nu$. For the stress in the plane of the plates, the theory of linear elasticity yields:

$$\sigma = \frac{E}{(1+\nu)(1-2\nu)}\left[(1-\nu)\epsilon + \nu(\epsilon + \epsilon_z)\right] \quad (7)$$

with the in-plane strain $\epsilon = \lambda - 1$ and in the normal direction $\epsilon_z = \lambda_z - 1$. We assume that the plates are under plane stress $\sigma_z = 0$, the thickness $e$ being small compared to the other dimensions. We obtain $(1-\nu)\epsilon_z + 2\nu\epsilon = 0$, which gives $\epsilon_z = -2\nu\epsilon/(1-\nu)$, which we replace in Eq. (7) to obtain:

$$\sigma = \frac{E}{1-\nu}\epsilon = \frac{E}{1-\nu}(\lambda - 1) \quad (8)$$

Similarly, for the pillars, we write the general expression for the stress in the principal direction:

$$\sigma^p = \frac{E}{(1+\nu)(1-2\nu)}\left[(1-\nu)\epsilon^p + \nu(\epsilon_x^p + \epsilon_y^p)\right] \quad (9)$$

We explicit $\epsilon_x^p = \epsilon_y^p$ by writing the stress in one of the horizontal directions:

$$\sigma_x^p = -P = \frac{E}{(1+\nu)(1-2\nu)}\left[(1-\nu)\epsilon_x^p + \nu(\epsilon_y^p + \epsilon^p)\right] \quad (10)$$

Indeed, the pillars are compressed by the internal pressure. We find $\epsilon_x^p = -(1+\nu)(1-2\nu)P/E - \nu\epsilon^p$, that we inject in Eq. (9) to get:

$$\sigma^p = E\epsilon^p - 2\nu P = E(\lambda^p - 1) - 2\nu P \quad (11)$$

We now couple this elastic linear constitutive law for plates and pillars in Eq. (8) and (11) with the force balance in the in-plane and vertical directions expressed in Eq. (5) and (6). We then obtain the following system of coupled non-linear equations:

$$\begin{cases} \frac{E}{1-\nu}(\lambda - 1) = P\frac{h}{2e}\lambda^2\lambda^p & (12a) \\ E(\lambda^p - 1) - 2\nu P = P\left(\frac{2\sqrt{3}}{\pi}\left(\frac{a}{d}\right)^2\lambda^2\lambda^p - 1\right) & (12b) \end{cases}$$

or equivalently with the non-dimensional pressure $\overline{P} = P/E$:

$$\begin{cases} \frac{\lambda-1}{1-\nu} = \overline{P}\frac{\Psi}{1-\Psi}\lambda^2\lambda^p & (13a) \\ \lambda^p - 1 - 2\nu\overline{P} = \overline{P}\left(\frac{\lambda^2\lambda^p}{\Phi} - 1\right) & (13b) \end{cases}$$

Eq. (13b) gives $\lambda^p = (1 + \overline{P}(2\nu - 1))/(1 - \overline{P}\lambda^2/\Phi)$ when $\overline{P}\lambda^2/\Phi \neq 1$. This result is then plugged into Eq. (13a) to obtain the following equation for $\lambda$:

$$\lambda^3 + \left\{(1-\nu)\frac{\Phi\Psi}{1-\Psi}\left[1 + \overline{P}(2\nu - 1)\right] - 1\right\}\lambda^2 - \frac{\Phi}{\overline{P}}\lambda + \frac{\Phi}{\overline{P}} = 0 \quad (14)$$

This linear elastic model predicts a finite pressure at which the stretch, both in the horizontal and vertical direction, diverges. This would be detrimental to the wing, which requires a large but finite expansion at a given pressure. Furthermore, this simple model does not take into account the microstructure of the wing: (i) the epithelial cells are covered by a rigid cuticle initially wrinkled that yields to a strain stiffening as the tissue expands; and (ii) the two bilayers are connected with microtubules that uncoil as the plates move apart until the stress in the pillars diverges when these filaments get straight. The aforementioned features, as illustrated in Fig. 4a, prompt us to adopt a

more realistic model that considers large strains and nonlinearities (strain stiffening) in the material.

**Hyperelastic Gent model.** We use the phenomenological Gent hyperelastic model to describe the strain stiffening. This constitutive model is characterized by two parameters: the shear modulus $\mu = E/(2(1+\nu))$ and a limiting value $J_m$ of the left Cauchy-Green deformation tensor first invariant. The second Piola-Kirchhoff stress tensor $S$ depends on the strain energy density $W$:

$$\mathbf{S}_G = 2\frac{\partial W}{\partial I_1}\mathbf{I} - p\mathbf{C}^{-1} \; ; \text{with} \; W = -\frac{\mu J_m}{2}\ln\left(1 - \frac{I_1 - 3}{J_m}\right) \quad (15)$$

where $\mathbf{C} = \mathbf{F}^T\mathbf{F}$ is the right Cauchy-Green deformation tensor, $\mathbf{F}$ is the deformation gradient tensor, $I_1 = \text{tr}(\mathbf{C})$, $J = \det(\mathbf{F})$, $\mathbf{I}$ is the identity, and $p$ is a reactive pressure due to incompressibility constraint.

The plates are under plane stress $S_{zz} = 0$ yielding the in-plane stress:

$$S = \left(1 - \frac{1}{\lambda^6}\right)\frac{\mu J_m}{J_m - 2\lambda^2 - \frac{1}{\lambda^4} + 3} \quad (16)$$

The pillars are compressed in the horizontal direction by internal pressure, $S_{xx}^p = S_{yy}^p = -\lambda^p P$, so that:

$$S_{zz}^p = -\frac{P}{(\lambda^p)^2} + \frac{\mu J_m}{J_m - (\lambda^p)^2 - \frac{2}{\lambda^p} + 3}\left(1 - \frac{1}{(\lambda^p)^3}\right) \quad (17)$$

The Cauchy stress relates to the second Piola-Kirchhoff by $\sigma_G = J^{-1}\mathbf{F}\mathbf{S}_G\mathbf{F}^T$. We now couple the hyperelastic constitutive law with the force balance in the in-plane and vertical directions expressed in Eq. (5)–(6). We then obtain the following system of coupled non-linear equations[42]:

$$\begin{cases} \left(\lambda^2 - \frac{1}{\lambda^4}\right)\frac{\mu J_m}{J_m - 2\lambda^2 - \frac{1}{\lambda^4} + 3} = P\frac{h}{2e}\lambda^2\lambda^p & (18a) \\ \left((\lambda^p)^2 - \frac{1}{\lambda^p}\right)\frac{\mu J_m}{J_m - (\lambda^p)^2 - \frac{2}{\lambda^p} + 3} = P\frac{2\sqrt{3}}{\pi}\left(\frac{a}{d}\right)^2\lambda^2\lambda^p & (18b) \end{cases}$$

or in non-dimensional form with $\overline{P} = P/E = P/(2\mu(1+\nu))$:

$$\begin{cases} \left(\lambda^2 - \frac{1}{\lambda^4}\right)\frac{J_m}{2(1+\nu)\left(J_m - 2\lambda^2 - \frac{1}{\lambda^4} + 3\right)} = \overline{P}\frac{\Psi}{1-\Psi}\lambda^2\lambda^p & (19a) \\ \left((\lambda^p)^2 - \frac{1}{\lambda^p}\right)\frac{J_m}{2(1+\nu)\left(J_m - (\lambda^p)^2 - \frac{2}{\lambda^p} + 3\right)} = \overline{P}\frac{\lambda^2\lambda^p}{\Phi} & (19b) \end{cases}$$

The in-plane stretch $\lambda$ and in the pillars, $\lambda^p$ are predicted numerically by solving the coupled non-linear equations as a function of the applied normalized pressure $\overline{P} = P/E$ (see Fig. 4d and Supplementary Note 8 for a parametric study of the model).

**Dynamical model.** We take into account the viscous dissipation measured experimentally by extending the Kelvin-Voigt model to a hyperelastic material. The second Piola-Kirchhoff stress tensor in the viscous branch is $\mathbf{S}_V = \tau\dot{\mathbf{S}}_G$ such that the total stress is $\mathbf{S} = \mathbf{S}_G + \tau\dot{\mathbf{S}}_G$. This yields $\sigma = J^{-1}\mathbf{F}\mathbf{S}\mathbf{F}^T$ and, replacing the Cauchy stress in Eq. (5) and (6), we obtain:

$$\begin{cases} \frac{J_m}{J_m - J_1}\left\{\lambda^2 - \frac{1}{\lambda^4} + 2\tau\dot{\lambda}\left[\frac{3}{\lambda^5} + \left(\lambda^2 - \frac{1}{\lambda^4}\right)\frac{2\lambda + \frac{2}{\lambda^3}}{J_m - J_1}\right]\right\} = 2(1+\nu)\overline{P}\frac{\Psi\lambda^2\lambda^p}{1-\Psi} & (20a) \\ \frac{J_m}{J_m - J_1^p}\left\{(\lambda^p)^2 - \frac{1}{\lambda^p} + 2\tau\dot{\lambda^p}\left[\frac{3}{2(\lambda^p)^2} + \left((\lambda^p)^2 - \frac{1}{\lambda^p}\right)\frac{\lambda^p - \frac{1}{\lambda^p}}{J_m - J_1^p}\right]\right\} + \frac{2\tau\overline{P}\dot{\lambda^p}}{\lambda^p} = 2(1+\nu)\overline{P}\frac{\lambda^2\lambda^p}{\Phi} & (20b) \end{cases}$$

where $J_1 = I_1 - 3 = 2\lambda^2 + 1/\lambda^4 - 3$ and $J_1^p = I_1^p - 3 = (\lambda^p)^2 + 2/\lambda^p - 3$. The expansion dynamics are predicted numerically by solving Eq. (12a)-

(12b). In particular, Fig. 5c shows the in-plane stretch $\lambda$ as a function of the normalized time $t/\tau$ for a constant applied pressure $\overline{P} = 0.18$. The experimental data (gray markers), for which the time is normalized by the measured viscous dissipation time $\tau = 10$ s, exhibit a good match with the model, without fitting parameters.

## Finite element method

We perform finite element method (FEM) simulations using the commercial software COMSOL Multiphysics (version 6.1) to (i) build a fundamental understanding of indentation experiments and (ii) support our model of wings expansion.

**Indentation.** We quantitatively study the effect of a bilayer on the indentation measurement by performing 2D axi-symmetric FEM simulations of a rigid spherical indenter (radius $R = 4.7\,\mu m$, $E_{ind} = 110$ GPa, $\nu_{ind} = 0.3$) applying a force to a bilayer composed of a soft cellular-like substrate (thickness 200 $\mu m$, width 400 $\mu m$) covered by a more rigid cuticle-like film 200 nm-thick. The substrate is modeled by a viscoelastic Kelvin-Voigt material with Young's modulus $E = 100$ kPa and viscous time $\tau = 10$ s, while the film is purely elastic with Young's modulus $E_f = 100$ MPa. The boundaries are fixed at the bottom and the side of the system while loading-unloading cycles are applied on the bilayer by imposing the force with increasing loading rates (see Supplementary Note 9).

**Expansion.** We numerically test our wing expansion model by conducting 3D FEM simulations of the pressure-driven deformation of a structure exhibiting the geometry of the wing: two plates with a thickness of $e = 6.5\,\mu m$ connected with pillars of height $h = 7.5\,\mu m$, diameter $d = 3.3\,\mu m$, and an interpillar distance $a = 6.2\,\mu m$ in a square lattice. $a$ is chosen so that the in-plane pillars density in the FEM corresponds to that measured on micro-CT scans (see Fig. 1b). The system is modeled using an isotropic and incompressible hyperelastic Gent material (Young's modulus $E = 100$ kPa, limiting value $J_m = 20$) and solved for 1/8th of the geometry with symmetric boundary conditions. We first perform a stationary study, in which we impose an increasing pressure from $P = 0$ to 20 kPa and measure the in-plane stretch $\lambda$, and vertical stretch, $\lambda^p$ (see Fig. 4d black and orange dashed lines and Supplementary Fig. 11). We then turn to a time-dependent study by adding a viscous dissipation (Kelvin-Voigt viscous time $\tau = 10$ s) in the material. We impose a constant pressure $P/E = 0.18$ at $t = 0$ and measure $\lambda$ and $\lambda^p$ over time. Snapshots of 1/2 of the geometry are shown at different times $t/\tau$ in Fig. 5c (see also Supplementary Movie 11).

## Statistics and reproducibility

No statistical method was used to predetermine the sample size. Given the experimental constraints we aimed to obtain a sample size large enough to allow testing statistical significance. The number of samples is indicated in the legend of each figure. Plots were created in Matlab version R2023b. In box plots, the median is indicated by a central thick line, while the interquartile range (containing 50% of the data points) is outlined by a box. Whiskers indicate the minimum and maximum data range. No data were excluded from the analyses.

## Reporting summary

Further information on research design is available in the Nature Portfolio Reporting Summary linked to this article.

## Data availability

Source data are provided in this paper as a Source Data file.

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

## Acknowledgements

We thank Yoël Forterre and Murillo Bento Santana for discussions, Marin Lebreuilly for help with initial indentation experiments, Nicolas Brouilly and the electron microscopy facility of the IBDM, Perrine Chaurand and the MATRIX plateform for the microtomography, Sham Tlili for help with two-photon microscopy, Benoit Dehapiot for help with hemolymph flux visualization, Claire Chardès for help with spinning disk confocal microscopy, Nathalie Ehret for help with profilometry and William Le Coz for help with automation. This work was supported by Agence National de la Recherche (ANR Tremplin BioSoftAct and ANR BioFibMat ANR-21-CE30-0019), S.H. acknowledges funding from Fondation pour la

Recherche Médicale (FRM: FDT202304016556). I.A.-S. acknowledges funding by the FONDECYT Postdoctoral Grant 3230753.

## Author contributions

S.H., I.A.-S., M.-J. D., R.C., and J.M. conceived and designed the study. S.H. performed the experiments and quantified the data. S.H., I.A.-S., and J.M. conceived the model and the FEM numerical simulations. S.H., I.A.-S., M.-J. D., R.C., and J.M. interpreted the data and wrote the manuscript.

## Competing interests

The authors declare no competing interests.
