## [Transparent Peer Review File · Nature Communications]

Mechanics of Drosophila wing deployment

Corresponding Author: Dr Joel Marthelot

Version 0:

Reviewer comments:

Reviewer #1

(Remarks to the Author)
Please attached PDF.

Reviewer #2

(Remarks to the Author)

The idea is interesting, the research is well done, potential applications are numerous. And the manuscript is well written, videos are stunning. So I will strongly recommend publication, after minor revision, see below.

- p 7: Curly, show images, perform a qualitative then quantitative comparison
- p 7: evaporation : unclear ; more generally, the three last sentences of p. 7 ("While the flies... is alive") deserve to be rewritten
- p 9: is it "provides evidence" or only "strongly suggests"?
- p 10: same as p 7, develop "Curly" and "evaporation"
- p 11: A male, a female: is it a question of sex? Or is it only a way to explain that $N=2$?
- p 13: engineering stress, true stress: discuss better which one should be used.
And discuss also which strain should be used (eg true / Hencky strain)
- p 13: Hertz contact. In general it applies to an elastic sphere and an elastic flat substrate. But in the reader's mind it is often associated with an elastic sphere on a much stiffer substrate. Here it seems the sphere is much stiffer than the substrate. Please clarify.
- p 14: after "uniaxial extension" add " λ^p ";
and check the last expression, I would have thought that $\lambda^p_{p_z} = \lambda^p$ and that $\lambda^p (\lambda^p_{p_x})^2 = 1$.
- p 16: add a few words to discuss physically how eqs 13 (or 18) reflect the same physics as eqs 5-6.
- p S3: I can guess what 'thermally contracting' means but a few words of explanation would help. Moreover, please explain why it is the best method to produce wrinkles.

Typos: p. 11 quantified; p 14 an increasing loading rates; p 17 non-dimensionnal; p S1 "as sketched Fig. 1B"

Reviewer #3

(Remarks to the Author)

Version 1:

Reviewer comments:

Reviewer #1

(Remarks to the Author)

The authors have made significant (and appropriate) changes in response to my report. I think the paper should proceed to publication.

Reviewer #2

(Remarks to the Author)

I find the revision adequate.

Reviewer #3

(Remarks to the Author)

Response to the Reviewer's Comments for Manuscript NCOMMS-24-31374-T

Wing deployment in *Drosophila melanogaster*

by S. Hadjaje, I. Andrade-Silva, M.-J. Dalbe, R. Clément and J. Marthelot

Please refer to the “diff version” appended at the end of this document for a detailed account of all the changes, corrections, and additions to the revised version of the manuscript. The original text that was modified/deleted is struck-through (in red), and the revised/new text is underlined (in blue).

Response to the report of Reviewer 1.

This work provides novel insight into the expansion of insect wings and is likely to be of broad interest to the readership of Nature Communications. Below, I have provided comments and questions that I would like the authors to address before this paper can proceed further.

We thank the reviewer for their appreciation of our work and for recommending our manuscript for publication in Nature Communications. We have addressed the comments as detailed below and thank the referee for helping us improve the quality of our article.

My main comments center on some of the numbers given by the authors for various quantities. These are intended as support for the physical picture they present; however, there is often not much detail of how these estimates are arrived at. Two of these seem particularly important to me.

1. It's unclear what the Young's modulus measured in the tensile tests (i.e. $E \sim 100\text{kPa}$) corresponds to physically. Given its small magnitude, it seems very unlikely that it is actually a material property of the insect wing. (I assume the wing is largely chitin, which has a Young's modulus on the order of GPa, see [doi:10.1039/C9NR02870F](https://doi.org/10.1039/C9NR02870F), for example.)

The reviewer is right that the adult wing is composed of two bonded layers of cuticle, a thin, flat, layer of sklerotized chitin with a total thickness of ≈ 500 nm in the intervein regions. The stiffness of the adult wing has been measured in the literature using AFM indentation techniques, and the Young's modulus of the cuticle is of the order of GPa.

Motivated by the reviewer comment, we perform nano-indentation experiments on fully expanded adult wings. The wings are dissected, placed on a thin layer of superglue coated on a glass plate and installed in the nano-indenter. We impose loading and unloading force cycles and obtain the $F_i - \delta$ curve shown in the Inset of Response Fig. 1a. We perform multiple indentations on different areas of adult wings on 5 different samples.

Response Fig. 1: Nanoindentation of adult wing. (a) Unloading phase of the indentation curves of an adult wing on a logarithmic scale. Inset: raw F_i - δ curve. **(b)** Histogram of Young's modulus E_c of adult wings. Each color corresponds to a different fly, the black line is the cumulative histogram.

For a rigid spherical probe indenting a soft, elastic, planar sample, Hertz model of contact simplifies to:

$$F_i = \frac{4}{3} \frac{E_c}{1 - \nu^2} R_{ind}^{1/2} \delta^{3/2} \quad (1)$$

with E_c the cuticle Young's modulus, $\nu = 0.5$ its Poisson's ratio, and R_{ind} the radius of the indenter.

We apply this model to elastic discharge in order to exclude the plastic effects observed during loading. In Response Fig. 1a, we plot the unloading curve F_i as a function of δ on a logarithmic scale. The curves align with a slope of $3/2$. We apply a linear fit to F_i as a function of $\delta^{3/2}$. The coefficient of the fit gives $4E_c R_{ind}^{1/2} / (3(1 - \nu^2))$, so we can inverse this expression to find the sample Young's modulus E_c .

All measurements are shown in Response Fig. 1b, each color corresponding to a different fly. The different data for a color correspond to different indentation times and positions on the same fly. The cumulative histogram is represented by the black contours and shows a peak at $E_c \approx 1$ GPa. Some sample appears stiffer (see the cyan bar in the histogram around 2-3 GPa) than others (blue sample ≈ 0.2 GPa). One hypothesis that could explain the differences from one wing to another is that the cuticle undergoes sklerotization after deployment, which tends to increase cuticle stiffness [1]. In our series of experiments, we did not systematically record the time elapsed since wing expansion. The adult wing stiffness measurements are very consistent with previous measurements of adult wings [2], confirming our experimental protocol and Young's modulus extraction procedure.

We have added this measurements in the Supplementary information 'Nanoindentation of the cuticle stiffness of adult wings'.

The authors should clarify this, and whether they believe they have measured an effective stretching modulus and what its origin is. For example, if it is not a material modulus, then could it be caused by some adhesion between initially contacting folds that requires a force to unfold, or perhaps intrinsic curvature could have a similar effect?

Response Fig. 2: (a) FEM numerical simulation of the stretching of a wrinkled bilayer and associated stress-stretch curve. Color code represents the strain in the tensile direction ϵ_{XX} . Red dotted line: elastic response of the substrate alone. (b) ϵ versus normalized time t/T from nanoindentation experiments. Data (colored markers, using 10 bins per sample to average all experiments, standard deviation shown with errorbars) and FEM (dashed lines, from slow indentation $\dot{F}_i = 10^{-1} \mu\text{N/s}$ in dark green, to faster $\dot{F}_i = 10^1 \mu\text{N/s}$ in gray) collapse on the model $\epsilon = (t/T)^{2/3}$ (black line). (c) FEM numerical simulation of the indentation of a bilayer composed of a visco-elastic cellular substrate (Young's modulus $E = 100 \text{ kPa}$, internal time $\tau = 10 \text{ s}$) covered by an elastic cuticle (thickness $t = 200 \text{ nm}$, Young's modulus $E_f = 100 \text{ MPa}$). This snapshot is taken while the indenter (radius $R = 4.7 \mu\text{m}$) imposed a force $F_i = 5 \mu\text{N}$ at rate $\dot{F}_i = 10^{-1} \mu\text{N/s}$, corresponding to the dark green curve in (b).

While the stiffness of chitin is high, the effective stiffness measured in tensile tests also depends on the geometry. The complex structure of the folded wing results in a much lower effective stiffness. The microstructure of a section of the wing (studied by TEM in Fig. 3b of the manuscript) shows that the folded wing is composed of two bilayers: two monolayers of soft epithelial cells covered by a rigid wrinkled cuticular layer (see the schematic in Fig. 4a of the manuscript). The reviewer is right to say that it is this intrinsic curvature on the microscopic scale (the unwrinkling of the cuticle) associated with the stretching of the soft epithelial substrate that is indeed at the origin of the material's effective non-linear constitutive law.

To determine the effective stiffness of the bilayer during unwrinkling of the cuticle, we study the mechanical response of a bilayer using numerical simulations. Response Fig. 2a shows σ as a function of the stretch λ . The curve shows two distinct regimes: first, at moderate stretch ($\lambda < 1.5$), the bilayer is mainly dominated by stretching of the soft substrate, while unwrinkling of the rigid cuticle film has minimal impact on the overall stiffness of the bilayer; then as the wrinkles disappear (intermediate snapshot $\lambda \sim 1.5$), the system becomes stiffer and the mechanical response is dominated by the stretching of the rigid cuticle film.

Motivated by the reviewer's comment, we compare the mechanical response of the bilayer with that of the substrate alone (red dotted line). We note that in the first linear regime, the effective stretching modulus of the bilayer is similar to the elastic response of the substrate with a Young's modulus $E \sim 100 \text{ kPa}$, which physically corresponds to the material properties of the epithelial cell substrate.

Finally, we verify that the properties of the bilayer are consistent with the nanoindentation measurements performed on the folded wing. By performing numerical calculations with the stiffness values of a thin cuticle

film of $E_f = 100$ MPa deposited on a viscoelastic substrate of epithelial cells with stiffness $E = 100$ kPa and viscous timescale $\tau = 10$ s, we recover indentation curves similar to the experimental observations (Response Fig. 2b and c). We note that the stiffness of the cuticle, that is not completely sclerotized, is lower than that measured in the adult wing, as reported in literature [1, 3].

Other sources of macroscopic effective stretching proposed by the reviewer, such as adhesion between contacting folds or initial macroscopic fold curvature, are negligible throughout the wing expansion phase. However, these effects are corrections that should play a key role in macroscopic folds unfolding and should be included to model the very low initial stiffness observed at small $\Lambda < 1.2$, when plotting the stress F/S_0 as a function of apparent stretch Λ (Fig. 4b of the manuscript). In this paper, we did not model this low initial stiffness (structural unfolding of macroscopic folds) and instead focused on tissue expansion (Fig. 4c of the manuscript).

Motivated by the reviewer's comment, we have added a more explicit schematic to explain the bilayer structure in Fig. 4a of the manuscript. We have also added the elastic response of the substrate in the section 'Wings mechanical properties measured through tensile tests' in Supplementary information.

Similarly, could the authors comment on whether the measured stiffness is isotropic or direction-dependent (as one might expect from unfolding wrinkles).

Response Fig. 3: Micro-wrinkles hexagonal tiling in folded (a) and adult (b) wings. (i-ii) Wing surface topography; (iii) Fourier transform of the wing surface that gives three peaks separated by an angle of $\approx 60^\circ$, the distance from the center gives the inverse of the pattern wave length.

We cannot directly measure tissue anisotropy by tensile methods due to the initial macroscopic folding of the wing, and are therefore forced to infer potential stiffness anisotropy by 3 indirect methods.

(1) The reviewer is right that we expect an anisotropic response if we consider the unfolding of 2D wrinkles such as those represented in our finite element calculations (see Response Fig. 2a) or observed when we make a TEM section of the wing (see Fig. 3b of the manuscript). However, these 2D wrinkles are a simplification of the wing's highly regular three-dimensional structure, characterized by volcano-like wrinkles featuring hexagonal tiling associated with the organization of the wing's epithelial cells. To demonstrate this hexagonal tiling, we perform a Fourier transform of the folded and extended wing surface from profilometric measurements of the wing topography (see Response Fig. 3). In both cases, we find a hexagonal organization of the volcano-like wrinkle characterized by 60-degree angles in reciprocal space, with a larger pattern wavelength for the extended wing. While this hexagonal organization might suggest an anisotropic response of the structure, we observe that the peaks in inverse space remain on a circle in the deployed state, indicating that the extension is isotropic in-plane.

(2) At the macroscopic scale, we characterize the stretching of the veins during wing expansion under an isotropic pressure load. We observe that veins oriented in different directions present the same stretch, indicating that extension is isotropic in the plane (see Fig. 2(a) and (c) of the manuscript).

(3) At the microscopic scale, we have imaged the contour of epithelial cells using spinning disk confocal microscopy and Ecad:GFP in Figure 3d of the manuscript. In the intervein region, we observe that while cell surface area increases during expansion under an isotropic pressure load, the cell shape remains isotropic hexagonal after expansion.

We have added a section 'Isotropic in-plane properties of the tissue' in the Supplementary information to discuss and recap these observations.

Finally, similar phenomenology has been reported before in other systems and has been referred to as the changes in length being 'buffered' by buckling (see doi:10.1126/science.aag0677, for example).

Thank you very much for the suggested reference, which is very pertinent. In Grandgeorge et al. the initial effective stiffness is dominated by the surface energy associated with the surface change of the fluid encapsulating the wrinkled material, whereas we observe that the effective stiffness of the wings is dominated by the expansion of the epithelial cell layer (the bilayer substrate), but the phenomenology of the stiffening observed is very similar.

We have added a sentence in the manuscript to underline the similarities with the phenomenology observed in other systems where changes in length are 'buffered' by buckling and we have added the reference to Grandgeorge et al. [4] and to a review by Dominic Vella [5].

2. Similarly, the authors assume that there must be viscoelastic evolution because the dynamics of inflation are much slower than the time scale $\eta_f L^2 / E h e$. I am not particularly familiar with ref. [40], so it would be good to outline the essential ingredients for this estimate. As an alternative, I wondered whether older work on airway reopening by Heil and co-workers (see for example doi:10.1017/S0022112002003452) gives a similar estimate

for the inflation time. Could the authors comment on this time scale and whether the slow peeling dynamics that Heil considers are accounted for in ref. [40]?

The inflation of a compliant flow conduit under the effect of an imposed axial pressure has been studied extensively in the context of compliant microfluidics and soft robotics under different geometries.

Elbaz and Gat [40] have theoretically studied the viscous-elastic interaction of a closed axisymmetric shell containing a viscous liquid. A simple scaling analysis comparing the deformation of the elastic structures with the internal viscous dissipation leads to the characteristic timescale $T \sim \eta_f L^2 / E h e$. The authors show that the problem can be reduced to an inhomogeneous linear diffusion equation from which analytical solutions can be derived. In particular, they show that by applying a Heaviside function of the pressure signal at the tube inlet, pressure and mass flow propagate along the tube with a characteristic time captured by T but with a large prefactor of the order of 10.

The insect's blood circulatory system is an open system. This means that blood flows throughout the insect's body. The digestive system, on the other hand, is a closed system, strictly separated from the blood system. The air that the insect swallows to increase its internal pressure therefore remains confined to the digestive system, and there is no air-liquid interface in the bloodstream. In the case of insects, the wing lumen is initially filled with liquid (hemolymph or insect blood). During expansion, only liquid is injected into the wing, and there is no interface between the liquid hemolymph and the air. There is no dissipation associated with lumen detachment, no air-liquid interface and no capillary number as observed in airway reopening and studied in detail by Heil and colleagues.

We have rewritten the manuscript to emphasize that there is no air-liquid interface during expansion.

I also have some more minor comments:

3. As I understand it, the authors are proposing that the material stretching is minimal in the wing expansion (i.e. there is no real strain of the material at the microscopic scale). Instead, the observed evolution is due to a combination of unfolding of macroscopic folds and microscopic (cell-scale) wrinkles. To me at least, it was unclear when the authors were referring to an effective/apparent stretching of the tissue compared to a material stretching. Could the authors clarify when they are referring to macroscopic vs. cellular observations? For example, I believe the section 'The wing not only unfolds but also stretches' refers to macroscopic observations of effective stretching, whilst 'Apparent stretching relies on unwrinkling at the cellular scale' explores the cause of these macroscopic observations at a cellular level.

The reviewer is right to say that there is negligible strain in the rigid cuticle during wing deployment. However, there is a stretching of the soft substrate of epithelial cells that dominates the rigidity during deployment.

Thanks to the reviewer's remark, we realized that the initial version of the paper was not sufficiently clear, due to the interchangeable terminology of 'deployment', 'expansion', 'stretching', 'unfolding', etc. We have therefore modified the wording, so that it is now consistent throughout the manuscript. The general phenomenon is now

Response Fig. 4: Macroscopic unfolding and expansion vs. microscopic stretching and unwrinkling. (a) General sketch illustrating the two-scale wing deployment process. Top schematics: unfolding and wing expansion. Bottom schematics: microscopic stretching of the epithelial cells (in green) and subsequent unwrinkling of the cuticle (red) and straightening of the pillars (black filaments). **(b)** Deployment Λ due to the opening of the folds (solid dark green line) and wing expansion λ (dashed line) measured on a wing during the deployment process.

called wing deployment: the fly ‘deploys’ its wings. This prompted us to change the title of the article to ‘Wing deployment in *Drosophila melanogaster*’. Deployment, from a macroscopic point of view, is largely represented by $\Lambda = L/L_F$. It can be qualitatively decomposed into 1/ ‘unfolding’ of the macro-folds, and 2/ ‘expansion’ of the entire wing surface (largely represented by $\lambda = l/l_F$). Expansion is dominated by the ‘stretching’ of soft cell substrate, but also causes the ‘unwrinkling’ of the cuticle (which takes place with negligible stiffness, as pointed out by the reviewer).

We now use consistent terminology throughout the paper. We have added a more detailed sketch of how the deployment decomposes at the micro and macro scales (Fig. 4a of the manuscript and Response Fig. 4a), and now compare the dynamics of $\Lambda = L/L_F$ and $\lambda = l/l_F$ (Supplementary Fig. 1c and Response Fig. 4b).

4. I am also a little concerned by the model the authors present for the effective stretching of the tissue using an isotropic hyperelastic model. Could the authors comment on the validity of this model since here the material is unwrinkling not stretching, is this process isotropic? or elastic? Furthermore, the authors have also assumed in-plane equibiaxial extension and incompressibility. I wouldn’t at first expect that these assumptions hold since the pillars break the in-plane symmetry and unwrinkling isn’t necessarily incompressible.

The elastic description of the material is motivated by observations of cases where the expansion process is interrupted. Response Fig. 5 shows snapshots of a wild-type fly that begins to expand its wings, but whose internal pressure decreases before the process is complete. We observe that the wings fold back, imperfectly, but with their longitudinal vein folds and marginal folds. This reversibility of folding prompts us to use an elastic description to model the deformation of the structure.

Response Fig. 5: Elastic recovery after the interruption of wing deployment in a wild-type fly.

The effective stretching model of the bilayer, a soft epithelial cell substrate covered by a wrinkled rigid layer shown in Response Fig. 2a, has the same characteristics as a hyperelastic model. We use a Gent model, where the initial stiffness E captures the stretching of the soft substrate and J_m models the sharp strain stiffening observed when the wrinkles have disappeared and the cuticle begins to stretch.

The reviewer is right that the pillars break the in-plane symmetry and that this is not taken into account in our simple model. In particular, the model does not take into account the deformation offset at the connection between the pillars and the plates, i.e. during inflation, the plates stretch while the pillars shrink in the xy -plane. This model is therefore more accurate when $\Psi \rightarrow 1$ and $\Phi \rightarrow 0$, i.e. when the pillars are slender structures and their physical connection to the plates does not greatly affect plate stretching. This is the reason for the small discrepancy observed between numerical predictions that take into account the actual geometry and connection of the pillars to the plate and the simple model seen in Supplementary Figure 5a. However, this approximation seems reasonable, as confirmed by the closeness of the model to the numerical calculation.

We have added snapshots of the elastic recovery in the Supplementary material section ‘Wings mechanical properties measured through tensile tests’ and Supplementary Movie S13. We have added a discussion of the limit of our simple model in Supplementary material section ‘Model parameters’.

5. The ‘volcano-like’ 3D microscopic structure is mentioned a couple of times in the main text, but only shown in the SI. I think it would be useful to have an image in the main text too.

The ‘volcano-like’ 3D microscopic structure, is shown in Fig. 3c of the manuscript. We have tried to make this point more legible by explicitly naming the structure in the figure legend.

In addition, as it is difficult to get understand the geometry of the structure from a single image, we have added bright-field microscopy observations and segmentation (shown in Response Fig. 6).

We have added a direct reference to Fig. 3c of the manuscript which shows the ‘volcano-like’ 3D microscopic structure. We have also added a new section ‘Volcano-like shape of the apical cell surface’ in the Supplementary material.

6. Before equation (19a), “dimentionnal” should be “dimensional”.

Response Fig. 6: Volcano-like shape of the apical cell surface. (a) brightfield micrographs showing folded wing epithelial cell contour (highlighted in white) and wing hairs (dark shadows). (b) Utrophin:GFP folded wing individual cell. (c) resulting segmentation of the white square in (b). (d-f) 3D reconstruction of Utrophin:GFP folded wing individual cell, showing a 3D perspective, top- and side-view respectively.

We thank the reviewer. We corrected this typo.

Response to the report of Reviewer 2.

The idea is interesting, the research is well done, potential applications are numerous. And the manuscript is well written, videos are stunning. So I will strongly recommend publication, after minor revision, see below.

We thank the referee for their strong support for manuscript publication.

- p 7: *Curly*, show images, perform a qualitative then quantitative comparison

Response Fig. 7: *Curly* mutant wing expansion at the organ level. Snapshots of the wings expansion from a top and lateral view.

Curly mutant is a phenotype widely used to track and identify mutations. In genetics, when working with genes that are not associated with an immediately visible phenotype, these mutations are often combined with the *Curly* mutation to highlight the mutation of interest by macroscopic observation. Motivated by the reviewer's remark, we characterize the kinematics of the *Curly* mutant during expansion (see Response Fig. 7). Although the initial macroscopic folding pattern of the wing is identical to that of the wild type, the wings unfold out of plane when expansion begins. In adults, the wings are characterized by a permanent upward curvature. The adult *Curly* is therefore unable to fly, and moves around in small hops.

A few studies have attempted to elucidate the physical mechanism leading to the *Curly* phenotype, but a mechanical description remains missing. Using transmission electron microscopy, Hurd et al. [6] observed that the wings of *Curly* mutants occasionally show abnormal pairing between the dorsal and ventral cuticle. One hypothesis is that these anomalies lead to a reduction in the surface area of the dorsal wing, causing the wing to curve upwards.

Response Fig. 8: Curvature during deployment in wild type wings. Profile of the marginal vein (i.e. the vein that forms the contour of the wing) from side-view (a) and top-view (b). Time is color-coded from light to dark green. (c) 3D reconstruction of a wing during deployment.

We then quantify the curvature during expansion, by tracking the profile of the marginal vein from a lateral view (Response Fig. 8a) and top view (Response Fig. 8b). We fit the lateral view with a circle to calculate the average curvature $k = 1/R$ (with R the radius of the fitted circle). The corresponding schematic three dimensional reconstruction of the wing during expansion is shown in Response Fig. 8c.

We now quantitatively evaluate the difference between (1) wild type fly, (2) *Curly* mutants and (3) artificial inflation of wild-type fly by repeating the same process. Response Fig. 9a shows the side-view of the marginal vein 3 minutes after onset of wing expansion for a wild type (green), *Curly* (yellow) and artificially inflated fly (brown). In the wild-type, the wing is initially flat when folded ($k = 0$), and curves downwards as soon as it begins to unfold ($k < 0$). It then returns to a flat expanded plate ($k = 0$). For *Curly* mutants and artificial inflation, the wing immediately curves upwards ($k > 0$) at the beginning of deployment, and reaches a plateau: the adult fly has a positive curvature. Fig. 8b shows that while artificially inflated wings qualitatively resemble the phenotype observe in *Curly*, it is quantitatively different with *Curly* wings 2-3 times more curved than artificially inflated fly wings.

We have included these qualitative and quantitative analyses in the Supplementary information in a new section ‘Quantification of curvature during deployment’ and a new Supplementary movie 12 that showing wing deployment in a *Curly* mutant.

- p 7: evaporation : unclear ; more generally, the three last sentences of p. 7 (“While the flies... is alive”) deserve to be rewritten

A comprehensive physical mechanism for the *Curly* phenotype is still elusive and may be linked to abnormal pairing between the dorsal and ventral cuticle, [6]. In our artificial deployment experiment, the expanded wing is taken from a wild-type fly with normal pairing. A physical reason for curvature may be a post-mortem difference in the elastic properties of the dorsal and ventral plates. Such asymmetry would result in curvature under pneumatic actuation, with the softer plates stretching more than the stiffer one, a property widely used in soft robotics to obtain curvature [7].

Response Fig. 9: (a) Shapes of wild type, *Curly* and inflated wings during the deployment process. (b) Evolution of the average side-view curvature of the wing k as the wing deploys. The vertical dashed line corresponds to the profile in (a).

As this discussion is similar to the one on page 10, we have condensed the two paragraphs in the discussion section of the article rather than scattering them throughout the manuscript, and we have also added more details in the section ‘in vivo pressure recording and artificial pressure increase’ in the supplementary materials.

We have rewritten the corresponding paragraph in the discussion section:

We observe that the wings tend to curl up when deployed under artificial conditions. This probably indicates differential variations in the elastic properties of the ventral and dorsal plates. It may also be indicative of an active regulating mechanism that ensures in-plane deployment of the wings when the fly is alive. Although this curling is reminiscent of the phenotype observed in *Curly* mutant flies, we show that the two phenomena are quantitatively different by comparing the curvature during deployment (see Supplementary Fig. 2-4 and Supplementary Movie 12).

- p 9: is it "provides evidence" or only "strongly suggests"?

It strongly suggests. Thank you for the more correct wording. We have followed the reviewer’s suggestion.

- p 10: same as p 7, develop "Curly" and "evaporation"

As this discussion is similar to that on page 7, we have condensed the two paragraphs into the discussion section of the paper rather than scattering them throughout the manuscript. We thank the referee for the suggestion to compare with *Curly* phenotype.

- p 11: A male, a female: is it a question of sex? Or is it only a way to explain that $N=2$?

The reviewer is right that this is an indication that the experiments had been conducted on two samples. We

explicitly add that these were 2 flies (N=2). We specify the sex of the fly as this is an indication of the typical size of the fly and wing (females being larger than males). However, there is a wide size distribution among individuals of the same sex. Furthermore, we observed no systematic differences in wing structure, kinematics, dynamics or mechanical properties between males and females.

We now specify $N = 2$ in the method.

- p 13: *engineering stress, true stress: discuss better which one should be used. And discuss also which strain should be used (eg true / Hencky strain)*

The reviewer is right that true strain $\ln(\lambda)$ is the most appropriate definition in our case and should also be used, as it better reflects the formalism of large deformations, which corresponds to our tensile experiments and the theoretical framework developed for the hyperelastic model.

We now plot the true stress σ as a function of the true strain $\ln(\lambda)$ in Fig. 4c of the main manuscript.

- p 13: *Hertz contact. In general it applies to an elastic sphere and an elastic flat substrate. But in the reader's mind it is often associated with an elastic sphere on a much stiffer substrate. Here it seems the sphere is much stiffer than the substrate. Please clarify.*

We use Hertz's classical contact model to deduce the elastic properties of the expanded wings and relate the force-displacement curves to the intrinsic mechanical properties of the tissue. The reviewer is right to say that, in our case, the sphere is much stiffer than the sample. This is the classic configuration when measuring the mechanical properties of a sample by nanoindentation.

Taking into account the elasticity of the indenter and the geometry of the substrate, the indenting force F_i can be related to the indentation depth δ as:

$$F_i = \frac{4}{3} E_{eff} R_{eff}^{1/2} \delta^{3/2} \text{ where } \frac{1}{E_{eff}} = \frac{1 - \nu_s^2}{E_s} + \frac{1 - \nu_{ind}^2}{E_{ind}} \text{ and } \frac{1}{R_{eff}} = \frac{1}{R_s} + \frac{1}{R_{ind}} \quad (2)$$

where the indices $_s$ and $_{ind}$ represent the sample and indenter, and E_{eff} and R_{eff} are the effective reduced Young's modulus and radius respectively. When the indenter is much stiffer than the sample, and the sample is flat with Poisson's ratio $\nu = 0.5$, $E_{eff} \approx E_s / (1 - \nu_s^2)$ and $R_{eff} \approx R_{ind}$, such that:

$$F_i = \frac{4}{3} \frac{E_s}{1 - \nu^2} R_{ind}^{1/2} \delta^{3/2} \quad (3)$$

with E_s the sample Young's modulus and R_{ind} the radius of the indenter.

We now describe these assumptions in greater details in the methods and in Supplementary materials in a new section 'Nanoindentation of the cuticle of adult wings'.

- p 14: *after "uniaxial extension" add λ^p ; and check the last expression, I would have thought that $\lambda_z^p = \lambda^p$ and that $\lambda^p (\lambda_x^p)^2 = 1$.*

The reviewer is right! This is a typo that we have corrected. We thank the reviewer for pointing it out.

- p 16: *add a few words to discuss physically how eqs 13 (or 18) reflect the same physics as eqs 5-6.*

We build a model of the wing's deployment and first carry out a force balance on the system. Equations Eq. 5-6 are the results of this force balance. Then, to solve these equations and predict in-plane and vertical elongation as a function of applied pressure, we need to give the material a constitutive law, i.e. write explicit relationships between stress, σ and σ^p , and elongation, λ and λ^p . To this end, we first implement the simplest possible constitutive law, Hooke's linear elastic law, before considering a more realistic constitutive law, which is the Gent hyperelastic model. The force balance depicted in equations Eq. 5-6, combined with the linear elastic constitutive law (respectively Gent model), gives predictions for λ and λ^p in Eq. 12-13 (respectively Eq. 18-19).

The physics-based connection between Eq. 12-13 (and also Eq. 18-19) and equations Eq. 5-6 is highlighted in the new manuscript.

- p S3: *I can guess what 'thermally contracting' means but a few words of explanation would help. Moreover, please explain why it is the best method to produce wrinkles.*

Thermal expansion and biological growth are rigorously kinematically equivalent [8]. Using thermal expansion to numerically mimic tissue growth is a common numerical method for studying instabilities arising from differential growth in biological structures [9]. To obtain the initial state of the wrinkled cuticle bilayer, we define a coefficient of thermal expansion in the longitudinal direction x of the epithelial cell substrate with a thermal strain $\epsilon_{th} = \alpha_{xx}\Delta T$, with α the thermal expansion coefficient. When a finite temperature difference ΔT is applied, the substrate shrinks along the longitudinal direction, so that the film becomes comparatively too long and buckles.

The thermal expansion method used to produce wrinkles is now described in more detail in the Supplementary information.

Typos: p. 11 quantified; p 14 an increasing loading rates; p 17 non-dimensionnal; p S1 "as sketched Fig. 1B"

Thank you very much. We have corrected the typos.

References

- [1] Svend Olav Andersen. Insect cuticular sclerotization: a review. *Insect biochemistry and molecular biology*, 40(3):166–178, 2010.

- [2] Ryan Wagner, Barry R Pittendrigh, and Arvind Raman. Local elasticity and adhesion of nanostructures on *drosophila melanogaster* wing membrane studied using atomic force microscopy. *Applied surface science*, 259:225–230, 2012.
- [3] Stuart E Reynolds. Hormonal regulation of cuticle extensibility in newly emerged adult blowflies. *Journal of Insect Physiology*, 22(4):529–534, 1976.
- [4] Paul Grandgeorge, Natacha Krins, Aurélie Hourlier-Fargette, Christel Laberty-Robert, Sébastien Neukirch, and Arnaud Antkowiak. Capillarity-induced folds fuel extreme shape changes in thin wicked membranes. *Science*, 360(6386):296–299, 2018.
- [5] Dominic Vella. Buffering by buckling as a route for elastic deformation. *Nature Reviews Physics*, 1(7):425–436, 2019.
- [6] Thomas Ryan Hurd, Feng-Xia Liang, and Ruth Lehmann. Curly encodes dual oxidase, which acts with heme peroxidase curly su to shape the adult *drosophila* wing. *PLoS genetics*, 11(11):e1005625, 2015.
- [7] Filip Ilievski, Aaron D Mazzeo, Robert F Shepherd, Xin Chen, and George McClelland Whitesides. Soft robotics for chemists. *Angewandte Chemie International Edition*, 2011.
- [8] Gareth Wyn Jones and S Jonathan Chapman. Modeling growth in biological materials. *Siam review*, 54(1):52–118, 2012.
- [9] Thomas Lessinnes, Derek E Moulton, and Alain Goriely. Morphoelastic rods part ii: growing birods. *Journal of the Mechanics and Physics of Solids*, 100:147–196, 2017.

Wing ~~expansion~~deployment in *Drosophila melanogaster*

Simon Hadjaje¹, Ignacio Andrade-Silva^{1,2}, Marie-Julie Dalbe³,
Raphaël Clément^{4*}, Joel Marthelot^{1*}

¹Aix-Marseille University, CNRS, IUSTI & Turing Centre for Living Systems (CENTURI), Marseille, France

²Departamento de Física, Facultad de Ciencias Físicas y Matemáticas,
Universidad de Chile, Santiago, Chile

³Aix Marseille Univ, CNRS, Centrale Med, IRPHE, Marseille, France

⁴Aix-Marseille University, CNRS, IBDM & Turing Centre for Living Systems (CENTURI), Marseille, France

*To whom correspondence should be addressed joel.marthelot@univ-amu.fr; raphael.clement@univ-amu.fr

During their final transformation, insects emerge from the pupal case and deploy their wings within minutes. The wings deploy from a compact origami structure, to form a planar ~~rigid and functional~~and rigid blade that allows the insect to fly. ~~The deployment~~Deployment is powered by a rapid increase in internal pressure, and by the subsequent flow of hemolymph into the deployable wing structure. Using a combination of imaging techniques, we characterize the internal and external structure of the wing in *Drosophila melanogaster*, the unfolding kinematics at the organ scale, and the hemolymph flow during deployment. We find that, beyond the mere unfolding of the macroscopic folds, wing deployment also involves ~~an expansion of cell surface and the unfolding of microscopic wrinkles in~~wing expansion, with the stretching of epithelial cells and the unwrinkling of the cuticle enveloping the wing. A quantitative computational model, incorporating mechanical measurements of the viscoelastic properties and microstructure of the wing, predicts the existence of an operating point for deployment and captures the dynamics of ~~expansion~~the process. This model ~~suggests~~shows that insects exploit material and geometric nonlinearities to achieve rapid and efficient reconfiguration of soft deployable structures.

1 Main

~~Embryonic development is the stage of major morphological transformations of~~During development, tissues and organs undergo dramatic morphological transformations [1–5]. The biological and physical mechanisms ~~by which these transformations are coordinated~~that coordinate these transformations in space and time are a central ~~issue~~question in developmental biology. The ~~wing shaping~~ of the fruit fly wing *Drosophila melanogaster* is a paradigmatic example of such continuous organ-scale transformation, ~~combining expansion that combines~~growth and buckling of the wing epithelium to shape the larval and pupal wings into a highly folded origami

structure [6–9]. However, how the origami ~~unfolds eventually deploys~~ into a functional wing ~~within minutes after minutes after pupal~~ emergence remains largely unknown. Early ~~work has works have~~ suggested a mechanical basis for wing ~~expansion deployment~~ in insects, associated with an increase in hemolymph pressure in the open circulatory system ~~[10–12] [10–13]~~ under hormonal control ~~[14,15] [14,16]~~, but an integrative physical model of this ~~rapid~~ transformation is lacking. Here, we describe wing actuation as a fluid-structure interaction problem based on experimental characterization of mechanical properties and unfolding kinematics at ~~the~~ organ and tissue scales. We show that wing pressurization, combined with constitutive properties and tissue geometry, results in an operating point that the insect uses for robust ~~unfolding deployment~~. This work sheds light on a neglected but fascinating morphogenetic process at the organ scale. It will also inform engineering strategies for deployable structures whose shape evolves from compact and folded to extended and operational, with applications ranging from biomedical to aerospace [17], and for morphing artificial matter that continuously changes shapes [18], with applications ranging from programmable mechanical metamaterials [19–23] to soft robotics [24–27].

Figure 1: ~~Geometry and pressure in the unfolding wing.~~ ~~(a) Geometry and pressure in the deploying wing.~~ ~~(a)~~ Snapshots of the wings ~~expansion deployment~~ with one of the longitudinal vein highlighted in green. ~~(b)~~ (i) Microtomography of the folded wings. (ii) Micro-CT scan cross-section showing macroscopic folds, vein structure (white arrows) and internal pillars. (iii) Perpendicular section revealing the hexagonal pillars organization. The sketch summarizes the wing structure: two plates (thickness e) connected with pillars (height h , diameter d) organized in a hexagonal lattice (interpillar distance a). ~~(c)~~ Fluorescent beads (white dots, hemolymph markers) flowing in ~~an expanding a deploying~~ wing. ~~(d)~~ *in vivo* recording of the internal pressure $P(t)$ of a newly emerged insect. Wings ~~expansion deployment~~ takes place throughout the grayed-out segment at a constant pressure $P_{exp} P^*$.

Figure 2: **Wing elongation during expansion.** **(a) Unfolding and expansion.** (a) Veins network in folded state (from micro-CT scans) and **adult deployed** state (from bright field microscopy images). Each color represents a specific vein. **(b) Apparent Macroscopic wing stretch-deployment** $\Lambda = L/L_F$ as a function of time for 3 different flies. L is the length between the two ends of the longitudinal vein highlighted in green in (a) and Fig. 1a, and L_F the initial folded length. Dotted vertical lines correspond to the snapshots in Fig. 1a. **(c) Arclength of expanded deployed veins** l_E-l_D versus folded veins l_F . Each color represents a vein shown in (a). Length measurements from micro-CT are represented by square markers, while veins tracked on videos are represented by circular markers (empty for females, full for males). Solid line is a linear fit of all experiments. Standard deviation and relative uncertainties due to video tracking are indicated by error bars.

Hemolymph injection into a two-dimensional network causes wing expansion throughout the wing causes deployment

We first design a protocol to image wing deployment macroscopically. Upon eclosion, the fly is glued to a thin fiber mounted on a micromanipulator, and **expansion-deployment** is recorded using bright-field microscopy from dorsal and lateral views. Fig. 1a shows snapshots of the dorsal view of the wing **expansion-deployment** (see Supplementary Movie 1). The wings start out highly folded, with macroscopic folds along the longitudinal veins. The two wings **expand-deploy** simultaneously, dramatically changing shape from a compact three-dimensional structure to a, to a wing with a transient downward curvature during deployment to a fully deployed flat wing (Fig. 1a 1a and Supplementary Fig. 2). Additional characteristic features of deployment are revealed by lateral observation. The strong activity of the pharyngeal pump (see Supplementary Movie 2) indicates that the fly swallows air prior to **wing expansion deployment** [28] [29], resulting in inflation of the ptilinum, a **reversible** pouch located on the fly's head. The abdominal contractions that occur simultaneously contribute to an increase in internal pressure, which results in the injection of hemolymph into the wing, causing it to **expand deploy** [10]. Note that the air swallowed by the insect remains confined to the digestive tract, and that there is no air-liquid interface in the circulation of hemolymph or in the wing during deployment. At the end of **expansion deployment**, the ptilinum deflates and the abdomen relaxes, indicating a decrease in internal pressure. After **expansion deployment**, the final shape of the wing is stabilized by cuticle tanning and sclerotization [16] [15].

To gain insight into the microscopic structure of the wing, we used high-resolution X-ray micro-CT just

Figure 3: **Cellular-scale unwrinking.** **(a) Cells stretching and cuticle unwrinking.** **(a)** Schematic view of a wing microscopic structure: 2 monolayers of epithelial cells (green) covered by a thin rigid cuticle (red) and connected by pillars. **(b)** Cross section following the dotted frame in (a), transmitted electron microscopy (TEM). **(c)** Wing surface scans (optical profilometer, $50 \times 50 \mu\text{m}^2$). Top right inset: 3D reconstruction of the volcano-shaped wrinkle on the apical surface of an epithelial cell (spining disk confocal microscopy, Utrophin:GFP). **(d)** Epithelial cells contour (spining disk confocal microscopy, Ecad:GFP). For (b-d) top images correspond to folded wings, while bottom images correspond to expanded-deployed wings. **(e)** Evolution of the main geometric characteristics from folded (F) to expanded-deployed (ED) stages (from top left to bottom right diagram): 2D epithelial cell surface area; 3D surface area integration of the cuticle; total wing thickness; and wrinkles height. Data are obtained by TEM (purple boxes, see (b)), profilometer scans (yellow, (c)), spinning disk (blue, (d)) and biphoton images (gray, see Supplementary Fig. S5d in supplementary).

before deployment (Fig. 1b(i) and Supplementary Movie 3). A cross-section normal to the proximo-distal axis (Fig. 1b(ii)) reveals that the wing is composed of two continuous, folded plates of thickness $e = 6.5 \mu\text{m}$, connected by regularly spaced pillars of height $h = 7.5 \mu\text{m}$. Veins, hollow tubular structures indicated by white arrows on Fig. 1b(ii), are already visible in the folded wings. The cross-section normal to the dorso-ventral plane reveals that the pillars are arranged in a regular hexagonal lattice of period $a = 6.7 \mu\text{m}$ and diameter $d = 3.3 \mu\text{m}$ (Fig. 1b(iii)). The inside of the wing thus forms a two-dimensional porous network as sketched in Fig. 1b.

To understand how the rise in pressure guides the structure deployment, we first examine the hemolymph distribution in the wing. Prior to expansion deployment, fluorescent beads are injected into the abdomen of the fly. Fig. 1c shows that the beads permeate the entire wing surface during expansion deployment, indicating that hemolymph is injected throughout the entire structure in the two-dimensional porous network, rather than just confined to the veins as observed in adults (see [30] and Supplementary Movie 4), and commonly assumed during deployment [7, 31].

The internal pressure during expansion-deployment is measured by stinging the thoracic region between the two wings (the scutellum) of newly emerged flies, with a capillary connected to a pressure transducer (see Supplementary Movie 5). The pressure increases over approximately 15 minutes before reaching a plateau of $P_{exp} \approx 2.5 P^* \approx 2.5 \text{ kPa}$, as shown in Fig. 1d. The wings expand-deploy at this constant pressure. Following

~~expansion deployment~~, the pressure suddenly drops to a lower value of a few hundred pascals. This coincides with the rapid deflation of the ptilinum, that is highly inflated during ~~expansion deployment~~, and the relaxation of muscle contractions. The abdomen goes from an elongated and stretched state to a shorter and relaxed state. The abdominal and thoracic muscles involved in the increase in pressure degenerate shortly after deployment [32]. The fly thus transiently generates a few kilopascals to deploy its wings within a few minutes. Remarkably, this transformation bears similarity to that of artificial inflatable structures, such as air mattresses or paddle boards, in which pressure guides in-plane expansion of two plates connected by pillars [33].

~~The wing~~ Wings not only unfolds unfold but also stretchexpand

To quantify ~~expansion deployment~~, we use the natural landmarks provided by the wing veins. The three-dimensional position of the vein network is reconstructed using micro-CT images in the folded state (Fig. 1b(iii)). This network is preserved in the deployed wing, allowing for a direct comparison between the folded and ~~expanded-deployed~~ states (Fig. 2a and Supplementary Fig. S1). To characterize the kinematics, we first ~~measure the apparent wing stretch~~ quantify the deployment $\Lambda = L/L_F$, where L is the length of the segment joining the extremities of a longitudinal vein, and L_F the corresponding initial folded length as shown in the inset in Fig. 2b. ~~The apparent stretch~~ Λ is thus dominated by the kinematic opening of macroscopic folds. We find that ~~expansion dynamics are the deployment dynamics is~~ highly reproducible between individuals (see Supplementary Movie 6), with wing overall length increasing by a factor of 3 within 2 minutes (Fig. 2b).

Next, we investigate whether the deployment of the folded wing is solely due to the kinematic opening of macroscopic folds, similar to the unfolding of inextensible origami paper [19–21] or the wing of adult *Cocinellidae* [34], or whether it involves a combination of structural unfolding and tissue ~~stretching expansion~~. To measure changes in tissue metrics, we compare the arclength of veins in the folded and ~~expanded-deployed~~ states. In Fig. 2c, we plot the arclength of the ~~expanded vein~~ l_E -deployed vein l_D against the initial folded arclength l_F . To complement these measurements, we track deformation of three individual veins (marked in blue, yellow, and green in Fig. 2a) using bright-field microscopy observations of dorsal and lateral views (circular markers in Fig. 2c). Notably, we find that the wings not only unfold during ~~expansion deployment~~, but also ~~lengthen expand~~ by approximately 1.6 times their original length. Veins oriented in different directions present the same stretch, indicating that ~~extension expansion~~ is isotropic in the plane. Therefore, the rapid and dramatic ~~expansion deployment~~ of the wings appears to be a dual morphing response that involves unfolding macroscopic folds ~~and additional isotropic tissue stretching, and isotropic tissue expansion~~. Next, we address how this tissue ~~stretching expansion~~ is manifested at the cellular level.

~~Apparent stretching relies on unwrinkling at~~ Epithelial cells stretch while the cellular scalecuticle only unwrinkles

In order to gain a deeper understanding of the kinematics of ~~unfolding expansion~~ at the cellular level, we conduct observations of cross-sections of wings normal to the proximo-distal axis (dotted plane in the sketch in Fig. 3a)

using transmission electron microscopy (TEM) in both the folded and ~~unfolded-deployed~~ states. Micrographs (Fig. 3b) show that the folded wing is composed of two layers of epithelial cells (one ventral, the other dorsal, shown in green) separated by a lumen, into which hemolymph flows. The epithelial cell layer is covered by a thin, wrinkled layer of cuticle (shown in red). ~~During the process of expansion~~As the wings unfold and expand, the cuticle unwrinkles while maintaining a constant thickness of 200 nm in the adult wing. This suggests that the cuticle flattens without undergoing any significant stretching. The corresponding micrograph (Fig. 3b, bottom) is obtained a few hours after ~~the expansion process~~deployment, at which point the epithelial cells have already undergone apoptosis, detached, and been washed out of the wing by the action of the wing hearts [35,36], leaving only the cells surrounding the veins. The variation in wing thickness under pressure is measured using two-photon microscopy. We observe that ~~the pillars stretch under hemolymph pressure~~wing thickness is approximately doubled, increasing the space between the two cell layers during ~~expansion~~deployment (see wing thickness in Fig. 3e and Supplementary Movie 7).

Using optical profilometry, we measure the topography of the apical surface of the folded and ~~expanded~~deployed wing (solid line plane in the sketch in Fig. 3a). The wrinkles are organized in a hexagonal pattern, corresponding to the cellular tiling (see Fig. 3c). From cell centroids, we generate the Voronoi tessellation in order to compute the 2D surface area and maximum height of the wrinkles. Fig. 3e shows that the height of the wrinkles decreases while the 2D surface area increases during ~~expansion~~deployment, indicating a general flattening of the wing surface. This increase ~~of in~~cell surface area is confirmed by direct measurement in the folded state and during ~~expansion~~deployment using spinning disk microscopy (Fig. 3d). Note that the cell shape after ~~expansion~~deployment shows no significant shape anisotropy, ~~suggesting isotropic cell stretching~~again indicating isotropic tissue expansion. We label ~~the cells' apical surface~~apical surface of the cells, i.e. the surface where the cuticle adheres, to segment individual cells and reconstruct their shape in 3D, ~~as shown in the inset of Fig. 3e.~~In folded wings, ~~the~~apical surface of the cells is shaped like a volcano (see as shown in the inset of Fig. 3c (see also Supplementary Movie 8 and Supplementary Fig. 6). The integration of the 3D outer surface (see boxplot 3D area in Fig. 3e) ~~reveals~~confirms that the cuticle has sufficient surface area to unwrinkle without stretching. Moreover, the ~~volcano initial shape is isometric with respect~~volcano-shaped wrinkle is isometric to a plane, ~~and~~allows the cuticle to unwrinkle without stretching during wing expansion. In summary, tissue expansion that occurs during wing deployment causes stretching and flattening of epithelial cells, and unwrinkling of the cuticle covering cells' apical surface.

Coupling between constitutive properties and geometry reveals an operating point for deployment

The kinematic observation of the wing microstructure is summarized in Fig. 4a. The ventral and dorsal plates are each composed of a bilayer of epithelial cells covered by a thin, rigid, and wrinkled cuticle. During ~~deployment~~and subsequent expansion, the epithelial cells flatten ~~and~~their apical surface, thus expanding their area ~~apparent area increases,~~ while the rigid cuticle unwrinkles without stretching. The two plates are connected by regularly spaced pillars organized in a hexagonal lattice. ~~Large microtubule bundles~~Pillars containing microtubule bundles

Figure 4: **Mechanics of the wing expansion** (a) **Cross-sectional sketch** **Mechanics of wing deployment**. (a) Schematic of the **folded two-scale deployment mechanism**. Macroscopic scale (top): **wing unfolding and expanded expansion**. Microscopic scale (bottom): **cellular stretching and subsequent cuticle unwrinkling and pillars straightening**. The sketch shows a cross section of the **folded (left) and deployed (right) wing structure** with wrinkles and **microtubules-pillars** under pressure. (b) F/S_0 versus **apparent stretch** $\Lambda = L/L_F$ obtained from tensile test of the wing. (Inset) **true stress** $\sigma = \lambda F/S_0$ versus **stretch** λ (**linear elastic model in dotted line**) and a hyperelastic Gent model with $J_m = 20$ (**solid line**). (c) **True stress** $\sigma = \lambda F/S_0$ versus **stretch** λ (**linear elastic model in dotted line**) and a hyperelastic Gent model with $J_m = 20$ (**solid line**). (d) Prediction of in-plane stretch λ (black, left axis) and pillars vertical stretch λ^p (orange, right axis) as a function of the normalized applied pressure $\bar{P} = P/E$ for a hyperelastic Gent model-material (model: solid lines; FEM: dashed lines). Pressure measurements normalized by a Young's modulus of $E = 16$ kPa are shown as gray markers. (Inset) **Constitutive law for a linear elastic model (dotted line) and a hyperelastic Gent model with $J_m = 20$ (solid line)**.

and microfilaments, observed in the early pupa [37, 38] and just before eclosion [39, 40], extend from the apical junctions to the basal junctions of the cells [39, 40] and **straighten under pressure**, thereby limiting the separation of the two plates **under pressure**.

To gain insight into the mechanics of **deployment** **the deployment process**, we start out by conducting tensile tests on dissected folded wings. One wing extremity is fixed to a motorized linear stage, and a constant velocity of $10 \mu\text{m/s}$ is imposed while the other wing extremity is attached to a rigid fiber connected to a load sensor (see Supplementary Movie 9). The force F is normalized by the wing's initial cross-sectional surface area S_0 , which was obtained through micro-CT scanning. In Fig. 4b, we plot F/S_0 as a function of **the apparent wing stretch** $\Lambda = L/L_F$. Two distinct regimes are observed: (i) for small values of $\Lambda \leq 1.25$, the macroscopic folds open and the material undergoes negligible stretching. The stress remains close to zero, showing that almost no force is required to unfold the origami-like macroscopic folds. (ii) Once the main folds open, the wing **stretches and stresses increase also stretches, and stress increases**. To disentangle the **effect effects** of macroscopic unfolding and **tissue stretching, and** determine the actual stiffness of the material, we follow the arclength l of a longitudinal vein. In **the inset of Fig. 4bc**, we plot the true stress $\sigma = F\lambda/S_0$ **assuming under the assumption of** incompressibility as a function of **stretch** $\lambda = l/l_F$ **the true strain** $\ln(\lambda) = \ln(l/l_F)$ **to account for large deformations**. We extract the

Young’s modulus of the material $E \approx 59\text{--}147\text{ kPa}$ $E \approx 113\text{ kPa}$, which captures the linear relationship observed at low strain (interval mean value from $n = 6$ experiments). Stiffening is observed at large larger strains, which we interpret as the effect of unwrinkling of the rigid cuticle layer as the tissue expands (see Supplementary Fig. S8e); 8d replicating this stiffening in FEM). This phenomenology bears similarities to the stiffening observed in artificial systems, where length variations are buffered by buckling [41, 42].

To make sense of these observations, we construct a mechanical model of expansion that couples structural deformation to hemolymph pressure increase. Deformation involves in-plane stretching stretch of the plates λ and elongation vertical stretch of the pillars λ^p , as well as thinning of the plates and pillars due to volume conservation (see sketch in Fig. 4a). Stresses in the pillars and plates compensate for the increase in internal pressure P . To capture the large-strain stress stiffening due to the in-plane unwrinkling process and the straightening of the microtubules microtubule bundles and microfilaments along the pillars, we consider a phenomenological hyper-elastic Gent model (see solid line in Inset Fig. 4c) with two parameters, E the Young’s modulus and the stress stiffening captured by the limiting value J_m . We find that λ and λ^p are solutions of a differential non-linear system of equations [43], which we solve numerically (see Methods for computation details). Fig. 4e d presents the theoretical predictions for λ and λ^p as a function of the applied non-dimensional pressure $\bar{P} = P/E$ (black and orange line, respectively). The model is supported by FEM numerical simulations and predicts an operating point $\bar{P} \approx 0.18$, at which the wing undergoes considerable deformation while maintaining a low operating pressure. At this operating pressure, the predicted vertical stretch of the pillars $\lambda^p \sim 4$ is in good agreement with the measurement of the total wing thickness obtained by two-photon microscopy (Fig. 3e). By fitting our experimental measurements on the $\lambda - \bar{P}$ diagram, we find a Young’s modulus of 16 kPa (gray markers in Fig. 4e d), which is slightly smaller than the value obtained with the tensile test.

To further evaluate the fluid-structure model, we artificially inflate fixed flies flies sacrificed just after they emerge from their pupal cases (see Methods Supplementary Fig. 14 and Supplementary Movie 10). We observe a very nonlinear behavior in plane of in-plane deformation and pressure, consistent with our theoretical model. When the pressure is maintained at a plateau below 10 kPa, the wings do not expand. Expansion deploy. Deployment occurs at operating pressures between 10 and 16 kPa. At pressures above 17 kPa, the microtubules do not support pillar expansion pillars can no longer support the stress and eventually rupture, and we observe the formation blisters and balloon-like wings. With the Young’s modulus measured in the tensile test, we obtain an operating point of $P/E \sim 0.15$, which is consistent with the one predicted by the model (see Fig. 4e). While the flies are wild-type, the wings tend to curl up when deployed under artificial conditions similar to the phenotypes observed in Curly flies. This could indicate differential evaporation between the dorsal and ventral plates, leading to differential variation in the elastic properties of the plates. Alternatively, it may be indicative of an active control mechanism that regulates the in-plane expansion of the wings when the fly is alive. d).

Expansion-Deployment dynamics is governed by the viscoelastic properties of the tissue

We now seek to gain insight into the dynamics of expansion deployment. A first hypothesis would be that the observed timescale results from the fluid-structure interaction between the flow of viscous hemolymph and the

Figure 5: **Dynamics of the wing expansion** (a) **Dynamics of wing deployment.** (a) Force-displacement curves obtained from nano-indentation experiments on folded wings, from slow indentation ($\dot{F}_i = 10^{-1} \mu\text{N/s}$, blue) to faster ($\dot{F}_i = 10^1 \mu\text{N/s}$, red). Inset: Deformation $\epsilon = (\delta/R)^{1/2}$ measured over time for a creep experiment during which a constant force F_i is applied (gray line). Experimental data (in black) are fitted with the model (yellow line). (b) ϵ versus normalized time t/T : both experimental data (from (a), colored markers) and FEM numerical simulations of viscoelastic bilayer indentation (green dashed line) collapse on the model $\epsilon = (t/T)^{2/3}$ (black line). Inset: local strain in the indentation direction $\epsilon_{zz} - \epsilon_{zz}$ from FEM simulations. (c) In-plane **deformation stretch** λ as a function of t/τ . Experimental measurements (normalized with $\tau = 10$ s): gray markers; model: solid line; FEM: dashed line. Snapshots of FEM simulations at $t/\tau = 0, 5$ and 25 are shown (color bar: strain in one of the in-plane directions)

deformation of the flexible porous network. For a typical hemolymph viscosity $\eta_f \sim 1$ mPa.s [44, 45], the characteristic timescale $\eta_f L^2 / (E h e)$ based on the coupling between wing elasticity and fluid viscosity [46] is on the order of milliseconds. This is several orders of magnitude shorter than the observed deployment timescale, indicating the presence of an additional dissipation mechanism.

In order to ascertain the viscoelastic properties of the wing, we perform nanoindentation measurements on the folded wing. A force F_i is applied at different loading rates while the indentation depth δ is measured (Fig. 5a). The effective stiffness appears to increase with the loading rate, a feature typical of viscoelastic materials. This indicates the existence of another source of dissipation in the material, with a characteristic time $\tau = \eta / E_i$ where η is the viscosity of the material and E_i the effective elastic response in indentation, with contributions from the epithelial cells as well as the cuticle.

Creep experiments (inset of Fig. 5a), which consist of a loading phase at constant rate followed by a constant-force relaxation phase, are accurately represented by a three elements Maxwell-Jeffrey model comprising a dashpot in series with a dashpot in parallel with an elastic spring of stiffness E_i . The system exhibits two distinct timescales: a short timescale $\tau_1 = 1.9 \pm 0.3$ s and a long relaxation time, $\tau_2 = 37 \pm 12$ s. The fit provides an estimate for the effective elastic response of the bilayer in indentation $E_i \sim 1.6$ MPa. This effective stiffness value is in good agreement with finite elements method (FEM) simulations (inset of Fig. 5b), obtained by indenting a bilayer composed of a thin elastic film (cuticle with Young's modulus $E_f = 100$ MPa and thickness 200 nm) deposited on an elastic substrate (epithelial cells with Young's modulus $E = 100$ kPa).

In the nanoindentation experiment, the long relaxation time can be neglected at the scale of the experiment (see Methods), and the model is reduced to a simple Kelvin-Voigt model consisting of a dashpot in parallel to an elastic spring with a single characteristic time. Assuming Hertz contact, the strain of a viscoelastic Kelvin-Voigt material under indentation can be described by the equation $\tau\dot{\epsilon} + \epsilon = F_i/(E_i R^2 \epsilon^2)$, where $\epsilon = (\delta/R)^{1/2}$ and $R = 4.7 \mu\text{m}$ the radius of the indenter. For a constant loading rate \dot{F}_i , and in the limit of $t \ll \tau$, the strain can be approximated by $\epsilon \sim (t/T)^{2/3}$, where $T = (2R^2 E_i \tau / (3\dot{F}_i))^{1/2}$. The experimental data collapse on this master curve (Fig. 5b), with a single value, $E_i \tau$ being fitted to the curves. A similar procedure is applied to six samples, each subjected to a loading pattern of 21 loading and unloading cycles with increasing loading rates. This yields $E_i \tau \approx 10 - 47 \text{ MPa}\cdot\text{s}$ and $\tau \approx 6 - 29 \text{ s}$, in agreement with previous measurement of viscous time in epithelial cells [47]. FEM indentation of the bilayer, which accounts for the viscoelastic properties of the substrate with a viscous time scale $\tau = 10 \text{ s}$, as depicted by the dashed green curve in Fig. 5b and Supplementary Fig. 12, aligns with the experimental curves.

We extend our quasistatic model of wing deployment to the dynamic case by taking into account the viscoelasticity of the material. In this scenario, the hyperelastic material has an internal time scale τ , and is subjected to a pressure step \bar{P}^* at $t = 0$. The model for λ as a function of t/τ (solid line in Fig. 5c) is supported by FEM simulations (dashed line in Fig. 5c and Supplementary Movie 11). Snapshots of the FEM are shown at $t/\tau = 0, 5, \text{ and } 25$, illustrating how the system deforms both in-plane and vertically. The system ultimately reaches its equilibrium state within a characteristic time of $\approx 10t/\tau$. This prediction is compared with experimental measurements of the time evolution of λ obtained by bright-field microscopy. We plot λ as a function of time normalized by the measured internal timescale of the material obtained with the indentation experiment: $\tau = 10 \text{ s}$. The experimental data align with the theoretical prediction and the FEM without additional fitting parameters. The equilibrium state of the expansion-deployment and its dynamics are well captured by this-our model, in which a visco-hyperelastic material is subjected to an increase in pressure. This provides-evidence-that-the-expansion-strongly-suggests-that-deployment is solely due to an increase in pressure and that its dynamics are regulated by the internal timescale of the wing material itself.

Discussion

Wing expansion-deployment in insects results from the coupling between material properties, geometry and fluid loading. Using the fruit fly *Drosophila melanogaster* as a model system, we first describe the complex structure of the folded wing, which consists of two sheets of epithelial cells separated by a lumen and covered by a wrinkled rigid cuticle. The sheets are held together by a hexagonal lattice of pillars ~~containing microtubules~~. Expansion-Deployment occurs a few minutes after emergence, as hemolymph flows in the lumen under a pressure increase of a few kPa. During expansion-deployment, we observe macroscopic unfolding of the wing folds, along with tissue stretching. This-stretching-is-due-to-the-unwrinkling-and-flattening-of-expansion. This expansion is due to the volcano-like 3D microscopic structure of the apical surface of each cell flattening and stretching of epithelial cells, and to the unwrinkling of the cuticle that covers cells' volcano-shaped apical surface. We then characterize the mechanical properties of the wings. Taking into account the geometry and strain-stiffening-strain-stiffening

hyperelasticity due to cuticle unwrinkling ~~of microtubules and pillars~~ stretching, we develop an effective model that analytically predicts in-plane expansion around an operating value of P/E . Finally, we extend our analysis to the dynamics of ~~expansion~~deployment. We model the viscoelastic behavior of epithelial cells using nanoindentation experiments. The implementation of dissipation in the effective model captures the ~~expansion~~ dynamics and is in agreement with experimental observations and ~~finite element~~FEM calculations.

Our model is based on an effective description of the complexity of the biological response, and depends only on a finite set of parameters measured independently in the experiments: an elastic Young's modulus E , a strain stiffness parameter J_m and a viscous time τ of the material. Despite its simplicity, the model gives a remarkable account of the kinematics and dynamics observed *in vivo*. In particular, it predicts the existence of an operating pressure $P/E = 0.18$ at which most of the ~~expansion~~deployment occurs. The pressure measured *in vivo* is slightly lower than the pressure predicted by the model or observed when we artificially inflate sacrificed flies after their removal from their pupal case. This discrepancy may be due to the fact that, for the sake of simplicity, we model the horizontal plates (~~cuticle-covered cells~~) and vertical pillars (~~microtubules~~) with the same effective constitutive law. Another potential source of discrepancy may be that tensile and indentation tests are carried out on dissected folded wings. Evaporation or cuticle hardening could artificially stiffen the moduli compared to *in vivo* conditions. ~~Differential evaporation between dorsal and ventral plates could also explain the observation of the Curly phenotype when wings are artificially inflated in sacrificed flies.~~

We observe that the wings tend to curl up when deployed under artificial conditions. This probably indicates differential variations in the elastic properties of the ventral and dorsal plates. It may also be indicative of an active regulating mechanism that ensures in-plane deployment of the wings when the fly is alive. Although this curling is reminiscent of the phenotype observed in Curly mutant flies, we show that two phenomena are quantitatively different by comparing the curvature during deployment (see Supplementary Fig. 3-4 and Supplementary Movie 12).

~~We proposed~~ We proposed an unprecedented morphometric and mechanical characterization of ~~the wing expansion~~insect wing deployment. A physical model summarizes our results and shows that insects exploit the coupling between geometric and material nonlinearities to achieve ~~wing expansion~~efficient deployment around an operating point. This physical actuation is biologically triggered after eclosion by a hormone-controlled behaviour: the repeated intake of air and abdominal muscle contractions leading to internal pressure increase. This provides an interesting example of post-developmental morphogenesis. ~~The~~ the groundwork is laid during larval and pupal development when the wings form, grow, undergo eversion and finally fold in the pupal case. However, upon eclosion the wings are still folded, thus not yet functional. The massive and rapid shape change that occurs as a result of air ingestion and abdominal contractions is mandatory to make them operational, and can be delayed by the insect if conditions are unfavorable [29]. Wing deployment is therefore a peculiar morphogenetic process: on the one hand, it takes place after development, and does not require cellular growth or mechanical activity, and on the other it requires active behavior of the animal to occur.

Our work also paves the way for future studies aimed at understanding the pressure-driven expansion of deployable biological structures, and addressing the wide diversity of wing sizes and shapes in insects [48, 49]. In addition to its interest in fundamental biomechanics, improving our understanding of the mechanics of such

deployable structures will find applications in the fields of material morphing physics and the engineering of flexible structures.

Acknowledgments: We thank Yoël Forterre for discussions, Marin Lebreuilly for help with initial indentation experiments, Nicolas Brouilly and the electron microscopy facility of the IBDM, Perrine Chaurand and the MATRIX platform for the microtomography. This work was supported by Agence National de la Recherche (ANR Tremplin BioSoftAct), SH acknowledges funding from Fondation pour la Recherche Médicale (FRM : FDT202304016556). I.A.-S. acknowledges funding by the FONDECYT Postdoctoral Grant 3230753.

2 Methods

2.1 Fly imaging

Strains and stock. Wild type Oregon-R flies are used throughout this study, with the exception of spinning disk fluorescent images, for which Ecad:GFP and Utrophin:GFP are used. Flies are maintained on standard fly food at 18°C in a temperature-controlled chamber with 12h light–dark cycles.

Bright-field microscopy. To record the expansion-deployment of the wings, a fly is collected immediately after eclosion from its pupal case and placed on a CO₂ pad to anesthetize and manipulate it. A micro-manipulator is used to approach and glue a thin fiber (120 μm diameter) to the dorsal part of the thorax of the fly, ensuring that the wings are free to move. The fly is then raised above the pad and the expansion-deployment of the wings is recorded from both a top view using a binocular microscope (Leica MZ16) and a lateral view using a Leica Z16 APO lens connected to two cameras (Imaging Source 37DFKBUX264).

Hemolymph flow visualization. To visualize hemolymph flow through the wings during expansion-deployment, fluorescent beads are injected into the fly’s abdomen (500 nm pink beads, Drummond nanoject II). The fly is then attached to a thin stick and placed under a fluorescent binocular microscope (Nikon SMZ1000).

Profilometer characterization of the wrinkles. Wings are dissected and molded on uncured elastomer (Kerr Polyvinylsiloxane Impression Material type 3). The use of a mold provides a rigid sample after polymerization eliminating hairs that could mask surface features. Subsequently, the molds are scanned using a profilometer with an optical pen (micromesure 2, optical pen CL1-MG140, 1.3 μm lateral resolution, 48 nm axial resolution). The surface 3D coordinates are analyzed using ImageJ and Matlab. Any overall surface curvature is corrected, the Voronoi tessellation of the wrinkles is generated, and the hexagonal organization, typical height, distance between wrinkles, and surface area of individual cells in both folded and expanded-deployed stages are extracted (Fig. 3c, Supplementary Fig. 7 and yellow boxplot Fig. 3e for quantification).

Micro-CT. Micro-CT is performed at CEREGE (UM34, MATRIX platform). Flies (N=2) are chemically fixed (glutaraldehyde + phosphate-buffered saline + paraformaldehyde), dehydrated with ethanol and dried with supercritical CO₂ (critical point bypass) and placed in a kapton tube for imaging. A male is scanned over a 12-hour period, with the scan centered on the wings with a x20 objective (resolution of 0.8 μm/voxel). A female is scanned over a 27-hour period, with the scan centered on a specific region of interest on the right wing with a x40 objective (0.32 μm/voxel). ImageJ (version 2.14.0/1.54f) is employed for post-treatment, 3D image visualization (plugins

3D viewer and volume viewer), veins tracking, and geometrical characterization of the wings. The geometrical parameters (plates thickness e , pillars diameter d , pillars height h , and interpillar distance a) are measured by threshold, ultimate points and Voronoi tessellation ~~processes~~ on high-resolution scans (see Fig. 1b(ii-iii) and Supplementary Fig. S5).

Transmitted Electron Microscope. Transmitted electron microscopy (TEM, FEI Tecnai g2 200 kv) is conducted at IBDM electronic microscopy platform. Wings are dissected from newly emerged flies (i) and adult flies 4h after wings ~~expansion~~ deployment (ii). Wings are then fixed using cryofixation and cryosubstitution. The total wing thickness is ~~quantified~~ quantified before and after ~~expansion~~ deployment using ImageJ (see boxplot wing thickness in Fig. 3e for quantification).

Spinning disk confocal microscope. A spinning disk confocal microscope is used to obtain the epithelial cells contour and the 3D apical cell surface from folded and ~~expanded~~ deployed wings. The microscope is equipped with CSU-X1 spinning disk unit (Yokogawa) mounted on a Nikon Ti eclipse stand, 100x/1.49NA378 Nikon objective, EMCCD Andor iXon3 DU897, and MicroManager software. Folded wings are dissected from newly emerged flies, placed on a microscope slide, and covered with a drop of oil to avoid desiccation. A similar protocol is employed for ~~expanded~~ deployed wings, which must be dissected shortly after ~~wing expansion as~~ deployment. ~~Indeed,~~ epithelial cells undergo apoptosis and are washed out from the wing after ~~expansion.~~ ~~Consequently,~~ ~~deployment, such that~~ mature wings no longer exhibit fluorescence. The cell contours of Ecad:GFP flies are obtained with the Tissue Analyzer plugin in ImageJ, and quantification (cell surface area, cell-to-cell distance) is conducted in Matlab (MathWorks R2023b). The apical cell surface is imaged by performing a high resolution z-stack on Utrophin:GFP wings. Utrophin, an Actin-binding protein, tags the apical cell cortex. The scans are visualized and thresholded in ImageJ, and the 3D surface areas of individual cells are integrated in Matlab (see Supplementary Movie 8 and inset in Fig 3c).

Two-photon microscopy. A two-photon microscope, equipped with a Zeiss 510 NLO (Inverse - LSM), a femtosecond laser (Laser Mai Tai DeepSee HP) and a 40 x/1.2 C Apochromat objective, is used to measure the total wing thickness *in vivo*. The fly thorax is attached to a stick, and the insect is placed such that the wing plane is aligned with the optical path of the microscope. Wing ~~expansion~~ deployment is concomitant with body movement on the order of the size of the insect (\sim mm), which makes the measurement of the wing thickness ($\sim 10 \mu\text{m}$) during ~~the expansion process~~ deployment in a live fly somewhat challenging. Nevertheless, we ~~could~~ capture fast z-stacks of the wing section starting from the wing distal tip. The analysis is conducted using ImageJ (see Supplementary Movie 7 and Fig. 3e for quantification). We measure a total wing thickness (composed of the height of the pillars h and the thickness of the plates e) of $\approx 36 \mu\text{m}$ during ~~expansion~~ deployment i.e. a 2-fold increase ~~compare~~ compared to folded wings cross-section. This yields a stretch of the pillars in the vertical direction of $\lambda^p \approx 4.8$.

2.2 Mechanical measurements

Pressure. The internal pressure of the insect is quantified as the wings ~~expand~~ deploy using an in-house experimental setup, mounted on a vibration-isolated optical table. The pressure probe is composed of a pressure

sensor (Honeywell 24PCBFA6D) connected to a rigid microfluidic channel. A glass capillary (Clark capillary glass GC100-15, 1 mm outside diameter, 0.58 mm internal diameter, tip size 10-50 μ m) previously pulled (Sutter Instrument P97-4832) is filled with a low viscosity silicon (AS 4 Wacker-Chemie, 0.004 mPa.s) and secured to one end of the channel, while a motorized syringe at the other end allows for volume control of the system. The tip of the syringe is filled with deionized water to create an oil/water meniscus in the large section of the capillary. This prevents an oil/hemolymph interface at the capillary tip, which could result in an additional Laplace pressure. The position and the shape of the oil/water meniscus are tracked to correct for any additional pressure due to a possible flux (hydraulic resistance) or non-flat meniscus (Laplace pressure). Flies with folded wings are collected immediately following their emergence from the pupal case and placed on the CO₂ pad. A micromanipulator is utilized to puncture the insect scutellum with the capillary mounted on the pressure probe. The CO₂ pad is then removed, and the wings ~~expansion-deployment~~ is observed from both top and side views. A LabVIEW interface (NI LabVIEW 2017) is employed to synchronize the pressure measurements, volume control of the syringe, and the images captured by the two cameras. Reproducibility was tested for $N = 4$ different flies (see Supplementary Fig. 13).

Tensile tests. Tensile tests are conducted on folded fly wings to characterize their mechanical properties. Wings are dissected from newly emerged flies and glued at their proximal extremity to a motorized linear-stage (glue Loctite AA 352, linear stage PI VT-80). The distal extremity of the wing is glued to a load sensor (Magtrol MBB-02-0.05) previously calibrated with known loads. A water tank is placed in close proximity to the wing to prevent desiccation. Following the application of the glue, a period of several minutes is allowed for it to harden before the linear stage is moved (see Supplementary Movie 9). The entire setup is controlled via a custom Labview interface, which includes monitoring the experiments with two cameras (providing top and side views), controlling the linear stage, and recording the force from the load sensor. The setup is placed on a vibration-isolated optical table. The raw measured force, F , is normalized by the cross-sectional surface area S_0 of the folded wing obtained via micro-CT scan to extract the engineering stress F/S_0 (see ~~supplementary § Geometrical parameters measure from micro-CT scan~~ Supplementary Fig. 5 for details on obtaining S_0). Upon stretching, we assume incompressibility and the cross-sectional surface area of the wing to decrease as $S = S_0/\lambda$. Consequently, the true stress σ , can be expressed as $\sigma = F/S = \lambda F/S_0$. We use ImageJ to track one of the longitudinal vein (underlined in green Fig. 4b and Supplementary Fig. S8a) during stretching and compute the tissue stretch $\lambda = l/l_F$ where l is the vein arclength and l_F is the corresponding initial folded length. The macroscopic ~~stretch~~deployment, Λ , is defined as the ratio between the end-to-end distance of the vein, L , and its corresponding initial value, L_F . ~~See (see Fig. 4b for a graph-plot of F/S_0 versus the macroscopic stretch Λ . The inset).~~ Fig. 4c depicts the true stress, σ , as a function of ~~the true strain $\ln(\lambda)$~~ . A linear fit at small stretch yields the Young's modulus, E .

Nano-indentation. Nano-indentation experiments are conducted on folded ~~wings and adult wings~~ (Supplementary Fig. 9) using a nano-indenter (Hysitron TI Premier). ~~The Folded~~ wings are dissected from flies just after emergence from the pupal case. The wing is then carefully placed on a thin layer of viscous epoxy glue coated on a microscope slide. This procedure allows the wing to slightly embed in the epoxy while leaving the wing top surface in the open air. Once the epoxy has cured, providing a rigid substrate for the wing, the microscope slide is secured in the nano-indenter with magnets. Two types of tests are conducted: (i) loading cycles with increasing

loading rates (maximum force $5\mu\text{N}$, rates $\dot{F}_i = 10^{-1} - 10^1 \mu\text{N/s}$) and (ii) creep tests with constant force (3 experiments at $F_i=3.2, 6$ and $8.1 \mu\text{N}$). All experiments are performed within a maximum of 30 minutes after wing dissection. During this period, no time-dependent effects on the wing mechanical properties are observed. The raw force, F_i , and indentation depth, δ , are analyzed using Matlab.

An indentation model of a viscoelastic material is constructed in order to extract the effective elasticity in indentation, E_i , and the viscosity, η , of the wing. The loading cycles tests are well captured by a simple Kelvin-Voigt material undergoing Hertz-like indentation. The total stress of a Kelvin-Voigt material is the combination of a linear elastic response and a viscous dissipation, such that $\sigma_i = E_i\epsilon + \eta\dot{\epsilon}$. This model is then coupled with Hertz theory of contact (see Eq. 1 in Supplementary material for prediction of elastic Hertz contact), $\sigma_i = F_i/(R\delta)$ and $\epsilon = (\delta/R)^{1/2}$. The strain of a viscoelastic Kelvin-Voigt material under indentation is thus the solution of the equation:

$$\tau\dot{\epsilon} + \epsilon = \frac{F_i}{E_i R^2 \epsilon^2}, \quad (1)$$

with $\tau = \eta/E_i$. Indentation cycles are performed at constant loading rate, so that $F_i = \dot{F}_i t$, with \dot{F}_i increasing with each cycle. In this case, strain can be solved from Eq. 1:

$$\epsilon = \left[\frac{\dot{F}_i \tau}{3R^2 E_i} \left(3\frac{t}{\tau} - 1 + e^{-\frac{3t}{\tau}} \right) \right]^{1/3} \underset{t \ll \tau}{\sim} \left(\frac{3\dot{F}_i}{2R^2 E_i \tau} \right)^{1/3} t^{2/3} = \left(\frac{t}{T} \right)^{2/3} \quad (2)$$

where $T = (2R^2 E_i \tau / (3\dot{F}_i))^{1/2}$. The experimental data are observed to collapse to a single line with a slope of 2/3 in a log-log scale in Fig. 5b. The data are fitted to the model in this limit to find $\eta = E_i \tau \approx 10 - 47 \text{ MPa.s}$. This range corresponds to the dispersion obtained by applying this procedure on six different samples, each subjected to a loading pattern of 21 loading and unloading cycles with **an**-increasing loading rates.

The experimental creep test comprises two distinct phases: a loading phase during which the force is increased at a constant rate, and a subsequent phase in which the force remains constant (see the force curve in gray in the Inset of Fig. 5a). The resulting strain ϵ relaxed slowly over time, a behavior which is well captured by a three elements Maxwell-Jeffrey model comprising a dashpot η_2 in series to a Kelvin-Voigt element (a dashpot η_1 in parallel with a spring E_i).

When coupled with Hertz contact, we obtain:

$$\tau_1 \tau_2 \ddot{\epsilon} + \tau_2 \dot{\epsilon} = \frac{F_i}{E_i R^2} \frac{1}{\epsilon^2} + \frac{\tau_1 + \tau_2}{E_i R^2} \left(\frac{\dot{F}_i}{\epsilon^2} - \frac{2F_i \dot{\epsilon}}{\epsilon^3} \right) \quad (3)$$

where $\tau_1 = \eta_1/E_i$ and $\tau_2 = \eta_2/E_i$. Eq. 3 is solved numerically, and the creep experimental data are fitted with E_i , τ_1 and τ_2 as fitting parameters (see Inset Fig. 5a, yellow line, for the fit).

Note that in the nano-indentation experiments, the solutions of the three-element Maxwell-Jeffrey model reduce to the simpler Kelvin-Voigt model described above. A loading of the form $F_i = \dot{F}_i t$ is imposed and we look for solutions of the form $\epsilon \approx (t/T)^\alpha$. For $\alpha = 2/3$, there are two groups of terms that evolve as $\sim t^{-1/3}$ and $\sim t^{-4/3}$. For short times, the dynamics is governed by terms in power of $-4/3$. Balancing these terms, we find the expression $\epsilon(t) \approx (t/T_0)^{2/3}$, with

$$T_0 = \left(\frac{2R^2 E_i \tau_{eff}}{3\dot{F}_i} \right)^{1/2}, \quad (4)$$

where $\tau_{eff} = \tau_1\tau_2/(\tau_1 + \tau_2)$ which tends toward τ_1 for $\tau_1 \ll \tau_2$. Numerical simulations of Eq. 3 for different values of \dot{F}_i confirm the above scaling.

2.3 Model

Wing geometry and force balance. We consider two infinite plates of thickness e , connected with pillars of diameter d , height h , organized in a hexagonal lattice of pitch a . We assume that both plates undergo spatially homogeneous equibiaxial extension in \mathbf{e}_x and \mathbf{e}_y , so that the principal stretches in the plates read $\lambda_x = \lambda_y = \lambda$. We assume that the material is incompressible, which implies that $\lambda_x\lambda_y\lambda_z = 1$ and $\lambda_z = 1/\lambda^2$. For the pillars, we consider a uniaxial extension in the \mathbf{e}_z direction. Incompressibility and isotropy lead to the following equation $\lambda_x^p\lambda_y^p\lambda_z^p = (\lambda_x^p)^2\lambda_z^p = 1$, superscript p refers to the pillars and we denote $\lambda_z^p = \lambda^p (\lambda_x^p)^2 = 1/\lambda^p$.

A force balance in the plates relates the Cauchy stress in the principal directions $\sigma_{XX} = \sigma_{YY} = \sigma_{zz}$ to the internal pressure P [43]. As the plates become thinner by a factor of $1/\lambda^2$ and the pillar height increases by λ^p , we obtain the equation $2\sigma e/\lambda^2 = Ph\lambda^p$ and:

$$\sigma = P \frac{h}{2e} \lambda^p \lambda^2 = P \frac{\Psi}{1 - \Psi} \lambda^p \lambda^2 \quad (5)$$

with $\Psi = h/(h + 2e)$ the relative height of the pillars in the non-deformed geometry.

Similarly for the pillars, the longitudinal Cauchy stress $\sigma_z^p = \sigma^p$ compensates for the increase in internal pressure, so that $\sigma^p A^p/\lambda^p = P (A\lambda^2 - A^p/\lambda^p)$ where A^p is area of the pillars in the plane and A is the total area of the plates in the plane. We introduce the density of the pillars in the plane $\Phi = A^p/A = \pi/(2\sqrt{3})(d/a)^2$, such that:

$$\sigma^p = P \left(\frac{2\sqrt{3}}{\pi} \left(\frac{a}{d} \right)^2 \lambda^p \lambda^2 - 1 \right) = P \left(\frac{\lambda^p \lambda^2}{\Phi} - 1 \right) \quad (6)$$

In order to solve these equations and obtain the values of (λ, λ^p) as a function of P , it is necessary to define the material's constitutive law.

Linear elastic model. In this section, we assume that the structure is composed of an isotropic homogeneous linear elastic material with Young's modulus E and Poisson ratio ν . For the stress in the plane of the plates, the theory of linear elasticity yields:

$$\sigma = \frac{E}{(1 + \nu)(1 - 2\nu)} [(1 - \nu)\epsilon + \nu(\epsilon + \epsilon_z)] \quad (7)$$

with the in-plane strain $\epsilon = \lambda - 1$ and in the normal direction $\epsilon_z = \lambda_z - 1$. We assume that the plates are under plane stress $\sigma_z = 0$, the thickness e being small compared to the other dimensions. We obtain $(1 - \nu)\epsilon_z + 2\nu\epsilon = 0$, which gives $\epsilon_z = -2\nu\epsilon/(1 - \nu)$, which we replace in Eq. 7 to obtain:

$$\sigma = \frac{E}{1 - \nu} \epsilon = \frac{E}{1 - \nu} (\lambda - 1) \quad (8)$$

Similarly for the pillars, we write the general expression for the stress in the principal direction:

$$\sigma^p = \frac{E}{(1 + \nu)(1 - 2\nu)} [(1 - \nu)\epsilon^p + \nu(\epsilon_x^p + \epsilon_y^p)] \quad (9)$$

We explicit $\epsilon_x^p = \epsilon_y^p$ by writing the stress in one of the horizontal direction:

$$\sigma_x^p = -P = \frac{E}{(1+\nu)(1-2\nu)} [(1-\nu)\epsilon_x^p + \nu(\epsilon_y^p + \epsilon^p)] \quad (10)$$

Indeed, the pillars are compressed by the internal pressure. We find $\epsilon_x^p = -(1+\nu)(1-2\nu)P/E - \nu\epsilon^p$, that we inject in Eq. 9 to get:

$$\sigma^p = E\epsilon^p - 2\nu P = E(\lambda^p - 1) - 2\nu P \quad (11)$$

~~With-~~

We now couple this elastic linear constitutive law for plates and pillars in Eq. 8-11 with the the force balance in the in-plane and vertical directions expressed in Eq. 5-6, we. We then obtain the following system of coupled non-linear equations:

$$\begin{cases} \frac{E}{1-\nu}(\lambda-1) = P \frac{h}{2e} \lambda^2 \lambda^p & (12a) \\ E(\lambda^p - 1) - 2\nu P = P \left(\frac{2\sqrt{3}}{\pi} \left(\frac{a}{d} \right)^2 \lambda^2 \lambda^p - 1 \right) & (12b) \end{cases}$$

or equivalently with the non-dimensional pressure $\bar{P} = P/E$:

$$\begin{cases} \frac{\lambda-1}{1-\nu} = \bar{P} \frac{\Psi}{1-\Psi} \lambda^2 \lambda^p & (13a) \\ \lambda^p - 1 - 2\nu \bar{P} = \bar{P} \left(\frac{\lambda^2 \lambda^p}{\Phi} - 1 \right) & (13b) \end{cases}$$

Eq. 13b gives $\lambda^p = (1 + \bar{P}(2\nu - 1)) / (1 - \bar{P}\lambda^2/\Phi)$ when $\bar{P}\lambda^2/\Phi \neq 1$. This result is then inserted-plugged into Eq. 13a to obtain the following equation for λ :

$$\lambda^3 + \left\{ (1-\nu) \frac{\Phi\Psi}{1-\Psi} [1 + \bar{P}(2\nu - 1)] - 1 \right\} \lambda^2 - \frac{\Phi}{\bar{P}} \lambda + \frac{\Phi}{\bar{P}} = 0 \quad (14)$$

This linear elastic model predicts a finite pressure at which the stretch, both in the horizontal and vertical direction, diverges. This would be detrimental for the wing, which requires a large but finite expansion at a given pressure. Furthermore, this simple model does not take into account the microstructure of the wing: (i) the epithelial cells are covered by a rigid cuticle initially wrinkled that yields to a strain stiffening as the tissue expands; and (ii) the two bilayers are connected with microtubules that uncoil as the plates move apart until the stress in the pillars diverges when these filaments get straight. The aforementioned features, as illustrated Fig. 4a, prompt us to adopt a more realistic model that considers large strains and nonlinearities (strain stiffening) in the material.

Hyperelastic Gent model. We use the phenomenological Gent hyperelastic model to describe the strain stiffening. This constitutive model is characterized by two parameters: the shear modulus $\mu = E/(2(1+\nu))$ and a limiting value J_m of the left Cauchy-Green deformation tensor first invariant. The second Piola-Kirchhoff stress tensor S depends on the strain energy density W :

$$\mathbf{S}_G = 2 \frac{\partial W}{\partial I_1} \mathbf{I} - p \mathbf{J} \mathbf{C}^{-1}; \text{ with } W = -\frac{\mu J_m}{2} \ln \left(1 - \frac{I_1 - 3}{J_m} \right) \quad (15)$$

where $\mathbf{C} = \mathbf{F}^T \mathbf{F}$ is the right Cauchy-Green deformation tensor, \mathbf{F} is the deformation gradient tensor, $I_1 = \text{tr}(\mathbf{C})$, $J = \det(\mathbf{F})$, \mathbf{I} is the identity and p is a reactive pressure due to incompressibility constraint.

The plates are under plane stress $S_{ZZ} = 0$, $S_{zz} = 0$ yielding the in-plane stress:

$$S = \left(1 - \frac{1}{\lambda^6}\right) \frac{\mu J_m}{J_m - 2\lambda^2 - \frac{1}{\lambda^4} + 3} \quad (16)$$

The pillars are compressed in the horizontal direction by internal pressure, $S_{XX}^p = S_{YY}^p = -\lambda P$, $S_{xx}^p = S_{yy}^p = -\lambda P$, so that:

$$S_{zz}^p = -\frac{P}{(\lambda^p)^2} + \frac{\mu J_m}{J_m - (\lambda^p)^2 - \frac{2}{\lambda^p} + 3} \left(1 - \frac{1}{(\lambda^p)^3}\right) \quad (17)$$

The Cauchy stress relates to the second Piola-Kirchhoff by $\sigma_G = J^{-1} \mathbf{F} \mathbf{S}_G \mathbf{F}^T$ such that with We now couple the hyperelastic constitutive law with the force balance in the in-plane and vertical directions expressed in Eq. 5-6, we obtain. We then obtain the following system of coupled non-linear equations [43]:

$$\left\{ \left(\lambda^2 - \frac{1}{\lambda^4}\right) \frac{\mu J_m}{J_m - 2\lambda^2 - \frac{1}{\lambda^4} + 3} = P \frac{h}{2e} \lambda^2 \lambda^p \right. \quad (18a)$$

$$\left. \left\{ \left((\lambda^p)^2 - \frac{1}{\lambda^p}\right) \frac{\mu J_m}{J_m - (\lambda^p)^2 - \frac{2}{\lambda^p} + 3} = P \frac{2\sqrt{3}}{\pi} \left(\frac{a}{d}\right)^2 \lambda^2 \lambda^p \right. \right. \quad (18b)$$

or in non-dimensionnal non-dimensional form with $\bar{P} = P/E = P/(2\mu(1+\nu))$:

$$\left\{ \left(\lambda^2 - \frac{1}{\lambda^4}\right) \frac{J_m}{2(1+\nu)(J_m - 2\lambda^2 - \frac{1}{\lambda^4} + 3)} = \bar{P} \frac{\Psi}{1-\Psi} \lambda^2 \lambda^p \right. \quad (19a)$$

$$\left. \left\{ \left((\lambda^p)^2 - \frac{1}{\lambda^p}\right) \frac{J_m}{2(1+\nu)(J_m - (\lambda^p)^2 - \frac{2}{\lambda^p} + 3)} = \bar{P} \frac{\lambda^2 \lambda^p}{\Phi} \right. \right. \quad (19b)$$

The deformations in the plane in-plane stretch λ and in the pillars λ^p are predicted numerically by solving the coupled non linear equations as a function of the applied normalized pressure $\bar{P} = P/E$ (see Fig. 4c, and Supplementary Fig. S10 for a parametric study of the model).

Dynamical model. We take into account the viscous dissipation measured experimentally by extending the Kelvin-Voigt model to a hyperelastic material. The second Piola-Kirchhoff stress tensor in the viscous branch is $\mathbf{S}_V = \tau \dot{\mathbf{S}}_G$ such that the total stress is $\mathbf{S} = \mathbf{S}_G + \tau \dot{\mathbf{S}}_G$. This yields $\sigma = J^{-1} \mathbf{F} \mathbf{S} \mathbf{F}^T$ and, replacing the Cauchy stress in Eq. 5-6, we obtain:

$$\left\{ \frac{J_m}{J_m - J_1} \left\{ \lambda^2 - \frac{1}{\lambda^4} + 2\tau \dot{\lambda} \left[\frac{3}{\lambda^5} + \left(\lambda^2 - \frac{1}{\lambda^4}\right) \frac{2\lambda + \frac{2}{\lambda^5}}{J_m - J_1} \right] \right\} = 2(1+\nu) \bar{P} \frac{\Psi \lambda^2 \lambda^p}{1-\Psi} \right. \quad (20a)$$

$$\left. \left\{ \frac{J_m}{J_m - J_1^p} \left\{ (\lambda^p)^2 - \frac{1}{\lambda^p} + 2\tau \dot{\lambda}^p \left[\frac{3}{2(\lambda^p)^2} + \left((\lambda^p)^2 - \frac{1}{\lambda^p}\right) \frac{\lambda^p - \frac{1}{(\lambda^p)^2}}{J_m - J_1^p} \right] \right\} + \frac{2\tau \bar{P} \dot{\lambda}^p}{\lambda^p} = 2(1+\nu) \bar{P} \frac{\lambda^2 \lambda^p}{\Phi} \right. \right. \quad (20b)$$

where $J_1 = I_1 - 3 = 2\lambda^2 + 1/\lambda^4 - 3$ and $J_1^p = I_1^p - 3 = (\lambda^p)^2 + 2/\lambda^p - 3$. The expansion dynamics is predicted numerically by solving Eq. 20a-20b. In particular, Fig. 5c shows the in-plane deformation-stretch λ as a function of the normalized time t/τ for a constant applied pressure $\bar{P} = 0.18$. The experimental data (gray markers), for which the time is normalized by the measured viscous dissipation time $\tau = 10$ s, exhibit a good match with the model, without fitting parameters.

2.4 Finite Element Method

We perform finite element method (FEM) simulations using the commercial software COMSOL Multiphysics (version 6.1) to (i) build a fundamental understanding of indentation experiments and (ii) support our model of wings expansion.

Indentation. We quantitatively study the effect of a bilayer on the indentation measurement by performing 2D axi-symmetric FEM simulations of a rigid spherical indenter (radius $R = 4.7\mu\text{m}$, $E_{ind} = 110\text{ GPa}$, $\nu_{ind} = 0.3$) applying a force to a bilayer composed of a soft cellular-like substrate (thickness $200\mu\text{m}$, width $400\mu\text{m}$) covered by a more rigid cuticle-like film 200 nm -thick. The substrate is modeled by a viscoelastic Kelvin-Voigt material with Young's modulus $E = 100\text{ kPa}$ and viscous time $\tau = 10\text{ s}$, while the film is purely elastic with Young's modulus $E_f = 100\text{ MPa}$. The boundaries are fixed at the bottom and the side of the system while loading-unloading cycles are applied on the bilayer by imposing the force with increasing loading rates (see Supplementary Fig. 12e-f).

Expansion. We numerically test our wing expansion model by conducting 3D FEM simulations of the pressure-driven deformation of a structure exhibiting the geometry of the wing: two plates with a thickness of $e = 6.5\mu\text{m}$ connected with pillars of height $h = 7.5\mu\text{m}$, diameter $d = 3.3\mu\text{m}$, and an interpillar distance $a = 6.2\mu\text{m}$ in a square lattice. a is chosen so that the in-plane pillars density in the FEM corresponds to that measured on micro-CT scans (see Fig. 1b). The system is modeled using an isotropic and incompressible hyperelastic Gent material (Young's modulus $E = 100\text{ kPa}$, limiting value $J_m = 20$) and solved for 1/8th of the geometry with symmetric boundary conditions. We first perform a stationary study, in which we impose an increasing pressure from $P = 0$ to 20 kPa and measure the in-plane deformation, stretch λ , and vertical deformation stretch, λ^p (see Fig. 4c black and orange dashed lines and Supplementary Fig. 11). We then turn to a time-dependent study by adding a viscous dissipation (Kelvin-Voigt viscous time $\tau = 10\text{ s}$) in the material. We impose a constant pressure $P/E = 0.18$ at $t = 0$ and measure λ and λ^p over time. Snapshots of 1/2 of the geometry are shown at different times t/τ in Fig. 5c (see also Supplementary Movie 11).

References

- [1] Nelson, C. M. On buckling morphogenesis. *J. Biomech. Eng.* **138**, 021005 (2016).
- [2] Shyer, A. E. *et al.* Villification: how the gut gets its villi. *Science* **342**, 212–218 (2013).
- [3] Tallinen, T. *et al.* On the growth and form of cortical convolutions. *Nat. Phys* **12**, 588–593 (2016).
- [4] Kim, S., Pochitaloff, M., Stooke-Vaughan, G. A. & Campàs, O. Embryonic tissues as active foams. *Nat. Phys* **17**, 859–866 (2021).
- [5] Mitchell, N. P. *et al.* Visceral organ morphogenesis via calcium-patterned muscle constrictions. *Elife* **11**, e77355 (2022).
- [6] Etournay, R. *et al.* Interplay of cell dynamics and epithelial tension during morphogenesis of the drosophila pupal wing. *Elife* **4**, e07090 (2015).

- [7] de la Loza, M. D. & Thompson, B. Forces shaping the drosophila wing. Mech. Dev. **144**, 23–32 (2017).
- [8] Harmansa, S., Erlich, A., Eloy, C., Zurlo, G. & Lecuit, T. Growth anisotropy of the extracellular matrix shapes a developing organ. Nat. Commun **14**, 1220 (2023).
- [9] Tsuboi, A., Fujimoto, K. & Kondo, T. Spatiotemporal remodeling of extracellular matrix orients epithelial sheet folding. Sci. Adv. **9**, eadh2154 (2023).
- [10] Eidmann, H. Untersuchungen über wachstum und häutung der insekten. Z. Morph. Ökol. Tiere **2**, 567–610 (1924).
- [11] Cottrell, C. The imaginal ecdysis of blowflies. observations on the hydrostatic mechanisms involved in digging and expansion. J. Exp. Biol. **39**, 431–448 (1962).
- [12] Moreau, R. Variations de la pression interne au cours de l'émergence et de l'expansion des ailes chez bombyx mori et pieris brassicae. J. Insect Physiol. **20**, 1475–1480 (1974).
- [13] Reynolds, S. E. Hormonal regulation of cuticle extensibility in newly emerged adult blowflies. Journal of Insect Physiology **22**, 529–534 (1976).
- [14] Dewey, E. M. et al. Identification of the gene encoding bursicon, an insect neuropeptide responsible for cuticle sclerotization and wing spreading. Curr. Biol. **14**, 1208–1213 (2004).
- [15] Honegger, H.-W., Dewey, E. M. & Ewer, J. Bursicon, the tanning hormone of insects: recent advances following the discovery of its molecular identity. J. Comp. Physiol. A **194**, 989–1005 (2008).
- [16] White, B. H. & Ewer, J. Neural and hormonal control of postecdysial behaviors in insects. Annu. Rev. Entomol. **59**, 363–381 (2014).
- [17] Pellegrino, S. Deployable structures in engineering. In Deployable structures, 1–35 (Springer, 2001).
- [18] Klein, Y., Efrati, E. & Sharon, E. Shaping of elastic sheets by prescription of non-euclidean metrics. Science **315**, 1116–1120 (2007).
- [19] Silverberg, J. L. et al. Using origami design principles to fold reprogrammable mechanical metamaterials. Science **345**, 647–650 (2014).
- [20] Filipov, E. T., Tachi, T. & Paulino, G. H. Origami tubes assembled into stiff, yet reconfigurable structures and metamaterials. Proc. Natl. Acad. Sci. U.S.A. **112**, 12321–12326 (2015).
- [21] Dudte, L. H., Vouga, E., Tachi, T. & Mahadevan, L. Programming curvature using origami tessellations. Nat. Mater. **15**, 583–588 (2016).
- [22] Faber, J. A., Arrieta, A. F. & Studart, A. R. Bioinspired spring origami. Science **359**, 1386–1391 (2018).

- [23] Melancon, D., Gorissen, B., García-Mora, C. J., Hoberman, C. & Bertoldi, K. Multistable inflatable origami structures at the metre scale. *Nature* **592**, 545–550 (2021).
- [24] Siéfert, E., Reyssat, E., Bico, J. & Roman, B. Bio-inspired pneumatic shape-morphing elastomers. *Nat. Mater.* **18**, 24–28 (2019).
- [25] Kim, W. *et al.* Bioinspired dual-morphing stretchable origami. *Sci. Robot.* **4**, eaay3493 (2019).
- [26] Jones, T. J., Jambon-Puillet, E., Marthelot, J. & Brun, P.-T. Bubble casting soft robotics. *Nature* **599**, 229–233 (2021).
- [27] Jones, T. J., Dupuis, T., Jambon-Puillet, E., Marthelot, J. & Brun, P.-T. Soft deployable structures via core-shell inflatables. *Phys. Rev. Lett.* **130**, 128201 (2023).
- [28] Peabody, N. C. *et al.* Bursicon functions within the drosophila CNS to modulate wing expansion behavior, hormone secretion, and cell death. *J. Neurosci.* **28**, 14379–14391 (2008).
- [29] Peabody, N. C. *et al.* Characterization of the decision network for wing expansion in drosophila using targeted expression of the *trpm8* channel. *J. Neurosci.* **29**, 3343–3353 (2009).
- [30] Pass, G. Beyond aerodynamics: The critical roles of the circulatory and tracheal systems in maintaining insect wing functionality. *Arthropod Struct. Dev.* **47**, 391–407 (2018).
- [31] Ewer, J. & Reynolds, S. Neuropeptide control of molting in insects. In *Hormones, brain and behavior*, 1–XVI (Elsevier, 2002).
- [32] Denlinger, D. L. & Zdárek, J. Metamorphosis behavior of flies. *Annu. Rev. Entomol.* **39**, 243–266 (1994).
- [33] Barker, E. D. Inflatable mattress. *United States Patent* (1951).
- [34] Saito, K., Nomura, S., Yamamoto, S., Niiyama, R. & Okabe, Y. Investigation of hindwing folding in ladybird beetles by artificial elytron transplantation and microcomputed tomography. *Proc. Natl. Acad. Sci. U.S.A.* **114**, 5624–5628 (2017).
- [35] Kiger Jr, J. A. *et al.* Tissue remodeling during maturation of the drosophila wing. *Dev. Biol.* **301**, 178–191 (2007).
- [36] Tögel, M., Pass, G. & Paululat, A. The drosophila wing hearts originate from pericardial cells and are essential for wing maturation. *Dev. Biol.* **318**, 29–37 (2008).
- [37] Sun, T. *et al.* Atypical laminin spots and pull-generated microtubule-actin projections mediate drosophila wing adhesion. *Cell Reports* **36** (2021).
- [38] Tran, N. V. *et al.* Programmed disassembly of a microtubule-based membrane protrusion network coordinates 3D epithelial morphogenesis in drosophila. *The EMBO Journal* **43**, 568–594 (2024).

- [39] Johnson, S. A. & Milner, M. J. The final stages of wing development in *Drosophila melanogaster*. Tissue Cell **19**, 505–513 (1987).
- [40] Bökel, C., Prokop, A. & Brown, N. H. Papillote and piopio: *Drosophila* zp-domain proteins required for cell adhesion to the apical extracellular matrix and microtubule organization. J. Cell Sci. **118**, 633–642 (2005).
- [41] Grandgeorge, P. et al. Capillarity-induced folds fuel extreme shape changes in thin wicked membranes. Science **360**, 296–299 (2018).
- [42] Vella, D. Buffering by buckling as a route for elastic deformation. Nature Reviews Physics **1**, 425–436 (2019).
- [43] Siéfert, E. & Roman, B. Morphogenesis through elastic phase separation in a pneumatic surface. C. R. Mec. **348**, 649–657 (2020).
- [44] Kenny, M. C., Giarra, M. N., Granata, E. & Socha, J. J. How temperature influences the viscosity of hornworm hemolymph. J. Exp. Biol. **221**, jeb186338 (2018).
- [45] Lechantre, A. et al. Microrheology of haemolymph plasma of the bumblebee *Bombus terrestris*. J. Exp. Biol. **226**, jeb245894 (2023).
- [46] Elbaz, S. B. & Gat, A. D. Dynamics of viscous liquid within a closed elastic cylinder subject to external forces with application to soft robotics. J. Fluid Mech. **758**, 221–237 (2014).
- [47] Bambardekar, K., Clément, R., Blanc, O., Chardès, C. & Lenne, P.-F. Direct laser manipulation reveals the mechanics of cell contacts in vivo. Proc. Natl. Acad. Sci. U.S.A. **112**, 1416–1421 (2015).
- [48] Salcedo, M. K., Hoffmann, J., Donoughe, S. & Mahadevan, L. Computational analysis of size, shape and structure of insect wings. Biol. Open **8**, bio040774 (2019).
- [49] Salcedo, M. K., Jung, S. & Combes, S. A. Autonomous expansion of grasshopper wings reveals external forces contribute to final adult wing shape. Integr. Comp. Biol. **63**, 1111–1126 (2023).

Supplementary information

Wing ~~expansion~~ deployment in *Drosophila melanogaster*

Simon Hadjaje¹, Ignacio Andrade-Silva^{1,2}, Marie-Julie Dalbe³,
Raphaël Clément⁴, Joel Marthelot¹

¹Aix-Marseille University, CNRS, IUSTI & Turing Centre for Living Systems (CENTURI), Marseille, France

²Departamento de Física, Facultad de Ciencias Físicas y Matemáticas,
Universidad de Chile, Santiago, Chile

³Aix Marseille Univ, CNRS, Centrale Med, IRPHE, Marseille, France

⁴Aix-Marseille University, CNRS, IBDM & Turing Centre for Living Systems (CENTURI), Marseille, France

Macroscopic origami folding and vein network.

Newly eclosed fly exhibits highly folded wings. Supplementary Fig. S1a shows a folded fly wing (top: dorsal side; bottom: ventral side) with the veins used as landmarks throughout this study indicated in colors (see Fig. 2). ~~This stereotypical folding, which occurs 2 of the manuscript.~~ These stereotypical folds, which occur along the longitudinal veins but also in ~~a~~ the perpendicular direction (“marginal fold” at the proximal end of the wing [1]), unfold within minutes with equally stereotyped dynamics. Supplementary Fig. S1b shows an adult wing and the associated vein network ~~after the organ has undergone unfolding and expansion.~~

Wing deployment is a dual morphing mechanism that involves unfolding macroscopic folds and additional isotropic tissue stretching. In Supplementary Fig. 1c we quantify the macroscopic deployment $\Lambda = L/L_F$ in time (solid dark green line), with L the length of the straight segment connecting the two extremities of the longitudinal vein shown in green in Supplementary Fig. 1a-b, and L_F the corresponding length in the folded state. We compare this dynamics of unfolding with the dynamics of the tissue stretch $\lambda = l/l_F$ (shown as the dashed green curve), where l is the arclength of the same green vein, and l_F the corresponding folded arclength. We note that the two mechanisms appear to be simultaneous.

Quantification of curvature during deployment.

Wing deployment is characterized by a complex change of curvature. The wing is initially flat when folded, then curves downwards as soon as it begins to unfold. It then returns to a flat, expanded plate, before reversing its curvature slightly at the very end of deployment in the transverse direction. To quantify this change in curvature, we track the edge of the wing on side- and top-view recordings (see Supplementary Fig. 2a-b). The side-view perspective gives the wing profile represented by a dotted line in the inset of Supplementary Fig. 2a, and the

Supplementary Figure 1: Folded and deployed wings with their corresponding vein network. (a) Dorsal and ventral side of a folded wing. Most of the veins follow the macroscopic folds and are located either on a mountain or in a valley fold. (b) Adult wing with its anterior edge at the top. The highlighted veins in a-b are the one that can also be visualized on the micro-CT scans. (c) Wing deployment Λ and stretch λ as a function of time during the deployment process.

Supplementary Figure 2: Curvature during deployment in wild-type wings. (a) Side-view and (b) top-view profiles of the marginal vein (i.e. the vein that forms the contour of the wing). Time is color-coded from light to dark green. (c) 3D reconstruction of a wing during deployment.

top-view gives the marginal wing contour (solid line in the inset). The different colors of the wing profile in Supplementary Fig. 2a correspond to different times during deployment (from 4:00 in light gray to 8:30 in dark green) and show that the wing curves downwards during deployment. Supplementary Fig. 2b shows the top-view of the wing silhouette during deployment from a compact folded structure (light gray) to a fully deployed wing blade (dark green). The corresponding schematic 3D reconstruction of the wing during deployment is shown in Supplementary Fig. 2c.

Other changes in curvature during wing deployment can also be observed in mutant flies such as *Curly* mutants, which are widely used to track and identify mutations. When working with genes that are not associated with immediately visible phenotypes, these mutations are often combined with the *Curly* mutation to highlight the mutation of interest through macroscopic observation. A few studies have attempted to elucidate the physical mechanism leading to the *Curly* phenotype, but a mechanical description remains missing. Using transmission

Supplementary Figure 3: *Curly* mutant wing deployment at the organ level. Snapshots of *Curly* wing from a top and lateral view during deployment.

electron microscopy, Hurd et al. [2] observed that the wings of *Curly* mutants occasionally show abnormal pairing between the dorsal and ventral cuticle. One hypothesis is that these anomalies lead to a reduction in the surface area of the dorsal wing, causing the wing to curve upwards.

We characterize the kinematics of *Curly* flies during deployment (see Supplementary Fig. 3). Although the initial macroscopic folding pattern of the wing is identical to that of the wild-type, the wings unfold out of plane when deployment begins. In adults, the wings are characterized by a permanent upward curvature. Adult *Curly* flies are therefore unable to fly, and move around in small hops.

Supplementary Figure 4: Wing curvature in wild type, *Curly* and artificially inflated wings. (a) Shapes of wild-type, *Curly* and inflated wings during the deployment process. (b) Evolution of the average side-view curvature of the wing k as the wing deploy. The vertical dashed line corresponds to the profile in (a).

We now quantitatively evaluate the difference in curvature dynamics between wild type fly, *Curly* mutant and artificial inflation of wild-type fly. We fit the side-view with a circle to calculate the average curvature

$k = 1/R$ (with R the radius of the fitted circle). Supplementary Fig. 4a shows the side-view of the marginal vein 3 minutes after the onset of wing deployment for a wild-type (green), *Curly* (yellow) and artificially inflated fly (brown). As shown in Supplementary Fig. 4b, the wild-type wing is initially flat when folded ($k = 0$), and curves downwards as soon as it begins to unfold ($k < 0$). It then returns to a flat, expanded plate ($k = 0$). For both *Curly* mutants and artificial inflation, the wing immediately curves upwards ($k > 0$) at the beginning of deployment, and reaches a plateau: the adult fly has a positive curvature. While artificially inflated wings qualitatively resemble the phenotype observe in *Curly* wings, they are quantitatively different, *Curly* wings being 2 to 3 times more curved than artificially inflated fly wings.

Geometrical parameters measured from micro-CT scan.

We use ImageJ to extract the geometrical parameters of the folded wing from micro-CT scans. As sketched in Fig. 1b of the manuscript, the folded wing is composed of two plates of thickness e connected through pillars of diameter d , height h and organized in a hexagonal lattice with a spacing a . We apply thresholding, ultimate points and Voronoi tessellation on high-resolution scans (see Fig. 1b(iii) of the manuscript, 1 voxel = $0.32 \mu\text{m}$) to measure the diameter of the pillars $d \approx 3.3 \mu\text{m}$ and the distance between them $a \approx 6.7 \mu\text{m}$. We take advantage of lower-magnification scans (see Supplementary Fig. S5b(ii) and Fig. S5a, 1 voxel = $0.8 \mu\text{m}$) to measure the plates thickness e and the gap between them h . We obtain a first binary mask by applying a threshold on the upper and lower plates (see Supplementary Fig. 5b). The open surface area of this mask thus gives $S_0 = 2el$ where l is the arclength of the section. We then apply a closing process to obtain the full silhouette of the section (see Supplementary Fig. 5c). We obtain l by performing a skeleton process on this last mask and get the average plate thickness $e = S_0/(2l) \approx 6.5 \mu\text{m}$. The close surface area of the mask also yields the surface $S = (2e + h)l$, which allows to calculate the average gap $h = (S - S_0)/l \approx 7.5 \mu\text{m}$. The total thickness of the wing (composed of the plates thickness and the gap in between) while the wing is deploying is obtained with two-photon imaging (see Supplementary Fig. S5d and Supplementary Movie 7).

Supplementary Figure 5: micro-CT and two-photon post-processing. (a-c) micro-CT: (a) Original cross-section normal to the proximo-distal axis of a folded wing. (b) Threshold highlighting the top and bottom plates. (c) A closing process yields the full silhouette of the section. (d) Two-photon: Cross section of a deploying wing.

Volcano-like shape of the apical cell surface.

Supplementary Figure 6: Volcano-like shape of the apical cell surface. (a) Brightfield micrographs showing the contour of epithelial cells (highlighted in white) and wing hairs (dark shadows) in folded wings. (b) Utrophin:GFP signal epithelial cells in folded wings. (c) Segmentation of the cell inside the white square in (b). (d-f) 3D reconstruction of the apical surface of single cell.

Before deployment, the cuticle is wrinkled, and each wrinkle corresponds to a cell whose apical surface has a particular 3D arrangement. Bright-field microscopy images already reveal the buckled contours of the cells in the folded wings (see Supplementary Fig. 6a). The wing hairs appear as black dots in the center of each cell. The outline of a single cell is highlighted in white: it has the shape of a star, with each branch intertwined with neighboring cells. Utrophin:GFP folded wings, shown in Supplementary Fig. 6b, allow the 3D shape of individual cells to be segmented. We crop a region of interest that captures a cell and threshold to obtain only the shape of the cell (see Supplementary Fig. 6c). We perform this process on the entire stack covering the cell and reconstruct the 3D shape of the cell by re-stacking the segmented images shown in Supplementary Fig. 6d-f. In folded wings, the apical surface of the cells is shaped like a volcano or truncated cone, with straight ridges on the sides. This shape is isometric to a plane and allows the cuticle to unwrinkle without stretching as the wing surface expands.

Isotropic in-plane properties of the tissue.

The potential anisotropy of the tissue cannot be measured directly by tensile methods due to the initial macroscopic folding of the wing, so we have to rely on indirect observations. We assume that the elastic response of the wing is isotropic in-plane at the mesoscopic scale from three observations:

(1) The wing presents a highly regular three-dimensional structure, characterized by volcano-like wrinkles arranged in a hexagonal tiling associated with the hexagonal organization of epithelial cells. To highlight this

hexagonal tiling, we perform a Fourier transform of the folded and expanded wing surface based on profilometric measurements of the wing topography (see Supplementary Fig. 7). In both cases, we find a hexagonal organization of the volcano-like wrinkle characterized by 60-degree angles in reciprocal space, with a larger pattern wavelength for the expanded wing. While this hexagonal organization might suggest an anisotropic response of the structure, we observe that the peaks in inverse space remain on a circle in the deployed wing, indicating that the extension is isotropic in-plane.

(2) At the macroscopic scale, we characterize the stretching of the veins as the wing expands under an isotropic pressure load. We observe that veins oriented in different directions exhibit the same stretch, indicating that extension is isotropic in the plane (see Fig. 2c of the manuscript).

(3) At the microscopic scale, we have imaged the contour of epithelial cells using spinning-disk confocal microscopy and Ecad:GFP in Fig. 3d of the manuscript. In the intervein region, we observe that while cell surface area increases during expansion under an isotropic pressure load, cell shape remains isotropic hexagonal after expansion.

Supplementary Figure 7: Volcano-like wrinkle hexagonal tiling in folded (a) and deployed (b) wings. (i-ii) Topography of the wing surface; (iii) Fourier transform of the wing surface topography. Peaks in reciprocal space with angle of $\approx 60^\circ$. Distance from the center gives the inverse of the pattern wavelength.

Wings mechanical properties measured through tensile tests.

Supplementary Fig. 8a shows snapshots of a wild-type fly that begins to expand its wings, but whose internal pressure decreases before the process is complete. We observe that the wings fold back, imperfectly, but with their longitudinal vein folds and marginal folds. This reversibility of folding prompts us to use an elastic description to model the deformation of the structure.

Tensile tests are conducted on dissected folded fly wings to characterize their mechanical properties. A dis-

Supplementary Figure 8: Tensile experiment. (a) Failure of wing expansion in a wild-type fly with elastic recovery. (b) Pictures of a tensile test experiment with the folded wing glued at both distal and proximal extremities. The followed vein is underlined in green. (c) True stress $\sigma = \lambda F / S_0$ versus true stretch $\ln(\lambda) = \ln(l/l_F)$ for all of the six experiments (data: colored markers; corresponding linear fit: colored solid lines; averaged fit: dashed black line). (d) FEM numerical simulation of the stretching of a wrinkled bilayer and computation of the associated stress-stretch. Color code is the strain in the tensile direction ϵ_{xx} . Red dotted line: theoretical prediction of $\sigma - \lambda$ relationship for the substrate alone.

placement is imposed on the proximal end of the wing, which is attached to a linear stage. The force, F , is measured using a load sensor attached to the distal end of the wing (see picture-pictures of the experiment Figin Supplementary Fig. S8a)–8b). Supplementary Fig. S8b shows the 8c shows the true stress for six experiments (colored markers) as a function of the stretch-true strain obtained by direct measurement of the variation in one of the longitudinal vein (shown in FigSupplementary Fig. S8a8b) and their associated linear fit at small strains (colored lines), which yield the Young’s modulus E . The average linear fit is shown as a dashed black line.

The strain stiffening of a wrinkled bilayer as it unwrinkles is illustrated using 2D FEM numerical simulations (COMSOL Multiphysics). The bilayer consists of a soft $10 \mu\text{m}$ thick substrate of epithelial cells with a Young’s modulus $E = 100 \text{ kPa}$ covered by a 200 nm thick rigid film of cuticle with a Young’s modulus $E_f = 100 \text{ MPa}$. To improve convergence, we consider a hyperelastic Gent model for the substrate with an arbitrarily large value of $J_m = 100$, which has a negligible impact for the values of stretching considered in the simulation. We impose symmetric boundary conditions on the bottom and left sides of the system while we prescribe the displacement on the right side. The top surface is free.

The initial state is obtained by thermally contracting the substrate to form wrinkles (see FEM snapshot, bottom left in Supplementary Fig. S8e)–8d). This numerical method is commonly used to study instabilities resulting

from differential growth in biological structures [3]. Thermal expansion and biological growth are rigorously kinematically equivalent [4]. To obtain the initial state of the wrinkled cuticle bilayer, we define a coefficient of thermal expansion in the longitudinal direction x of the epithelial cell substrate with a thermal strain $\epsilon_{tb} = \alpha_{xx}\Delta T$, where α is the coefficient of thermal expansion. When a finite temperature difference ΔT is applied, the substrate shrinks along the longitudinal direction, so that the film becomes comparatively too long and buckles.

We then impose a displacement ~~to~~ on the right boundary ~~condition~~. Supplementary Fig. S8e-8d shows the normal stress along the right boundary ~~condition~~ as a function of the stretch, λ . The curve ~~exhibits~~ shows two distinct regimes: (i) at moderate stretch ($\lambda < 1.5$), stress is low and is ~~mainly~~ dominated by the stretching of the soft substrate, while the unwrinkling of the rigid film has a minimal impact on the overall stiffness of the bilayer. ~~Quantitatively, the effective stretch modulus of the bilayer is similar to the elastic response of the substrate (rod dotted line)~~; (ii) as the wrinkles disappear (intermediate snapshot $\lambda \sim 1.5$), the system becomes stiffer and the mechanical response is dominated by the stretching of the rigid film. ~~This~~

~~These~~ simulations illustrate the ~~overall global~~ strain stiffening of the wing observed in the experiments, which ~~stems results~~ from microscopic unwrinkling of the cuticle, and support the choice of the Gent's hyperelastic model to account for the effective mechanical response of the ~~composite tissue of the wing bilayer forming the wing~~.

Nanoindentation of the cuticle of adult wings. The adult wing is composed of a thin flat layer of sklerotized chitin with a total thickness of approximately 500 nm in the intervein regions. We perform nanoindentation experiments on fully deployed adult wings. The wings are dissected and placed on a thin layer of superglue coated on a glass plate. We impose loading and unloading force cycles and obtain the $F_i - \delta$ curve shown in the Inset of Supplementary Fig. 9a. We perform multiple indentations on different areas of adult wings on 5 different samples.

Supplementary Figure 9: Nanoindentation of adult wing. (a) Force-displacement indentation of an adult wing on a logarithmic scale (unloading phase only). Inset: raw $F_i - \delta$ curve. **(b)** Histogram of Young's modulus E_c of adult wings. Each color corresponds to a different fly, the black line is the cumulative histogram.

In Supplementary Fig. 9a, we plot on a logarithmic scale the elastic discharge in order to exclude plastic effects observed during loading. The curves line up with a slope of 3/2 in agreement with the prediction of the Hertz

contact model for a rigid spherical probe indenting a soft, elastic, planar sample:

$$F_i = \frac{4}{3} \frac{E_c}{1 - \nu^2} R^{1/2} \delta^{3/2} \quad (1)$$

with E_c the cuticle Young's modulus, $\nu = 0.5$ its Poisson's ratio, and R the radius of the indenter. We invert this expression to extract the Young's modulus E_c of each sample.

All measurements are shown in Supplementary Fig. 9b, with each color corresponding to a different fly. Different data for a fly correspond to different indentation times and positions. The cumulative histogram is represented by the black contours and shows a peak at $E_c \approx 1$ GPa. The dispersion observed from one wing to the next can be explained by the fact that the cuticle undergoes sclerotization after deployment, which tends to increase cuticle stiffness [5]. The stiffness measured is very consistent with AFM measurements on adult wings in the literature [6].

Model parameters.

The wing expansion model depends on both mechanical parameters (Young's modulus E , Gent limiting value J_m) and geometrical features (plates thickness e , pillars height h , diameter d , interpillar distance a). The aforementioned parameters are quantified through microscopic characterization and mechanical testing. However, it is of interest to assess the impact of these parameters on model predictions, and the robustness of predictions to small parameter variations.

Supplementary Fig. S10a illustrates that at small strain ($\lambda < 1.5$), the linear elastic law does not differ greatly from a hyperelastic material. A purely linear elastic model predicts a finite pressure at which stretching, in both horizontal and vertical directions, diverges. We therefore proceed to opt for a more realistic model that incorporates the effects of large strains and non-linearities (strain-stiffening) in the material [7]. The value of J_m is determined by matching predictions of the strain-stretch in the pillars λ^p (Supplementary Fig. S10b) with experimental observations. The maximum pillar strain-stretch measured *in vivo* is $\lambda^p \sim 4.8$. A value of $J_m = 20$ corresponds well to this limiting value at working pressure $\bar{P} \sim 0.18$. To keep the model simple, we have chosen to use the same parameter J_m for the biaxial expansion of the plates.

Supplementary Figures S10c-d illustrate the influence of geometry on the prediction for the stretch-in-the-plates in-plane stretch λ (top graphs) and the stretch in the pillars in λ^p (bottom). We note in particular that more slender pillars (i.e. $\Psi \sim 1$ or $\Phi \sim 0$ in terms of non-dimensional parameters) result in a shift of the curves towards lower working pressure.

To validate our model, we perform FEM numerical simulations of the inflation of a wing-shaped structure composed of a hyperelastic Gent material ($E = 100$ kPa, $J_m = 20$). Supplementary Fig. S11a shows simulation predictions of the in-plane stretch λ as a function of the normalized applied pressure \bar{P} (dotted lines) for which we vary the system size (36, 64 and 37 pillars) and pillar arrangement (square and hexagonal lattice, adjusting the distance a between the pillars to keep the pillars density $\Phi = 0.22$ constant, see FEM snapshots Supplementary Fig. S11b). All predicted λ follow the same curve, which shows that (i) the system is sufficiently large for boundary effects to be negligible; and (ii) confirms that pillar organization has no key role in the model, the important parameter being the pillar density Φ . It should be noted that the our simple model (shown in black solid line

Supplementary Figure 10: Model and influence of the mechanical and geometric parameters. (a) Elastic vs hyperelastic constitutive law: (top) in-plane normalized stress σ/E as a function of the in-plane stretch λ and (bottom) perpendicular normalized stress $(\sigma^p + P)/E$ as a function of the pillar stretch λ^p for an elastic constitutive law (dotted line) and hyperelastic Gent models (solid lines, different parameters J_m). (b-d) (top) in plane stretch λ and (bottom) perpendicular stretch λ^p versus normalized pressure \bar{P} varying parameters J_m (b), pillars relative height Ψ (c), and in-plane pillars density Φ (d).

Supplementary Fig. S11a) does not take into account the **actual connection of the pillars to the membrane, which explains deformation offset at the connection between the pillars and the plates, (i.e. during inflation, the plates stretch while the pillars shrink in the xy -plane), which is likely to contribute to the discrepancy observed with the FEM.**

Viscous dissipation probed through nanoindentation experiments

We perform creep tests and loading-unloading cycles in nanoindentation to measure the wing material viscous dissipation. Supplementary Fig. S12a-c show three creep tests for which a constant force F_i is applied (see Supplementary Fig. S12d for the loading signal). Each curve is fitted by a three elements Maxwell-Jeffrey model (yellow line), yielding to a short timescale $\tau_1 = 1.9 \pm 0.3$ s, a long relaxation time, $\tau_2 = 37 \pm 12$ s and an effective elastic response of the bilayer in indentation $E_i \sim 1.6$ MPa.

Loading-unloading cycles tests in nanoindentation – for which the long relaxation time can be neglected at the time scale of the experiments – are well captured by a Kelvin-Voigt model as shown Supplementary Fig. S12e. The experimental data (represented with colored markers, using 10 bins per sample to average all data series) follow the model $\epsilon = (t/T)^{2/3}$, where $T = (2R^2 E_i \tau / (3\dot{F}_i))^{1/2}$, with a single value, $E_i \tau$ being fitted to the curves. To confirm the model, we perform FEM numerical simulations of indenting a bilayer composed of a thin elastic film of Young's modulus $E_f = 100$ MPa deposited onto a soft viscoelastic substrate ($E = 100$ kPa, $\tau = 10$ s) at

Supplementary Figure 11: FEM simulations of inflating a hyperelastic wing-shaped structure. (a) Stretch λ as a function of the normalized applied pressure \bar{P} : all FEM predictions (dashed lines) varying the size of the system (36, 64 and 37 pillars) and the pillars organization (square and hexagonal lattice) collapse on the same curve. The model is shown as the black solid line. (b) FEM snapshots at different pressure of 1/4th of the geometry for (i) 36 pillars organized in a square lattice, and (ii) 37 pillars organized in a hexagonal lattice. Color code: strain in one of the in-plane principal direction.

different loading rates (10^{-1} to $10^0 \mu\text{N/s}$). The simulations, shown as dashed line Supplementary Fig. S12e, also follow the master curve.

***in vivo* pressure recording and artificial pressure increase**

Supplementary Fig. S13 shows the temporal evolution of pressure P (black curve) and stretch λ (red) for four distinct flies. We observe that pressure initially increases, reaching a plateau of $\approx 1.5 - 3.5$ kPa during which most – at least 50% – of the wings deployment occurs.

We wonder whether wing deployment can be triggered by an external artificial increase of pressure. To test this hypothesis, a wild-type fly is placed under ether vapor for 10 minutes immediately after emerging from the pupal case. We then puncture its scutellum using a glass capillary (50-75 μm outside diameter) connected to a syringe pump and a pressure sensor. PBS is injected to impose different pressure plateaus and the fly is observed for at least 2 minutes to identify any sign of expansion-deployment before increasing the pressure. No deployment of the wings-wing deployment is observed below the pressure plateau of $P \sim 10$ kPa. Between $P = 10 - 16$ kPa, the wings deploy in about 20 minutes and exhibit curly wings. -10 minutes. Supplementary Fig. S14a and Supplementary Movie 10 show top- and side-view snapshots of such an experiment. The wings curved upwards. A physical reason for curvature may be a post-mortem difference in the elastic properties of the dorsal and ventral plates. Such asymmetry would result in curvature under pneumatic actuation, with the softer plates stretching more than the stiffer one, a property widely used in soft robotics to achieve curvature [8]. At higher pressure ($P > 17$ kPa), the pillars break, the ventral and dorsal layers delaminate, resulting in blistered or even balloon like-balloon-like wings (see Supplementary Fig. S14b). Note that at this stage, increasing the pressure does not further stretch-stretches the tissue, which further justifies the use of a hyperelastic strain-stiffening model to describe the wing.

Supplementary Figure 12: Nanoindentation on folded wings. (a-c) Deformation $\epsilon = (\delta/R)^{1/2}$ as a function of time for three different creep experiments (experimental data: in black; Maxwell-Jeffrey fit: yellow line), each one undergoing a different force plateau F_i (see (d) for the corresponding loading signals). (e) ϵ versus normalized time t/T from nanoindentation experiments. Data (colored markers, using 10 bins per sample to average all experiments, standard deviation shown with errorbars) and FEM (dashed lines, from slow indentation $\dot{F}_i = 10^{-1} \mu\text{N/s}$ in dark green, to faster $\dot{F}_i = 10^1 \mu\text{N/s}$ in gray) collapse on the model $\epsilon = (t/T)^{2/3}$ (black line). (f) FEM numerical simulation of the indentation of a bilayer composed of a visco-elastic cell substrate (Young's modulus $E = 100 \text{ kPa}$, internal time $\tau = 10 \text{ s}$) covered with an elastic cuticle film (thickness $t = 200 \text{ nm}$, Young's modulus $E_f = 100 \text{ MPa}$). This snapshot is taken as the indenter (radius $R = 4.7 \mu\text{m}$) imposes a force $F_i = 5 \mu\text{N}$ at rate $\dot{F}_i = 10^{-1} \mu\text{N/s}$, corresponding to the dark green curve in (b).

Description of supplementary movies

Supplementary Movie S1.1. Wing deployment (wild type female *Drosophila*) from top and side views (timer mm:ss).

Supplementary Movie S2.2. Side view of a fly increasing its internal pressure by swallowing air (white arrow indicates pharyngeal pumping) and contracting abdominal muscles.

Supplementary Movie S3.3. Micro-CT scan of a folded wing (ImageJ plugin 3D viewer). Z-stack of cross sections of the same wing normal to the proximo-distal axis reveals the internal structure (i.e. dorsal and ventral plates connected by pillars).

Supplementary Movie S4.4. Binocular fluorescent microscopy of hemolymph flows labeled with fluorescent beads during deployment (left) and in an adult wing (right). Hemolymph invades the entire structure during deployment, in stark contrast to hemolymph flow in an adult wing, where it is restricted to the network of veins.

Supplementary Movie S5.5. *in vivo* measurement of internal pressure during wing deployment. The scutellum of a newly emerged fly is poked with a glass capillary connected to a pressure sensor, and wing deployment is recorded. Inset: measured pressure $P(t)$ over time.

Supplementary Movie S6.6. Combining recordings of wing deployment in 6 individuals shows reproducibility of the process (3 females above, 3 males below). The time at which the wings begin to spread-deploy is chosen as the reference for combining the videos.

Supplementary Figure 13: *in vivo* pressure measurements. Pressure recording (black curves) in 4 different flies and corresponding stretch λ (red) in time as the wings deploy. Most of the wing deployment happens on a pressure plateau of 1.5 – 3.5 kPa.

Supplementary Figure 14: Artificial pressure increase. (a) A pressure plateau of $P = 15$ kPa is applied in a fixed wild type fly with folded wings, triggering curved wings **expansion-deployment** in ~ 10 minutes. (b) An additional increase in the pressure to $P = 17$ kPa leads to microtubule pillars breakage and wing blade delamination resulting in **blister-blistered** or ballon-like wings.

Supplementary Movie S7.7. 3D reconstruction of the distal tip of a wing during deployment obtained with a two-photon microscope, enabling a measurement of the wing thickness *in vivo* **during deployment** of $\sim 35 \mu\text{m}$ (**compare with the during deployment (while the thickness is $\sim 18 \mu\text{m}$ thick wing** before deployment, see TEM Fig. 3b and quantification boxplot "wing thickness" Fig. 3e).

Supplementary Movie S8.8. 3D reconstruction of the apical surface of a single cell (Utrophin:GFP) from a z-stack obtained by fluorescent spinning disk microscopy. Integration of the 3D shape enables measurement of the apical cell surface area (see Folded condition of the boxplot "3D Area", Fig. 3e).

Supplementary Movie S9.9. Tensile test on a dissected folded wing. The distal end of the wing is connected to a force sensor, while we impose a displacement on the proximal end glued to a moving stage (see Fig. 4b and **Supplementary** Fig. S8a for the measured stress-strain curves).

Supplementary Movie S10.10. An artificial increase in pressure triggers wings deployment. We poke the scutellum of a sacrificed, newly emerged wild type fly with a glass capillary connected to a syringe **and**. **We** impose a first pressure plateau of 15 kPa, at which point the wings unfold and curl upward. At $t = 23$ minutes a further

increase in the pressure plateau to 17 kPa leads to rupture of the microtubule pillars and delamination of the dorsal and ventral layers, resulting in balloon-like wings (see Supplementary Fig. S14 for corresponding snapshots).

Supplementary Movie S11.11. 3D FEM simulations (COMSOL Multiphysics) of the inflation of wing-like structure composed of two plates of thickness $e = 6.5 \mu\text{m}$ connected by pillars (height $h = 7.5 \mu\text{m}$, diameter $d = 3.3 \mu\text{m}$, interpillar distance $a = 6.2 \mu\text{m}$). The visco-hyperelastic material ($E = 100 \text{ kPa}$, $J_m = 20$, $\tau = 10 \text{ s}$) is submitted to a pressure step of $P = 16 \text{ kPa}$ at $t = 0$. Color indicate in-plane strain.

Supplementary Movie 12. Wing deployment in a *Curly* mutant from a top- and side-view. Scale bar: 1 mm; timer mm:ss.

Supplementary Movie 13. Elastic recovery after the interruption of wing deployment in a wild-type fly. Scale bar: 1 mm; timer mm:ss.

References

- [1] Alice Tsuboi, Koichi Fujimoto, and Takefumi Kondo. Spatiotemporal remodeling of extracellular matrix orients epithelial sheet folding. *Sci. Adv.*, 9(35):eadh2154, 2023.
- [2] Thomas Ryan Hurd, Feng-Xia Liang, and Ruth Lehmann. Curly encodes dual oxidase, which acts with heme peroxidase curly su to shape the adult drosophila wing. *PLoS genetics*, 11(11):e1005625, 2015.
- [3] Thomas Lessinnes, Derek E Moulton, and Alain Goriely. Morphoelastic rods part ii: growing birods. *Journal of the Mechanics and Physics of Solids*, 100:147–196, 2017.
- [4] Gareth Wyn Jones and S Jonathan Chapman. Modeling growth in biological materials. *Siam review*, 54(1):52–118, 2012.
- [5] Svend Olav Andersen. Insect cuticular sclerotization: a review. *Insect biochemistry and molecular biology*, 40(3):166–178, 2010.
- [6] Ryan Wagner, Barry R Pittendrigh, and Arvind Raman. Local elasticity and adhesion of nanostructures on drosophila melanogaster wing membrane studied using atomic force microscopy. *Applied surface science*, 259:225–230, 2012.
- [7] Emmanuel Siéfert and Benoît Roman. Morphogenesis through elastic phase separation in a pneumatic surface. *C. R. Mec.*, 348(6-7):649–657, 2020.
- [8] Filip Ilievski, Aaron D Mazzeo, Robert F Shepherd, Xin Chen, and George McClelland Whitesides. Soft robotics for chemists. *Angewandte Chemie International Edition*, 2011.

Paper: Wing expansion in *Drosophila melanogaster*

Authors: S. Hadjaje, I. Andrade-Silva, M.-J. Dalbe, R. Clément, and J. Marthelot

Paper summary: The authors present an impressive series of experiments to measure the final stage of the development of wings in *Drosophila melanogaster*: the inflation of the wings to unfold them for flight. This unfolding is achieved by increasing the pressure in the hemolymph, which causes the wings to effectively ‘stretch’. This apparent stretching is proposed to be caused by the ironing-out of wrinkles at a cellular scale. They proceed to measure the effective mechanical properties of the wing by applying tensile tests on a dissected wing and indentation tests on wing tissue. They then compare their observations to FEM simulations of a hyperelastic Gent model. Finally, the authors consider the dynamics of this inflation and propose that the tissue’s behavior has a significant viscoelastic component.

Referee Comments: This work provides novel insight into the expansion of insect wings and is likely to be of broad interest to the readership of *Nature Communications*. Below, I have provided comments and questions that I would like the authors to address before this paper can proceed further.

My main comments center on some of the numbers given by the authors for various quantities. These are intended as support for the physical picture they present; however, there is often not much detail of how these estimates are arrived at. Two of these seem particularly important to me.

1. It’s unclear what the Young’s modulus measured in the tensile tests (*i.e.* $E \approx 100\text{kPa}$) corresponds to physically. Given its small magnitude, it seems very unlikely that it is actually a material property of the insect wing. (I assume the wing is largely chitin, which has a Young’s modulus on the order of GPa, see [doi:10.1039/C9NR02870F](https://doi.org/10.1039/C9NR02870F), for example.) The authors should clarify this, and whether they believe they have measured an effective stretching modulus and what its origin is. For example, if it is not a material modulus, then could it be caused by some adhesion between initially contacting folds that requires a force to unfold, or perhaps intrinsic curvature could have a similar effect? Similarly, could the authors comment on whether the measured stiffness is isotropic or direction-dependent (as one might expect from unfolding wrinkles). Finally, similar phenomenology has been reported before in other systems and has been referred to as the changes in length being ‘buffered’ by buckling (see [doi:10.1126/science.aag0677](https://doi.org/10.1126/science.aag0677), for example).
2. Similarly, the authors assume that there must be viscoelastic evolution because the dynamics of inflation are much slower than the time scale $\eta_f L^2 / (E h e)$. I am not particularly familiar with ref. [40], so it would be good to outline the essential ingredients for this estimate. As an alternative, I wondered whether older work on airway reopening by Heil and co-workers (see for example [doi:10.1017/S0022112002003452](https://doi.org/10.1017/S0022112002003452)) gives a similar estimate for the inflation time. Could the authors comment on this time scale and whether the slow peeling dynamics that Heil considers are accounted for in ref. [40]?

I also have some more minor comments:

3. As I understand it, the authors are proposing that the material stretching is minimal in the wing expansion (*i.e.* there is no real strain of the material at the microscopic scale). Instead, the observed evolution is due to a combination of unfolding of macroscopic folds and microscopic (cell-scale) wrinkles. To me at least, it was unclear when the authors were referring to an effective/apparent stretching of the tissue compared to a material stretching. Could the authors clarify when they are referring to macroscopic vs. cellular observations? For example, I believe the section ‘The wing not only unfolds but

also stretches' refers to macroscopic observations of effective stretching, whilst 'Apparent stretching relies on unwrinkling at the cellular scale' explores the cause of these macroscopic observations at a cellular level.

4. I am also a little concerned by the model the authors present for the effective stretching of the tissue using an isotropic hyperelastic model. Could the authors comment on the validity of this model since here the material is unwrinkling not stretching, is this process isotropic? or elastic? Furthermore, the authors have also assumed in-plane equibiaxial extension and incompressibility. I wouldn't at first expect that these assumptions hold since the pillars break the in-plane symmetry and unwrinkling isn't necessarily incompressible.
5. The 'volcano-like' 3D microscopic structure is mentioned a couple of times in the main text, but only shown in the SI. I think it would be useful to have an image in the main text too.
6. Before equation (19a), "dimentionnal" should be "dimensional".